# Near-Optimal Regret for Adversarial MDP with Delayed Bandit Feedback

**Tiancheng Jin**
University of Southern California
tiancheng.jin@usc.edu

**Tal Lancewicki**
Tel Aviv University
lancewicki@mail.tau.ac.il

**Haipeng Luo**
University of Southern California
haipengl@usc.edu

**Yishay Mansour**
Tel Aviv University and Google Research
mansour.yishay@gmail.com

**Aviv Rosenberg**[*]
Amazon
avivros@amazon.com

## Abstract

The standard assumption in reinforcement learning (RL) is that agents observe feedback for their actions immediately. However, in practice feedback is often observed in delay. This paper studies online learning in episodic Markov decision process (MDP) with unknown transitions, adversarially changing costs, and unrestricted delayed bandit feedback. More precisely, the feedback for the agent in episode $k$ is revealed only in the end of episode $k + d^k$, where the delay $d^k$ can be changing over episodes and chosen by an oblivious adversary. We present the first algorithms that achieve near-optimal $\sqrt{K + D}$ regret, where $K$ is the number of episodes and $D = \sum_{k=1}^{K} d^k$ is the total delay, significantly improving upon the best known regret bound of $(K + D)^{2/3}$.

## 1 Introduction

Delayed feedback has become a fundamental challenge that sequential decision making algorithms must face in almost every real-world application. Notable examples include communication between agents [9], video streaming [8] and robotics [32]. Broadly, delays occur either for computational reasons, e.g., in autonomous vehicles and wearable technology, or when they are an inherent part of the environment like healthcare, finance and recommendation systems.

Although a prominent challenge in practice, there is only limited theoretical literature on delays in reinforcement learning (RL). Recently, [18] studied regret minimization in episodic Markov decision processes (MDPs) but assume that the delays (and costs) are *stochastic*, i.e., sampled i.i.d from a fixed (unknown) distribution, which is a limiting assumption since it does not allow dependencies between costs and delays that are very common in practice. The case of *adversarial* delays and costs was also studied recently [28]. However, they focus on *full-information* feedback where the learner observes the entire cost function, which is not realistic in many applications, and obtain only sub-optimal regret bounds for *bandit* feedback (where the learner observes only the costs on the traversed trajectory).

In this paper we significantly advance our understanding of delayed feedback in adversarial MDPs with bandit feedback. More precisely, we consider episodic MDPs with unknown transition function,

---

[*]Research conducted while the author was a student at Tel Aviv University.

36th Conference on Neural Information Processing Systems (NeurIPS 2022).

Table 1: Regret bounds for Adversarial MDPs with unknown transition and unrestricted delayed bandit feedback. $K$ is the number of episodes, $D$ is the total delay, $H$ is the horizon, $S$ is the number of states and $A$ is the number of actions. Algorithms presented in this paper appear in grey.

| Algorithm | Regret | Efficient | Regret w.h.p |
|---|---|---|---|
| D-OPPO [28] | $\widetilde{O}(HS\sqrt{A}K^{2/3} + H^2 D^{2/3})$ | ✓ | ✓ |
| Delayed Hedge | $\widetilde{O}(H^2 S\sqrt{AK} + H^{3/2}\sqrt{SD})$ | ✗ | ✓ |
| Delayed UOB-FTRL | $\widetilde{O}(H^2 S\sqrt{AK} + H^{3/2}SA\sqrt{D})$ | ✓ | ✗ |
| Delayed UOB-REPS | $\widetilde{O}(H^2 S\sqrt{AK} + (HSA)^{1/4} \cdot H\sqrt{D})^*$ | ✓ | ✓ |
| Lower bound [28] | $\Omega(H^{3/2}\sqrt{SAK} + H\sqrt{D})$ | | |

$^*$Under unknown dynamics Delayed UOB-REPS has an additional additive term in the regret that scales linearly with $d_{max}$. One can avoid the dependency in $d_{max}$ but with a slightly weaker bound than the one that appears in this table - for more details see Remark D.1 in the supplementary material.

adversarially changing costs (bounded in $[0, 1]$) and unrestricted delayed bandit feedback, i.e., the learner observes the costs suffered in episode $k$ only in the end of episode $k + d^k$ where the sequence of delays $\{d^k\}_{k=1}^K$ are chosen by an oblivious adversary. We develop the first algorithms for this setting that achieve near-optimal regret and provide a major improvement over the currently best known regret bound [28] - see Table 1 for more details.

In the following paragraph we provide an overview of our contributions and the structure of the paper. In Section 3 we devise an inefficient Hedge [13] based algorithm that treats every deterministic policy as an arm. This can be seen as a warm-up – a relatively simple and elegant solution that shows that order $\sqrt{K + D}$ regret is attainable with delayed bandit feedback. Moreover, our adaptation of Hedge to the setting of adversarial MDP with unknown transition and bandit feedback presents highly non-trivial algorithmic and technical features that may be of independent interest. Then, we focus on the pressing question: *Can delayed bandit feedback be handled both optimally and efficiently?* We answer this affirmatively by presenting two efficient algorithms with near-optimal regret. Through our unique analysis and algorithmic design, we shed light on the great challenges of handling efficiently delayed bandit feedback. In Section 4 we consider a relatively standard algorithm we call Delayed UOB-FTRL, based on the Follow the Regularized Leader (FTRL) framework, and focus on a unique novel analysis that may be of independent interest. As seen in Table 1, our analysis of Delayed UOB-FTRL shows regret similar to the inefficient Delayed Hedge. However, it has worse dependence on $S$ and $A$, and has regret guarantee on expectation rather than with high probability (w.h.p). In Section 5 we propose our final solution which is mainly algorithmic: we introduce the algorithm Delayed UOB-REPS that has a novel importance-sampling estimator which generalizes the standard estimator and accommodates it to the delays. This approach allows us to follow the path of more standard analysis, but most importantly, ensures w.h.p the best regret so far (see Table 1). The first term of the regret bound matches the best known regret for adversarial MDP with non-delayed bandit feedback [22], while the second term matches the lower bound of [28] up to a factor of $(HSA)^{1/4}$.

## 1.1 Additional Related Work

**Delays in RL.** While delays are popular in the practical RL literature [39, 30, 8, 32, 12], there is limited theoretical literature on the subject. Most previous work [26, 45] considered constant delays in observing the current state. However, the challenges in that setting are different than the ones considered in this paper (see [28] for more details). As discussed in the introduction, most related to this paper are the recent works of [28] and [18].

**Delays in multi-arm bandit (MAB).** Delays were extensively studied in MAB and optimization both in the stochastic setting [1, 43, 44, 34, 6, 49, 14, 29, 10], and the adversarial setting [35, 7, 41, 3, 52, 19, 15, 42]. However, as discussed in [28], delays introduce new challenges in MDPs that do not appear in MAB.

**Regret minimization in RL.** There is a rich literature on regret minimization in both stochastic [20, 2, 21, 23, 46, 47, 48] and adversarial [51, 36, 37, 38, 22, 25, 5, 40, 31, 24, 17] MDPs. Note that regret minimization in standard episodic MDPs is a special case of the model considered in this paper where $d^k = 0$ for every episode $k$.

## 2 Preliminaries

We consider the problem of learning adversarial MDPs under delayed feedback. A finite-horizon episodic MDP is defined by a tuple $\mathcal{M} = (\mathcal{S}, \mathcal{A}, H, p, \{c^k\}_{k=1}^K)$, where $\mathcal{S}$ and $\mathcal{A}$ are finite state and action spaces of sizes $|\mathcal{S}| = S$ and $|\mathcal{A}| = A$, respectively, $H$ is the horizon (i.e., episode length) and $K$ is the number of episodes. $p : \mathcal{S} \times \mathcal{A} \times [H] \to \Delta_{\mathcal{S}}$ is the *transition function* which defines the transition probabilities. That is, $p_h(s'|s, a)$ is the probability to move to state $s'$ when taking action $a$ in state $s$ at time $h$. $\{c^k : \mathcal{S} \times \mathcal{A} \times [H] \to [0, 1]\}_{k=1}^K$ are *cost functions* which are chosen by an *oblivious adversary*, such that $c_h^k(s, a)$ is the cost of taking action $a$ in state $s$ at time $h$ of episode $k$.

A *policy* $\pi : \mathcal{S} \times [H] \to \Delta_{\mathcal{A}}$ is a function such that $\pi_h(a|s)$ is the probability to take action $a$ when visiting state $s$ at time $h$. The value $V_h^{\pi, p'}(s; c)$ is the expected cost of $\pi$ with respect to cost function $c$ and transition function $p'$ starting from state $s$ in time $h$, i.e., $V_h^{\pi, p'}(s; c) = \mathbb{E}^{\pi, p'}\left[\sum_{h'=h}^H c_{h'}(s_{h'}, a_{h'}) \mid s_h = s\right]$, where $\mathbb{E}^{\pi, p'}[\cdot]$ denotes the expectation with respect to policy $\pi$ and transition function $p'$, that is, $a_{h'} \sim \pi_{h'}(\cdot \mid s_{h'})$ and $s_{h'+1} \sim p'_{h'}(\cdot \mid s_{h'}, a_{h'})$.

**Learner-environment interaction.** At the beginning of episode $k$, the learner picks a policy $\pi^k$, and starts in an initial state $s_1^k = s_{\text{init}}$. In each time $h \in [H]$, it observes the current state $s_h^k$, draws an action from the policy $a_h^k \sim \pi_h^k(\cdot|s_h^k)$ and transitions to the next state $s_{h+1}^k \sim p_h(\cdot|s_h^k, a_h^k)$. The feedback of episode $k$ contains the cost function over the agent's trajectory $\{c_h^k(s_h^k, a_h^k)\}_{h=1}^H$, i.e., bandit feedback (as opposed to full-information feedback which contains the whole cost function). This feedback is observed only at the end of episode $k + d^k$, where the *delays* $\{d^k\}_{k=1}^K$ are unknown and chosen by the oblivious adversary together with the costs. If $d^k = 0$ for all $k$, this model scales down to standard online learning in adversarial MDP.

**Occupancy measure.** Given a policy $\pi$ and a transition function $p'$, the *occupancy measure* $q^{\pi, p'} \in [0, 1]^{HS^2A}$ is a vector, where $q_h^{\pi, p'}(s, a, s')$ is the probability to visit state $s$ at time $h$, take action $a$ and transition to state $s'$. We also denote $q_h^{\pi, p'}(s, a) = \sum_{s'} q_h^{\pi, p'}(s, a, s')$ and $q_h^{\pi, p'}(s) = \sum_a q_h^{\pi, p'}(s, a)$. By [36], the occupancy measure encodes the policy and the transition function through the relations $\pi_h(a \mid s) = q_h^{\pi, p'}(s, a)/q_h^{\pi, p'}(s)$; $p'_h(s' \mid s, a) = q_h^{\pi, p'}(s, a, s')/q_h^{\pi, p'}(s, a)$. The set of all occupancy measures with respect to an MDP $\mathcal{M}$ is denoted by $\Delta(\mathcal{M})$. Importantly, the value of a policy from the initial state can be written as the dot product between its occupancy measure and the cost function, i.e., $V_1^{\pi, p'}(s_{\text{init}}; c) = \langle q^{\pi, p'}, c \rangle$. Whenever $p'$ is omitted from the notations $q^{\pi, p'}$ and $V^{\pi, p'}$, this means that they are with respect to the true transition function $p$.

**Regret.** The learner's performance is measured by the *regret* which is the difference between the cumulative expected cost of the learner and the best fixed policy in hindsight:

$$R_K = \sum_{k=1}^K V_1^{k, \pi^k}(s_{\text{init}}) - \min_\pi \sum_{k=1}^K V_1^{k, \pi}(s_{\text{init}}) = \sum_{k=1}^K \langle q^{\pi^k}, c^k \rangle - \min_{q \in \Delta(\mathcal{M})} \sum_{k=1}^K \langle q, c^k \rangle,$$

where $V_h^{k, \pi}(s) = V_h^{\pi, p}(s; c^k)$.

**Confidence set.** Since the transition function is unknown, we maintain standard Bernstein-based confidence sets $\mathcal{P}^k$ for each episode $k$ that contain $p$ with high-probability. For the exact definition of $\mathcal{P}^k$ see Algorithms 5 and 9, and the fact that $p \in \mathcal{P}^k$ for every $k$ w.h.p is proved for example in [22] (for more details see the appendix). Using $\mathcal{P}^k$ we can define a confidence set of occupancy measures by

$$\Delta(\mathcal{M}, k) = \{q^{\pi, p'} \mid \pi \in (\Delta_{\mathcal{A}})^{\mathcal{S} \times [H]}, p' \in \mathcal{P}^k\},$$

which is a polytope with polynomial constraints as shown in [36]. Note that as long as $p \in \mathcal{P}^k$, $\Delta(\mathcal{M}) \subseteq \Delta(\mathcal{M}, k)$.

**Additional notations.** In general, episode indices always appear as superscripts and in-episode steps as subscripts. $\bar{p}_h^k(s'|s, a)$ is the empirical mean estimation of $p_h(s'|s, a)$ based on the trajectories available to the algorithm at the beginning of the episode $k$. $n_h^k(s, a, s')$ denotes the total number of visits at state $s$ in which the agent took action $a$ at time $h$ and transitioned to $s'$ by the end of episode $k - 1$, and $n_h^k(s, a) = \sum_{s'} n_h^k(s, a, s')$. Similarly, $m_h^k(s, a, s')$ denotes the total number of visits from rounds $j$ such that $j + d^j \leq k - 1$ at state $s$ in which the agent took action $a$ at time

---

**Algorithm 1** Delayed Hedge

---

1: **Initialization:** Set $\omega^1$ to be the uniform distribution over all deterministic policies, and $\mathcal{P}^1$ to be set of all transitions functions.
2: **for** $k = 1, 2, ..., K$ **do**
3:      Execute policy $\pi^k$ sampled from $\omega^k$, observe trajectory $\{s_h^k, a_h^k\}_{h=1}^H$.
4:      Update confidence set $\mathcal{P}^k$, compute upper occupancy bound $u^k$ and exploration bonus $b^k$ by:

$$u_h^k(s,a) = \max_{p' \in \mathcal{P}^k} \sum_{\pi \in \Omega} \omega^k(\pi) q_h^{\pi, p'}(s,a) \quad ; \quad b^k(\pi) = \max_{p' \in \mathcal{P}^k} \|q^{\pi, \bar{p}^k} - q^{\pi, p'}\|_1.$$

5:      **for** $j : j + d^j = k$ **do**
6:          Observe costs $\{c_h^j(s_h^j, a_h^j)\}_{h=1}^H$, compute loss estimator $\hat{c}^j$ defined in Eq. (1), and estimated loss by $\widehat{\ell^j}(\pi) = \langle q^{\pi, \bar{p}^j}, \widehat{c}^j \rangle$.
7:      **end for**
8:      Update policy distribution $\omega^{k+1}$ by: $\omega^{k+1}(\pi) \propto \omega^k(\pi) \cdot \exp\left(\eta b^k(\pi) - \eta \sum_{j:j+d^j=k} \widehat{\ell^j}(\pi)\right)$.
9: **end for**

---

$h$ and transitioned to $s'$, and $m_h^k(s,a) = \sum_{s'} m_h^k(s,a,s')$. $\mathcal{F}^k = \{j : j + d^j = k\}$ denotes the set of episodes such that their feedback arrives in the end of episode $k$. The notations $\widetilde{O}(\cdot)$ and $\lesssim$ hide constant and poly-logarithmic factors including $\log(K/\delta)$ for some confidence parameter $\delta$, the indicator of event $E$ is denoted by $\mathbb{I}\{E\}$, and $x \vee y = \max\{x, y\}$.

**Simplifying assumptions.** Throughout this paper we assume that $K$ and $D = \sum_{k=1}^K d^k$ are known and that the maximal delay $d_{max} = \max_k d^k \leq \sqrt{D}$. Both of these assumptions are made only for simplicity of presentation and can be easily relaxed using standard *doubling* and *skipping* procedures as shown for example by [41, 28, 4]. In addition, we focus on the case of non-delayed trajectory feedback, where the learner observes the trajectory immediately at the end of the episode and only the feedback regarding the cost is delayed. Delayed trajectory feedback mainly affects approximation errors and the ideas presented in [28] for handling such delay apply to our case as well. Finally, the regret bounds in the main text hide low-order terms that depends polynomially in $H, S$ and $A$ but only poly-logarithmically in $K$ - the full bounds appear in the appendix.

## 3 Delayed Hedge

In this section, we consider running a Hedge-based algorithm over all $\Omega = \mathcal{A}^{\mathcal{S} \times [H]}$ deterministic policies. Algorithm 1, which we call Delayed Hedge, is inefficient but gives the first order-optimal regret bounds for adversarial MDP with delayed bandit feedback. Although the main issue that Delayed Hedge tackles is delayed feedback, we note that there are many additional challenges introduced by the unknown transitions and the bandit feedback when we maintain a distribution over policies instead of a single stochastic policy.

Delayed Hedge maintains a distribution $\omega^k$ over deterministic policies (starting from a uniform distribution), and in the beginning of episode $k$ samples a policy $\pi^k$ to execute. Thus, the expected loss incurred in episode $k$ is $\sum_{\pi \in \Omega} \omega^k(\pi) \langle q^{\pi, p}, c^k \rangle$. The algorithm updates the distribution $\omega^k$ based on the exponential weights update, for which we need to compute an estimated loss for every policy $\pi \in \Omega$.

To do so, first we estimate the cost in each state-action pair. Due to unknown dynamics, following [22] we use the confidence sets to compute optimistic importance weighted estimator that will induce exploration:

$$\hat{c}_h^k(s,a) = \frac{c_h^k(s,a)\mathbb{I}\{s_h^k = s, a_h^k = a\}}{u_h^k(s,a) + \gamma}, \tag{1}$$

where $u_h^k(s,a) = \max_{p' \in \mathcal{P}^k} \sum_{\pi \in \Omega} \omega^k(\pi) q_h^{\pi, p'}(s,a)$ is an upper occupancy bound on the probability to visit $(s,a)$ in step $h$ of episode $k$, and $\gamma$ is a small bias added for high probability regret [33].

---

**Algorithm 2** Delayed UOB-FTRL

---

1: **Initialization:** Set $\pi^1$ to be uniform policy, and $\mathcal{P}^1$ to be set of all transitions functions..
2: **for** $k = 1, 2, ..., K$ **do**
3:     Execute policy $\pi^k$, observe trajectory $\{s_h^k, a_h^k\}_{h=1}^H$, update confidence set $\mathcal{P}^k$ and compute upper occupancy bound $u_h^k(s,a) = \max_{p' \in \mathcal{P}^k} q_h^{\pi^k, p'}(s,a)$.
4:     **for** $j : j + d^j = k$ **do**
5:         Observe costs $\{c_h^j(s_h^j, a_h^j)\}_{h=1}^H$ and compute the standard loss estimator $\hat{c}^j$ by

$$\hat{c}_h^j(s,a) = \frac{c_h^j(s,a)\mathbb{I}\{s_h^j = s, a_h^j = a\}}{u_h^j(s,a) + \gamma}. \tag{2}$$

6:     **end for**
7:     Compute occupancy measure by: $q^{k+1} = \arg\min_{q \in \cap_{j=1}^{k+1} \Delta(\mathcal{M}, j)} \langle q, \sum_{j+d^j \leq k} \hat{c}^j \rangle + \phi(q)$, where $\phi(q) = \frac{1}{\eta} \sum_{h,s,a,s'} q_h(s,a,s') \log q_h(s,a,s')$.
8:     Update policy: $\pi_h^{k+1}(a \mid s) = q_h^{k+1}(s,a)/q_h^{k+1}(s)$.
9: **end for**

---

Then, we use the empirical transition function $\bar{p}^k$ to compute the estimated loss $\widehat{\ell}^k(\pi) = \langle q^{\pi,\bar{p}^k}, \widehat{c}^k \rangle$ for each policy $\pi$. To ensure optimism, we introduce the exploration bonus $b^k(\pi) = \max_{p' \in \mathcal{P}^k} \|q^{\pi,\bar{p}^k} - q^{\pi,p'}\|_1$. As long as the real transition function $p$ is in confidence set $\mathcal{P}^k$, optimism is indeed ensured in the sense that $\langle q^{\pi,\bar{p}^k}, c \rangle - b^k(\pi)$ is always no more than the true cost $\langle q^{\pi,p}, c \rangle$ for any policy $\pi$ and $[0,1]$-valued cost function $c$.

With the estimated loss and the exploration bonus for each $\pi$, the distribution $\omega^{k+1}$ is now updated in a manner similar to that of [15]: $\omega^{k+1}(\pi) \propto \omega^k(\pi) \cdot \exp\left(\eta b^k(\pi) - \eta \sum_{j:j+d^j=k} \widehat{\ell^j}(\pi)\right)$. Note that all information required for this update has been received by the learner at the end of episode $k$. With the help of all these definitions, we prove the following regret bound for Delayed Hedge, and defer the details to Appendix A including the complete algorithm and regret analysis.

**Theorem 3.1.** *With appropriate choices of parameters, Delayed Hedge ensures* $R_K = \widetilde{O}\left(H^2 S\sqrt{AK} + H^{3/2}\sqrt{SD}\right)$ *with high probability (w.h.p.).*

## 4 Delayed UOB-FTRL

In this section, we adjust the UOB-REPS algorithm [22] to delayed feedback and present the Delayed UOB-FTRL algorithm (Algorithm 2) - the first efficient algorithm to attain order-optimal regret for adversarial MDP with delayed bandit feedback. The proof is based on a novel analysis without additional changes to the algorithm. Namely, we use standard loss estimators (defined in Eq. (2)). Our algorithm is based on the Follow-the-Regularized-Leader (FTRL) framework, which is widely used for deriving online learning algorithm in adversarial environments. Notable examples are [51] that applies FTRL over occupancy measure space to solve the adversarial MDP problem with known transition, and [52] that uses FTRL to achieve optimal regret for MAB with delayed feedback.

In our context, in the beginning of episode $k$, FTRL computes,

$$q^k = \underset{q \in \cap_{j=1}^k \Delta(\mathcal{M}, j)}{\arg\min} \langle q, \widehat{L}_k^{obs} \rangle + \phi(q), \tag{3}$$

where $\widehat{L}_k^{obs} = \sum_{j+d^j < k} \widehat{c}^j$ is the cumulative losses observed prior to episode $k$, and $\phi(q) = \frac{1}{\eta} \sum_{h,s,a,s'} q_h(s,a,s') \log q_h(s,a,s')$ is the Shannon entropy regularizer. Note that Eq. (3) is a convex optimization problem with linear constraints and thus can be solved efficiently [51, 36]. The policy $\pi^k$ to be played in the episode is then extracted from $q^k$. Thus, our algorithm can be regarded as a direct extension to MDP of FTRL for delayed feedback. However, unlike the successes in MAB, it is highly unclear whether optimal regret could be obtained in adversarial MDPs with FTRL even if the transition function is known.

In Theorem 4.1, we show that Delayed UOB-FTRL enjoys order-optimal regret. Through the key steps of the analysis, we shall take a closer look at the key reason why traditional analysis fails: in occupancy measure space, the interplay between different entries of loss functions is significantly harder to analyze. Thus, many critical properties used in [52] do not hold anymore. The complete algorithm and proof are deferred to Appendix B.

**Theorem 4.1.** *With appropriate choices of parameters, Delayed UOB-FTRL (Algorithm 2) ensures*
$$\mathbb{E}[R_K] = \widetilde{O}\big(H^2 S\sqrt{AK} + HSA\sqrt{HD}\big).$$

*Proof sketch of Theorem 4.1.* Let $q^\star = q^{\pi^\star, p}$ be the occupancy measure associated with the optimal policy $\pi^\star$. We adopt the regret decomposition of [22]:

$$R_K = \underbrace{\sum_{k=1}^{K}\left\langle q^{\pi^k} - q^k, c^k \right\rangle}_{\text{EST}} + \underbrace{\sum_{k=1}^{K}\left\langle q^k, c^k - \widehat{c}^k \right\rangle}_{\text{BIAS}_1} + \underbrace{\sum_{k=1}^{K}\left\langle q^k - q^\star, \widehat{c}^k \right\rangle}_{\text{REG}} + \underbrace{\sum_{k=1}^{K}\left\langle q^\star, \widehat{c}^k - c^k \right\rangle}_{\text{BIAS}_2}.$$

EST, BIAS$_1$ and BIAS$_2$ are standard and bounded in [22] w.h.p by $\widetilde{O}(\gamma HSAK + H^2 S\sqrt{AK} + H/\gamma)$.

Now, we focus on bounding REG. To this end, we denote by $\widehat{L}_k = \sum_{j=1}^{k-1} \widehat{c}^k$ the non-delayed cumulative loss, and introduce the convex conjugate functions $F_k^\star$ with respect to the regularizer $\phi(\cdot)$:

$$F_k^\star(x) = -\min_{q\in\Delta(\mathcal{M},k)}\left\{\phi(q) - \langle x, q\rangle\right\}.$$

We now use $F_k^\star$ to decompose REG into the following three terms as

$$\sum_{k=1}^{K} -F_k^\star(-\widehat{L}_k^{\text{obs}}) + \left\langle q^k, \widehat{c}^k \right\rangle + F_k^\star(-\widehat{L}_k^{\text{obs}} - \widehat{c}^k) + \sum_{k=1}^{K} -F_k^\star(-\widehat{L}_k - \widehat{c}^k) + F_k^\star(-\widehat{L}_k) - \left\langle q^\star, \widehat{c}^k \right\rangle$$

$$+ \sum_{k=1}^{K}\left\{-F_k^\star(-\widehat{L}_k^{\text{obs}} - \widehat{c}^k) + F_k^\star(-\widehat{L}_k^{\text{obs}}) - \left(-F_k^\star(-\widehat{L}_k - \widehat{c}^k) + F_k^\star(-\widehat{L}_k)\right)\right\}. \tag{4}$$

The first term is associated with the unseen loss $\widehat{c}^k$. It is relatively standard and bounded by $\widetilde{O}(\eta HSAK)$ w.h.p. The second term can be regarded as the regret of a "cheating" algorithm which does not suffer delay and sees one step into the future. This term can be bounded by $\widetilde{O}(H/\eta)$ similarly to [15]. The third term which only relates to delayed feedback, is the most critical object in the analysis.

In the previous work of [52] for multi-arm bandit, the authors managed to rewrite and then upper bound the delay-caused term for every episode $k$ by

$$\int_0^1 \left\langle \widehat{c}^k, \nabla F_k^\star(-\widehat{L}_k^{\text{obs}} - x\widehat{c}^k) - \nabla F_k^\star(-\widehat{L}_k - x\widehat{c}^k)\right\rangle dx \leq \eta \sum_{i\in[N]} p^k(i) \cdot \widehat{c}^k(i) \cdot \left(\widehat{L}_k(i) - \widehat{L}_k^{\text{obs}}(i)\right),$$

where $[N]$ is the set of arms and $p^k(i)$ is the probability that the algorithm chooses arm $i$ in episode $k$. Here, the first step uses Newton-Leibniz theorem and the differentiability of convex conjugates, and the second step follows directly from [52, Lemma 3]. Importantly, the second step is largely based on the specific structure of the simplex (over which MAB algorithms operate), which yields the simple behavior of FTRL-based algorithms (e.g., EXP3). Specifically, it is based on the following observation. Suppose that we increase the cumulative loss of arm $i$. Now consider the behavior of $p(i')$, the probability of taking arm $i'$ where $p$ is computed from the FTRL framework. One can verify that $p(i')$ will increase for $i' \neq i$ and decrease for $i' = i$. In other words, the relationship between any pair of arms is competitive, and this property is critical to achieve the optimal regret with delayed feedback in [52].

However, this property does not hold for MDPs because the constraints of the transition function can dictate positive correlation between entries of the occupancy measure. Similarly, consider two state-action pairs $(s, a, h)$ and $(s', a', h')$ from different states. It is highly unclear whether increasing the cumulative loss of $(s, a, h)$ will increase or decrease the probability $q_{h'}(s', a')$ of reaching $s'$ in time $h'$ and taking action $a'$. In fact, the relation is related to the specific transition function of

the MDP. For example, the FTRL algorithm will decrease the probability in the cases where taking action $a$ at state $s$ in step $h$ is necessary to reach $(s', a', h')$, and will increase in other cases where not taking action $a$ at state $s$ of step $h$ is necessary.

Therefore, an alternative analysis is required in our case. Specifically, we are able to bound the delayed-caused term by

$$\int_0^1 \left\langle \widehat{c}^k, \nabla F_k^\star(-\widehat{L}_k^{\text{obs}} - x\widehat{c}^k) - \nabla F_k^\star(-\widehat{L}_k - x\widehat{c}^k) \right\rangle dx \leq 2 \left\| \widehat{c}^k \right\|_{\nabla^{-2}\phi(\xi)} \left\| \widehat{L}_k - \widehat{L}_k^{\text{obs}} \right\|_{\nabla^{-2}\phi(\xi)}$$

$$\leq 2\eta \sum_{j=1, j+d^j \geq k}^{k-1} \left( \sum_{h,s,a} \widehat{c}_h^k(s,a) \right) \cdot \left( \sum_{h,s,a} \widehat{c}_h^j(s,a) \right)$$

where the first step uses the properties of convex conjugates for some valid occupancy measure $\xi$ (See Lemma B.6 for more details) with $\|x\|_M = \sqrt{x^\top M x}$ being the matrix norm for any vector $x$ and positive definite matrix $M$, and the second step follows from the facts that $\nabla^{-2}\phi(\xi)$ is a diagonal matrix with values $\{\eta \cdot \xi_h(s,a) : \forall (h, s, a)\}$ on its diagonal and $\xi_h(s,a) \leq 1$.

While we managed to overcome the complex dependencies between different states in the MDP, it comes at the price of a looser regret bound. The final bound does not have $q_h^k(s,a)$ in the summations which leads to an extra factor of $SA$. This follows from the application of Hölder's inequality and also the relaxation of intermediate occupancy measure $\xi$.

Taking the summation over all episodes, we have that the third term in Eq. (4) is bounded by $\widetilde{O}(\eta H^2 S^2 A^2 D)$ in expectation. Finally, with proper choice of the parameters $\eta$, $\gamma$ and $\delta$, combining the bounds for EST, BIAS$_1$, BIAS$_2$ and the three terms in Eq. (4) finishes the proof. □

## 5  Delayed UOB-REPS with Delay-adapted Estimator

Finally, we present our last algorithm, Delayed UOB-REPS equipped with our novel importance sampling estimator which we call *delay-adapted importance sampling estimator*. The algorithm appears as Algorithm 3 and in its full version together with the analysis for known and unknown dynamics in Appendices C and D.

Much like Delayed UOB-FTRL, the algorithm is efficient; but it outperforms Delayed UOB-FTRL in two important aspects: (i) it guarantees high-probability regret bound (and not only expected regret), and (ii) the delay term in its regret bound is tighter. In fact, as long as $A \leq S$ (which happens in most cases), it obtains an improvement even on the regret of the inefficient Delayed Hedge algorithm.

To maintain the occupancy measures $q^k$ from which the executed policies $\pi^k$ are extracted, Delayed UOB-REPS uses the Online Mirror Decent (OMD) update rule:

$$q^{k+1} = \underset{q \in \Delta(\mathcal{M}, k+1)}{\arg\min} \eta \left\langle q, \sum_{j: j+d^j = k} \hat{c}^j \right\rangle + \text{KL}(q \parallel q^k),$$

where $\eta$ is a learning rate and $\text{KL}(q \parallel q')$ is the unnormalized KL-divergence (see the full algorithm in Appendix D for the definition of KL-divergence). We note that OMD is standard in the O-REPS literature, and has similar guarantees to FTRL. In this case, OMD will be much more useful than FTRL because we can utilize its update rule to prove certain properties for the relation between consecutive occupancy measures (see Lemma D.8).

We do not use the standard importance sampling estimator, but the following delay-adapted estimator:

$$\hat{c}_h^k(s,a) = \frac{c_h^k(s,a)\mathbb{I}\{s_h^k = s, a_h^k = a\}}{\max\{u_h^k(s,a), u_h^{k+d^k}(s,a)\} + \gamma}. \tag{5}$$

The delay-adapted estimator specifically tackles one of the main technical challenges in analyzing algorithms under delayed feedback (especially in MDPs) – bound their stability. It is a biased estimator, and in fact has larger bias than the standard importance sampling estimator, but allows us to directly control the stability of the algorithm.

To describe the intuition behind the delay-adapted estimator, let us first consider a fixed delay $d^k = d$. The policy $\pi^{k+d}$ is updated based on the episodes $1, ..., k-1$. Thus, playing $\pi^{k+d}$ at episode $k$ is

---
**Algorithm 3** Delayed UOB-REPS with Delay-adapted Estimator
---
1: **Initialization:** Set $\pi^1$ to be uniform policy.
2: **for** $k = 1, 2, ..., K$ **do**
3:     Execute policy $\pi^k$, observe trajectory $\{s_h^k, a_h^k\}_{h=1}^H$, update confidence set $\mathcal{P}^k$ and compute
    upper occupancy bound $u_h^k(s,a) = \max_{p' \in \mathcal{P}^k} q_h^{\pi^k, p'}(s,a)$.
4:     **for** $j : j + d^j = k$ **do**
5:         Observe costs $\{c_h^j(s_h^j, a_h^j)\}_{h=1}^H$ and compute the delay-adapted cost estimator $\hat{c}^j$ by Eq. (5).
6:     **end for**
7:     Update occupancy measure by: $q^{k+1} = \arg\min_{q \in \Delta(\mathcal{M}, k+1)} \eta \left\langle q, \sum_{j \in \mathcal{F}^k} \hat{c}^j \right\rangle + \mathrm{KL}(q \parallel q^k)$.
8:     Update policy: $\pi_h^{k+1}(a \mid s) = q_h^{k+1}(s,a)/q_h^{k+1}(s)$.
9: **end for**
---

equivalent to running OMD on the same loss estimators but in a non-delayed environment. Standard analysis for delayed feedback (e.g., [41, 3] for MAB or [28] for MDPs) utilizes this fact to bound the regret with respect to the estimated cost by the sum of: (i) the regret of playing $\pi^{k+d}$; (ii) the "drift" between the playing $\pi^{k+d}$ and $\pi^k$:

$$\sum_{k=1}^K \langle q^k - q^\star, \hat{c}^k \rangle \lesssim \underbrace{\sum_{k=1}^K \langle q^k - q^{k+d}, \hat{c}^k \rangle}_{\text{DRIFT}} + \frac{H}{\eta} + \eta \underbrace{\sum_{h,s,a,k} q_h^{k+d}(s,a) \hat{c}_h^k(s,a)^2}_{\text{STABILITY}}. \qquad (6)$$

The term $\frac{H}{\eta}$ is usually referred to as the PENALTY, and the bound (i) $\leq$ PENALTY + STABILITY is by standard OMD guarantees. The *standard* importance sampling estimator defined in Eq. (2) is approximately unbiased (ignoring $\gamma$ and transition approximation errors), so the left-hand-side of Eq. (6) is approximately the regret in expectation. On the other hand, to bound the STABILITY term, one needs to control the ratio $q_h^{k+d}(s,a)/q_h^k(s,a)$ since $\hat{c}_h^k(s,a)$ has $q_h^k(s,a)$ in the denominator and not $q_h^{k+d}(s,a)$ (for simplicity we ignore the bias between $q^k$ and $u^k$).

In MAB, this ratio is essentially bounded by a constant, but the proof heavily relies on the simple update form of OMD on the simplex (i.e., EXP3), as explained in Section 4. However, it still remains unclear whether this ratio is bounded by a constant when running OMD or FTRL on a more general convex set such as $\Delta(\mathcal{M})$. While in the proof of Theorems 3.1 and 4.1 we are able to avoid bounding the ratio in the stability term itself by using a "cheating" regret approach, a similar issue re-appears in the drift term. In Theorem 3.1 we bound the ratio between distributions by utilizing the simple update form (for the specific argument see Eq. (21) in Appendix A), and in Theorem 4.1 we solve this issue with the help of convex conjugates (specifically, Hölder's inequality with respect to the Hessian of the regularizer at an intermediate occupancy measure $\xi$), but this comes at the cost of expected regret guarantees and looser bound on the delay term of the regret.

The main idea of the delay-adapted estimator is to re-weight the cost of episode $k$ using both $q^{k+d}$ and $q^k$. The first allows us to control the stability and avoids the need to bound the ratio $q_h^{k+d}(s,a)/q_h^k(s,a)$, while the second keeps the bias sufficiently small. More precisely, we re-weight using their maximum, which remarkably, causes the estimator's bias to scale similarly to the DRIFT term.

Finally, there are a few important points to notice with respect to our new estimator before we analyze the regret of Algorithm 3 in Theorem 5.1. First, since the estimator $\hat{c}^k$ is computed only in the end of episode $k + d^k$ (when the feedback from episode $k$ arrives), we have already computed both $u^k$ and $u^{k+d^k}$ at that point and the estimator is well-defined. Second, it generalizes the standard importance sampling estimator and adapts it to the delays. That is, whenever there is no delay, our estimator is identical to the standard importance sampling estimator. Third, there is no additional computational cost in computing the new estimator since we compute $u^k$ for every $k$ anyway. Moreover, there is no additional space complexity because every algorithm for adversarial environments with delayed feedback keeps the probabilities to play actions in episode $k$ until its feedback is received in the end of episode $k + d^k$.

**Theorem 5.1.** *With appropriate choices of parameters, Delayed UOB-REPS with the delay-adapted estimator (Algorithm 3) ensures with high probability that $R_K = \widetilde{O}\big(H^2 S \sqrt{AK} + (HSA)^{1/4} \cdot H\sqrt{D}\big)$.*

The second term in the regret improves the guarantee of Delayed UOB-FTRL with the standard estimator by a factor of $H^{1/4}(SA)^{3/4}$. It also improves Delayed Hedge by $(HS)^{1/4}$, but on the other hand has an extra factor $A^{1/4}$. Generally, this term is tight up to the $(HSA)^{1/4}$ factor [28]. The first term in the regret matches the state-of-the-art regret bound for non-delayed adversarial MDPs [22]. In Appendix C we consider the case of known transitions, and present Delayed O-REPS with the delay-adapted estimator that achieves the following regret bound. It has similar delay term but its first term is optimal up to poly-log factors [51].

**Theorem 5.2.** *Assume that the transition function is known to the learner. With high probability, Delayed O-REPS with the delay-adapted estimator (Algorithm 7) ensures that $R_K = \widetilde{O}\big(H\sqrt{SAK} + (HSA)^{1/4} \cdot H\sqrt{D}\big)$.*

We conclude the section with a proof sketch of our main theorem (for the unknown transition case).

*Proof sketch of Theorem 5.1.* We first break the regret as follows:

$$R_K = \underbrace{\sum_{k=1}^{K}\langle q^{\pi^k} - q^k, c^k\rangle}_{\text{EST}} + \underbrace{\sum_{k=1}^{K}\langle q^k, c^k - \hat{c}^k\rangle}_{\text{BIAS}_1} + \underbrace{\sum_{k=1}^{K}\langle q^\star, \hat{c}^k - c^k\rangle}_{\text{BIAS}_2}$$

$$+ \underbrace{\sum_{k=1}^{K}\langle q^k - q^{k+d^k}, \hat{c}^k\rangle}_{\text{DRIFT}} + \underbrace{\sum_{k=1}^{K}\langle q^{k+d^k} - q^\star, \hat{c}^k\rangle}_{\text{REG}}.$$

EST is the standard transition approximation error term which is bounded w.h.p by $\widetilde{O}(H^2 S\sqrt{AK})$ [22]. For BIAS$_2$ we use the fact that the delay-adapted estimator is always smaller than the standard estimator and bound it by $\widetilde{O}(H/\gamma)$ similarly to [22].

The real advantage of the estimator appears in the REG term. Similar to the fixed delay case, we can bound REG by,

$$\frac{H}{\eta} + \underbrace{\eta \sum_{k,h,s,a} q_h^{k+d^k}(s,a)\hat{c}_h^k(s,a)\left(\sum_{j\in\mathcal{F}^{k+d^k}}\hat{c}_h^j(s,a)\right)}_{\text{STABILITY}} \leq \frac{H}{\eta} + \eta \sum_{k,h,s,a}\sum_{j\in\mathcal{F}^{k+d^k}}\hat{c}_h^j(s,a),$$

where the inequality above is exactly where we utilize the delay-adapted estimator, as by its definition $\hat{c}_h^k(s,a) \leq 1/u_h^{k+d^k}(s,a) \leq 1/q_h^{k+d^k}(s,a)$, where the last inequality holds w.h.p. Then, using a standard concentration of $\hat{c}_h^k(s,a)$ around $c_h^k(s,a) \leq 1$ we get that STABILITY $\lesssim \eta(HSAK + d_{max}/\gamma)$. Importantly, the concentration arguments hold only because the maximum of $u^k$ and $u^{k+d^k}$ appears in the estimator's denominator. If it were only $u^{k+d^k}$, we could not have bounded the distance between the estimator $\hat{c}^k$ and the real cost $c^k$.

For the DRIFT term, let $\widetilde{H}^k$ be the realization of all episodes $j$ such that $j + d^j < k$. Note that $u^k$ and $u^{k+d^k}$ are completely determined by the history $\widetilde{H}^{k+d^k}$, and on the other hand, the $k$-th episode is not part of this history. Next, we take the absolute value on each element of $q^k - q^{k+d^k}$ and apply a concentration bound to obtain: DRIFT $\lesssim \sum_{k=1}^{K}\mathbb{E}\big[\langle|q^k - q^{k+d^k}|, \hat{c}^k\rangle \mid \widetilde{H}^{k+d^k}\big] + \frac{H}{\gamma}$.

The specific definition of the history $\widetilde{H}^{k+d^k}$ is crucial because now we have:

$$\text{DRIFT} \lesssim \sum_{k=1}^{K}\mathbb{E}\left[\langle|q^k - q^{k+d^k}|, \hat{c}^k\rangle \mid \widetilde{H}^{k+d^k}\right] + \frac{H}{\gamma} = \sum_{k=1}^{K}\langle|q^k - q^{k+d^k}|, \mathbb{E}\left[\hat{c}^k \mid \widetilde{H}^{k+d^k}\right]\rangle + \frac{H}{\gamma}$$

$$\leq \sum_{k=1}^{K}\|q^k - q^{k+d^k}\|_1 + \frac{H}{\gamma} \leq \sum_{k=1}^{K}\sum_{j=1}^{d^k}\|q^j - q^{j+1}\|_1 + \frac{H}{\gamma} \lesssim \sum_{k=1}^{K}\sum_{j=1}^{d^k}\sqrt{\text{KL}(q^j \parallel q^{j+1})} + \frac{H}{\gamma},$$

where the third step follows since w.h.p $\mathbb{E}\big[\hat{c}_h^k(s,a) \mid \widetilde{H}^{k+d^k}\big] = \frac{q_h^{\pi^k}(s,a)c_h^k(s,a)}{\max\{u_h^k(s,a),u_h^{k+d^k}(s,a)\}} \le 1$, the fourth step uses the triangle inequality, and the last is by Pinsker inequality. Finally, we utilize the OMD update (which uses KL as regularization) to obtain a bound on $\text{KL}\big(q^j \parallel q^{j+1}\big)$ and finally a bound $\widetilde{O}(\eta\sqrt{H^3SA}(D+K)+H/\gamma)$ on the DRIFT term. For BIAS$_1$, we apply a similar concentration on the cost estimators around $\mathbb{E}\big[\hat{c}^k \mid \widetilde{H}^{k+d^k}\big]$ and show that BIAS$_1$ is mainly bounded by,

$$\sum_k \|\max\{u^{k+d^k},u^k\} - q^k\|_1 + \gamma HSAK \le 2\sum_k \|u^k - q^k\|_1 + \sum_k \|q^{k+d^k} - q^k\|_1 + \gamma HSAK,$$

where the maximum is taken element-wise. For last, the first sum is bounded similarly to the EST term while the second sum is bounded similarly to the DRIFT term. Summing the regret from the different terms and optimizing over $\eta$ and $\gamma$ completes the proof. $\qquad\square$

## 6    Conclusions and Future Work

In this paper we made a substantial contribution to the literature on delayed feedback in RL. We presented the first algorithms that achieve near-optimal regret bounds for the challenging setting of adversarial MDP with delayed bandit feedback. Our key algorithmic contribution is a novel delay-adapted importance sampling estimator, and we develop various new techniques to analyze delayed bandit feedback in adversarial MDPs.

We leave a few interesting questions open for future work. First, there is still a gap of $(HSA)^{1/4}$ in the delay term between our upper bounds and the lower bound of [28]. Second, it remains an open question whether our new estimator is necessary to obtain optimal regret in the presence of delays, or is it possible to achieve optimal regret with standard algorithms. Finally, our algorithms are based on the O-REPS framework but it remains an important open problem to achieve $\widetilde{O}\big(\sqrt{K+D}\big)$ regret with policy optimization (PO) methods that are widely used in practice, and were recently shown to achieve near-optimal regret in adversarial MDP with non-delayed bandit feedback [31].

## Acknowledgements

TL, YM and AR have received funding from the European Research Council (ERC) under the European Union's Horizon 2020 research and innovation program (grant agreement No. 882396), by the Israel Science Foundation (grant number 993/17), Tel Aviv University Center for AI and Data Science (TAD), and the Yandex Initiative for Machine Learning at Tel Aviv University. HL is supported by NSF Award IIS-1943607 and a Google Faculty Research Award.

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
