# Appendix

**Algorithm 4** Delayed Hedge

**Input:** State space $\mathcal{S}$, Action space $\mathcal{A}$, Horizon $H$, Number of episodes $K$, Learning rate $\eta > 0$, Exploration parameter $\gamma > 0$, Confidence parameter $\delta > 0$.

**Initialization:** Set $\omega^1(\pi) = \frac{1}{|\Omega|}$ for every deterministic policy $\pi \in \Omega$; set $n_h^1(s,a) = 0, n_h^1(s,a,s')$ for every $(s,a,s',h) \in \mathcal{S} \times \mathcal{A} \times \mathcal{S} \times [H]$ and $\mathcal{P}^1$ be the set of all transition functions.

**for** $k = 1, 2, ..., K$ **do**

    Play a randomly sampled policy from distribution $\omega^k$ and observe trajectory $\{(s_h^k, a_h^k)\}_{h=1}^H$.

    Compute upper occupancy bound $u_h^k(s,a) = \max_{p' \in \mathcal{P}^k} \sum_{\pi \in \Omega} \omega^k(\pi) q_h^{p',\pi}(s,a)$.

    Define confidence set $\mathcal{P}^{k+1}$ by Algorithm 5.

    **for** $j : j + d^j = k$ **do**

        Observe feedback $\{c_h^j(s_h^j, a_h^j)\}_{h=1}^H$.

        Compute loss estimator $\widehat{c}_h^j(s,a) = \frac{c_h^j(s,a)\mathbb{I}\{s_h^j=s, a_h^j=a\}}{u_h^j(s,a)+\gamma}$ for every $(s,a,h) \in \mathcal{S} \times \mathcal{A} \times [H]$.

    **end for**

    Update probability distribution over policy space:

$$\omega^{k+1}(\pi) \propto \omega^k(\pi) \cdot \exp\left(\eta \cdot b^k(\pi) - \eta \sum_{j:j+d^j=k} \widehat{\ell}^j(\pi)\right), \forall \pi \in \mathcal{A}^{\mathcal{S} \times [H]}$$

    where $\widehat{\ell}^j(\pi) = \sum_{h=1}^H \sum_{s,a} q_h^{\pi,\bar{p}^j}(s,a)\widehat{c}_h^j(s,a)$ denotes the loss suffered by policy $\pi$ with respect to the loss estimator $\widehat{c}^j$ and transition function $\bar{p}^j$, $b^k(\pi) = \max_{p' \in \mathcal{P}^k} \left\| q^{\pi,\bar{p}^k} - q^{\pi,p'} \right\|_1$ is the exploration bonus for policy $\pi$ at episode $k$.

**end for**

---

**Algorithm 5** Update confidence set

**Input:** trajectory $\{(s_h^k, a_h^k)\}_{h=1}^H$.

Update visit counters: $n_h^{k+1}(s_h^k, a_h^k) \leftarrow n_h^k(s_h^k, a_h^k) + 1, n_h^{k+1}(s_h^k, a_h^k, s_{h+1}^k) \leftarrow n_h^k(s_h^k, a_h^k, s_{h+1}^k) + 1$ for every $h \in [H]$.

Compute empirical transitions function $\bar{p}^{k+1}$: $\bar{p}_h^{k+1}(s' \mid s,a) = \frac{n_h^{k+1}(s,a,s')}{n_h^{k+1}(s,a) \vee 1} \quad \forall(s,a,s',h)$.

Define confidence sets $\mathcal{P}^{k+1}$ such that $p' \in \mathcal{P}^{k+1}$ if and only if, for every $(s,a,s',h)$, $p'$ ensures $\sum_{s'} p_h'(s'|s,a) = 1$ and:

$$\left| p_h'(s'|s,a) - \bar{p}_h^{k+1}(s'|s,a) \right| \leq \sqrt{\frac{16\bar{p}_h^{k+1}(s'|s,a)\log\frac{10HSAK}{\delta}}{n_h^{k+1}(s,a) \vee 1}} + \frac{10\log\frac{10HSAK}{\delta}}{n_h^{k+1}(s,a) \vee 1}.$$

---

# A  Delayed Hedge

In this section, we consider running Hedge over the policy space, that is, the set of all deterministic policies. We propose Algorithm 4 with unknown transition and bandit feedback, which ensures $\widetilde{\mathcal{O}}\left(\sqrt{K} + \sqrt{D}\right)$ regret as shown in Theorem A.1 (ignoring dependence on other parameters).

**Theorem A.1.** *With $\eta = \gamma = \sqrt{\frac{S\iota}{HD+HSAK}}$, Algorithm 4 ensures that*

$$R_K = O\left(H^2 S\sqrt{AK\iota} + H^{3/2}\sqrt{SD\iota} + H^3 S^3 A\iota^3 + H^2 d_{max}\iota\right).$$

*with probability at least $1 - 64\delta$ and the coefficient $\iota = \log\frac{HSAK}{\delta}$.*

## A.1 Proof of the Main Theorem

*Proof of Theorem A.1.* We first decompose the regret decomposition as:

$$R_K = \sum_{k=1}^{K} \left\langle \omega^k - \omega^\star, \ell^k \right\rangle = \underbrace{\sum_{k=1}^{K} \left\langle \omega^k, \ell^k - \widehat{\ell}^k + b^k \right\rangle}_{\text{EST}} + \underbrace{\sum_{k=1}^{K} \left\langle \omega^k - \omega^\star, \widehat{\ell}^k - b^k \right\rangle}_{\text{REG}} + \underbrace{\sum_{k=1}^{K} \left\langle \omega^\star, \widehat{\ell}^k - b^k - \ell^k \right\rangle}_{\text{BIAS}}. \quad (7)$$

By combining Lemmas A.2 to A.4, we arrive at the following bound of regret with learning rate $\eta$, exploration parameter $\gamma$ and confidence parameter $\delta$, with probability at least $1 - 64\delta$ that

$$R_K = \mathcal{O}\left( \frac{HS\ln(A)}{\eta} + \eta H^2 \left(D + H^2 SAK\right) + \gamma HSAK + \left(\frac{\eta}{\gamma} H^2 \left(d_{max} + 1\right) + \frac{H}{\gamma}\right)\iota\right) \quad (8)$$
$$+ \mathcal{O}\left( H^2 S\sqrt{AK\iota} + H^3 S^3 A \ln K \iota^2\right).$$

Setting the learning rate and exploration parameter $\eta = \gamma = \sqrt{\frac{S\ln(A)}{HD + HSAK}}$, one can verify that the regret $R_K$ is bounded by $\mathcal{O}\left( H^2 S\sqrt{AK\iota} + H^{3/2}\sqrt{SD\iota} + H^3 S^3 A \iota^3 + H^2 d_{max}\iota\right)$. $\qquad\square$

Throughout the rest of this section, we will bound the three terms separately in Lemmas A.2 to A.4.

## A.2 Bound on the Bias of the Cost Estimator (BIAS in Eq. (7))

**Lemma A.2** (BIAS). *With probability at least $1 - 7\delta$, Algorithm 4 ensures that $\text{BIAS} = \mathcal{O}\left(\frac{H\iota}{\gamma}\right)$.*

*Proof.* Similar to the analysis in [22], we have BIAS bounded by

$$\sum_{k=1}^{K} \left\langle \omega^\star, \widehat{\ell}^k - b^k - \ell^k \right\rangle = \sum_{k=1}^{K} \left\langle q^{\pi^\star, \bar{p}^k}, \widehat{c}^k \right\rangle - \sum_{k=1}^{K} b^k(\pi^\star) - \sum_{k=1}^{K} \left\langle q^{\pi^\star, p}, c^k \right\rangle$$
$$\leq \mathcal{O}\left( \frac{H}{\gamma} \log\left(\frac{HSA}{\delta}\right)\right) + \sum_{k=1}^{K} \left\langle q^{\pi^\star, \bar{p}^k} - q^{\pi^\star, p}, c^k \right\rangle - b^k(\pi^\star)$$
$$\leq \mathcal{O}\left( \frac{H}{\gamma} \log\left(\frac{HSA}{\delta}\right)\right) + \sum_{k=1}^{K} \left\| q^{\pi^\star, \bar{p}^k} - q^{\pi^\star, p} \right\|_1 - b^k(\pi^\star)$$
$$= \mathcal{O}\left( \frac{H}{\gamma} \log\left(\frac{HSA}{\delta}\right)\right), \quad (9)$$

where the second step applies Lemma A.8 with probability at least $1 - 6\delta$; the third step applies Hölder's inequality; the last step follows from the event $p \in \cap_k \mathcal{P}^k$ which holds with probability at least $1 - \delta$, and the definition of exploration bonus $b^k(\pi)$. $\qquad\square$

## A.3 Bound on the Transition Estimation Error (EST in Eq. (7))

**Lemma A.3** (EST). *With probability at least $1 - 8\delta$, Algorithm 4 ensures that*

$$\text{EST} = \mathcal{O}\left( \gamma HSAK + H^2 S\sqrt{AK \log \iota} + S^3 H^3 A \ln K\iota\right).$$

*Proof.* Observe that, $\sum_{k=1}^{K}\left\langle\omega^k,\ell^k-\widehat{\ell}^k+b^k\right\rangle$ can be upper bounded under the event that $p\in\cap_k\mathcal{P}^k$ by

$$\sum_{k=1}^{K}\sum_{\pi\in\Omega}\omega^k(\pi)\left(\left\langle q^{\pi,p},c^k\right\rangle-\left\langle q^{\pi,\bar{p}^k},\widehat{c}^k\right\rangle\right)+\sum_{k=1}^{K}\sum_{\pi\in\Omega}\omega^k(\pi)b^k(\pi)$$

$$=\sum_{k=1}^{K}\sum_{\pi\in\Omega}\omega^k(\pi)\left\langle q^{\pi,p}-q^{\pi,\bar{p}^k},c^k\right\rangle+\sum_{k=1}^{K}\sum_{\pi\in\Omega}\omega^k(\pi)\left\langle q^{\pi,\bar{p}^k},c^k-\widehat{c}^k\right\rangle+\sum_{k=1}^{K}\sum_{\pi\in\Omega}\omega^k(\pi)b^k(\pi)$$

$$\leq\sum_{k=1}^{K}\left\langle q^k,c^k-\widehat{c}^k\right\rangle+2\sum_{k=1}^{K}\sum_{\pi\in\Omega}\omega^k(\pi)b^k(\pi) \tag{10}$$

where $q^k=\sum_{\pi\in\Omega}\omega^k(\pi)q^{\pi,\bar{p}^k}$ is the estimated occupancy measure at episode $k$, and the second step follows from the definition of $b^k$ and Hölder's inequality.

Note that, $\left\langle q^k,\widehat{c}^k\right\rangle$ is bounded by $H$ because $\bar{p}^k\in\mathcal{P}^k$ and $u_h^k(s,a)\geq q_h^k(s,a)$ by its definition. Thus, with the help of Azuma's inequality, we have with probability at least $1-\delta$,

$$\sum_{k=1}^{K}\left\langle q^k,\mathbb{E}^k\left[\widehat{c}^k\right]-\widehat{c}^k\right\rangle\leq\mathcal{O}\left(H\sqrt{K\ln\left(\frac{1}{\delta}\right)}\right).$$

where $E^k[\cdot]=E[\cdot\mid\mathcal{H}^k]$ and $\mathcal{H}^k$ is the history of episodes $1,...,k-1$. We then focus on the term $\sum_{k=1}^{K}\left\langle q^k,c^k-\mathbb{E}^k\left[\widehat{c}^k\right]\right\rangle$ and rewrite it as

$$\sum_{k=1}^{K}\sum_{h,s,a}q_h^k(s,a)c_h^k(s,a)\left(1-\frac{\mathbb{E}^k\left[\mathbb{I}\{s_h^k=s,a_h^k=a\}\right]}{u_h^k(s,a)+\gamma}\right)$$

$$=\sum_{k=1}^{K}\sum_{h,s,a}q_h^k(s,a)c_h^k(s,a)\left(1-\frac{\widehat{q}_h^k(s,a)}{u_h^k(s,a)+\gamma}\right)$$

$$=\sum_{k=1}^{K}\sum_{h,s,a}\frac{q_h^k(s,a)}{u_h^k(s,a)+\gamma}\left(u_h^k(s,a)-\widehat{q}_h^k(s,a)+\gamma\right)c_h^k(s,a)$$

$$\leq\gamma HSAK+\sum_{k=1}^{K}\sum_{h,s,a}\left|u_h^k(s,a)-\widehat{q}_h^k(s,a)\right| \tag{11}$$

where $\widehat{q}^k=\sum_{\pi\in\Omega}\omega^k(\pi)q^{\pi,p}$ is the occupancy measure with the true transition $p$, and the last step comes from the fact that $u_h^k(s,a)\geq q_h^k(s,a)$ for all state-action pairs according to its definition.

Fixed the state-action pair $(s,a)$ and let $p'\in\mathcal{P}^k$ be the transition function that yields $u_h^k(s,a)$ for simplicity. Then, we have the following inequality under the event $p\in\bigcap_k\mathcal{P}^k$ that

$$u_h^k(s,a)-\widehat{q}_h^k(s,a)=\sum_{\pi\in\Omega}\omega^k(\pi)\left(q_h^{\pi,p'}(s,a)-q_h^{\pi,p}(s,a)\right)$$

$$=\sum_{\pi\in\Omega}\omega^k(\pi)\sum_{m=0}^{h-1}\sum_{x,y,z}q_m^{\pi,p}(x,y)\cdot(p_m(z|x,y)-p_m'(z|x,y))\cdot q_{h|m+1}^{\pi,p'}(s,a|z)$$

$$\Rightarrow\left|u_h^k(s,a)-\widehat{q}_h^k(s,a)\right|\leq\sum_{\pi\in\Omega}\omega^k(\pi)\sum_{m=0}^{h-1}\sum_{s,a,s'}q_m^{\pi,p}(x,y)\cdot\epsilon_m^k(z|x,y)\cdot q_{h|m+1}^{\pi,p'}(s,a|z)$$

where the second step follows from [24, Lemma D.3.1] with the conditional occupancy measure $q_{h|m+1}^{\pi,p'}(s,a|z)$ being the conditional probability of visiting state-action pair $(s,a)$ at step $h$ from state $z$ at state $m+1$ with policy $\pi$ and transition $p'$; the third step comes from taking the absolute value of both sides and the fact that

$$\epsilon_h^k(s'|s,a)\triangleq\mathcal{O}\left(\min\left\{1,\sqrt{\frac{p_h^k(s'|s,a)\iota}{n_h^k(s,a)\vee 1}}+\frac{\iota}{n_h^k(s,a)\vee 1}\right\}\right)\geq|p_h'(s'|s,a)-p_h(s'|s,a)| \tag{12}$$

for any transition tuple $(s, a, s')$ and step $h$ under the event $p \in \bigcap_k \mathcal{P}^k$ according to [24, Lemma D.3.3]. In addition, we have $q_{h|m+1}^{\pi,p'}(s,a|z) - q_{h|m+1}^{\pi,p}(s,a|z)$ bounded by

$$
\sum_{o=m+1}^{h-1} \sum_{u,v,w} q_{o|m+1}^{\pi,p}(u,v|z) \cdot \left( p_o^k(w|u,v) - p_o'(w|u,v) \right) \cdot q_{h|o+1}^{\pi,p'}(s,a|w)
$$

$$
\leq \pi_h(a|s) \sum_{o=m+1}^{h-1} \sum_{u,v,w} q_{o|m+1}^{\pi,p}(u,v|z) \cdot \left| p_o^k(w|u,v) - p_o'(w|u,v) \right|
$$

$$
\leq \pi_h(a|s) \sum_{o=m+1}^{h-1} \sum_{u,v} q_{o|m+1}^{\pi,p}(u,v|z) \cdot \min\left\{ 2, \sum_w \epsilon_o^k(w|u,v) \right\}
$$

where the first step uses the fact that $q_{h|o+1}^{\pi,p'}(s,a|w) \leq \pi_h(a|s) \cdot q_{h|o+1}^{\pi,p'}(s|w) = \pi_h(a|s)$; the second step follows from similar argument above; the last step uses the fact that $\sum_w \left| p_o^k(w|u,v) - p_o'(w|u,v) \right| \leq 2$.

Combining these inequalities, we have the second term of Equation (11), $\sum_{k=1}^k \sum_{h,s,a} \left| u_h^k(s,a) - \widehat{q}_h^k(s,a) \right|$, bounded by

$$
\sum_{k=1}^K \sum_{\pi \in \Omega} \omega^k(\pi) \sum_{h,s,a} \sum_{m=0}^{h-1} \sum_{x,y,z} q_m^{\pi,p}(x,y) \cdot \epsilon_m^k(z|x,y) \cdot q_{h|m+1}^{\pi,p}(s,a|z)
$$

$$
+ \sum_{k=1}^K \sum_{\pi \in \Omega} \omega^k(\pi) \sum_{h,s,a} \sum_{m=0}^{h-1} \sum_{x,y,z} \sum_{o=m+1}^{h-1} \sum_{u,v} q_m^{\pi,p}(x,y) \cdot \epsilon_m^k(z|x,y) \cdot q_{o|m+1}^{\pi,p}(u,v|z) \cdot \min\left\{ 2, \sum_w \epsilon_o^k(w|u,v) \right\} \cdot \pi_h(a|s).
$$

$$(13)$$

Note that, the first term of Eq. (13) can be bounded (under the event $p \in \bigcap_k \mathcal{P}^k$) as

$$
\sum_{k=1}^K \sum_{\pi \in \Omega} \omega^k(\pi) \sum_{h,s,a} \sum_{m=0}^{h-1} \sum_{x,y,z} q_m^{\pi,p}(x,y) \cdot \epsilon_m^k(z|x,y) \cdot q_{h|m+1}^{\pi,p}(s,a|z)
$$

$$
= \sum_{k=1}^K \sum_{\pi \in \Omega} \omega^k(\pi) \sum_{m=0}^H \sum_{x,y,z} q_m^{\pi,p}(x,y) \cdot \epsilon_m^k(z|x,y) \cdot \left( \sum_{h=m+1}^H \sum_{s,a} q_{h|m+1}^{\pi,p}(s,a|z) \right)
$$

$$
\leq H \sum_{k=1}^K s \sum_{\pi \in \Omega} \omega^k(\pi) \sum_{m=0}^H \sum_{x,y,z} q_m^{\pi,p}(x,y) \cdot \epsilon_m^k(z|x,y)
$$

$$
= H \sum_{k=1}^K \sum_{m=0}^H \sum_{x,y} \left( \sum_{\pi \in \Omega} \omega^k(\pi) q_m^{\pi,p}(x,y) \right) \cdot \left( \sum_z \epsilon_m^k(z|x,y) \right)
$$

$$
= H \sum_{k=1}^K \sum_{m=0}^H \sum_{x,y} \widehat{q}_m^k(x,y) \cdot \left( \sum_z \epsilon_m^k(z|x,y) \right)
$$

$$
= \mathcal{O}\left( H \sum_{k=1}^K \sum_{m=0}^H \sum_{x,y} \widehat{q}_m^{\pi,p}(x,y) \left( \sqrt{\frac{S\iota}{n_m^k(x,y) \vee 1}} + \frac{S\iota}{n_m^k(x,y) \vee 1} \right) \right)
$$

$$
= \mathcal{O}\left( H^2 S \sqrt{AK \log \iota} \right)
$$

$$(14)$$

where the second steps follows from the fact that $\sum_{s,a} q_{h|m+1}^{\pi,p}(s,a|z) = 1$ for any policy $\pi$ and step $h \geq m+1$; the fourth step uses the definition of $\widehat{q}^k$, the true occupancy measure at episode $k$; the fifth step uses the properties of $\epsilon^k$ under the event $p \in \bigcap_k \mathcal{P}^k$; the final step applies Lemma A.6, which yields a high probability bound with the help of a standard Bernstein-type concentration inequality for martingale.

Observing that $\sum_{h=o+1}^{H} \sum_{s,a} \pi_h(a|s) \le SH$, we can reorder the summation and bound the second term of Eq. (13) by $SH$ multiplying

$$\sum_{k=1}^{k} \sum_{\pi \in \Omega} \omega^k(\pi) \sum_{m=0}^{h-1} \sum_{x,y,z} \sum_{o=m+1}^{h-1} \sum_{u,v} q_m^{\pi,p}(x,y) \cdot \epsilon_m^k(z|x,y) \cdot q_{o|m+1}^{\pi,p}(u,v|z) \cdot \min\left\{2, \sum_w \epsilon_o^k(w|u,v)\right\}.$$

Similar to the proof in Appendix B.2 of [22], we can further rewrite and bound the term above by

$$\mathcal{O}\left(\sum_{k=1}^{K} \sum_{\pi \in \Omega} \omega^k(\pi) \sum_{m=0}^{H-1} \sum_{x,y,z} \sum_{o=m+1}^{H} \sum_{u,v,w} q_m^{\pi,p}(x,y) \cdot \sqrt{\frac{p_m(z|x,y)\iota}{n_m^k(x,y) \vee 1}} \cdot q_{o|m+1}^{\pi,p}(u,v|z) \cdot \sqrt{\frac{p_o(w|u,v)\iota}{n_o^k(u,v) \vee 1}}\right)$$

$$+ \mathcal{O}\left(\sum_{k=1}^{K} \sum_{\pi \in \Omega} \omega^k(\pi) \sum_{m=0}^{H-1} \sum_{x,y,z} \frac{q_m^{\pi,p}(x,y)\iota}{n_m^k(x,y) \vee 1} \left(\sum_{o=m+1}^{H} \sum_{u,v} q_{o|m+1}^{\pi,p}(u,v|z) \min\left\{\sum_w \epsilon_o^k(w|u,v), 2\right\}\right)\right)$$

$$+ \mathcal{O}\left(\sum_{k=1}^{K} \sum_{\pi \in \Omega} \omega^k(\pi) \sum_{o=0}^{H} \sum_{u,v,w} \left(\sum_{m=0}^{o-1} \sum_{x,y,z} q_m^{\pi,p}(x,y) \cdot p_m(z|x,y) \cdot q_{o|m+1}^{\pi,p}(u,v|z)\right) \cdot \frac{\iota}{n_o^k(u,v) \vee 1}\right)$$

by using the property of $\epsilon_h^k$ as in Eq. (12) and the fact that $\sqrt{xy} \le x + y$ for any $x, y > 0$, therefore, $\epsilon_h^k(s'|s,a) \le \mathcal{O}\left(p_h(s'|s,a) + \frac{\iota}{n_h^k(s,a) \vee 1}\right)$ holds for any $(s,a,s')$.

Clearly, the later two are able to be reformulated and then bounded as

$$\mathcal{O}\left(\sum_{k=1}^{K} \sum_{\pi \in \Omega} \omega^k(\pi) \sum_{m=0}^{H-1} \sum_{x,y,z} \frac{q_m^{\pi,p}(x,y)\iota}{n_m^k(x,y) \vee 1} \left(\sum_{o=m+1}^{H} \sum_{u,v} q_{o|m+1}^{\pi,p}(u,v|z) \min\left\{\sum_w \epsilon_o^k(w|u,v), 2\right\}\right)\right)$$

$$+ \mathcal{O}\left(\sum_{k=1}^{K} \sum_{\pi \in \Omega} \omega^k(\pi) \sum_{o=0}^{H} \sum_{u,v,w} \left(\sum_{m=0}^{o-1} \sum_{x,y,z} q_m^{\pi,p}(x,y) \cdot p_m(z|x,y) \cdot q_{o|m+1}^{\pi,p}(u,v|z)\right) \cdot \frac{\iota}{n_o^k(u,v) \vee 1}\right)$$

$$= \mathcal{O}\left(H \sum_{k=1}^{K} \sum_{\pi \in \Omega} \omega^k(\pi) \sum_{m=0}^{H-1} \sum_{x,y,z} \frac{q_m^{\pi,p}(x,y)\iota}{n_m^k(x,y) \vee 1} + H \sum_{k=1}^{K} \sum_{\pi \in \Omega} \omega^k(\pi) \sum_{o=0}^{H} \sum_{u,v,w} \frac{q_o^{\pi,p}(u,v)\iota}{n_o^k(u,v) \vee 1}\right)$$

$$= \mathcal{O}\left(SH\iota^2 \sum_{k=1}^{K} \sum_{m=0}^{H-1} \sum_{x,y} \frac{\widehat{q}_m^k(x,y)}{n_m^k(x,y) \vee 1} + SH\iota^2 \sum_{k=1}^{K} \sum_{o=0}^{H} \sum_{u,v} \frac{\widehat{q}_o^k(u,v)}{n_o^k(u,v) \vee 1}\right)$$

$$= \mathcal{O}\left(S^2 HA \ln K \iota^2\right) \tag{15}$$

where the first step comes from the facts that $\sum_{o=m+1}^{H} \sum_{u,v} q_{o|m+1}^{\pi,p}(u,v|z) \le H$ for any $z$, and $\sum_{x,y,z} q_m^{\pi,p}(x,y) \cdot p_m(z|x,y) \cdot q_{o|m+1}^{\pi,p}(u,v|z) = q^{\pi,p}(u,v)$ for any $(u,v)$ according to the definitions of conditional occupancy measures; the second step follows from the definition of $\widehat{q}^k$; the last step applies Lemma A.6 with probability at least $1 - 2\delta$.

On the other hand, the first term can be written as $SH\iota$ multiplied by the following (ignoring some constants):

$$\sum_{k=1}^{K}\sum_{\pi\in\Omega}\omega^k(\pi)\sum_{m=0}^{H-1}\sum_{x,y,z}\sum_{o=m+1}^{H}\sum_{u,v,w}q_m^{\pi,p}(x,y)\cdot\sqrt{\frac{p_m(z|x,y)}{n_m^k(x,y)\vee 1}}\cdot q_{o|m+1}^{\pi,p}(u,v|z)\cdot\sqrt{\frac{p_o(w|u,v)}{n_o^k(u,v)\vee 1}}$$

$$=\sum_{k=1}^{K}\sum_{\pi\in\Omega}\omega^k(\pi)\sum_{m=0}^{H-1}\sum_{x,y,z}\sum_{o=m+1}^{H}\sum_{u,v,w}\sqrt{\frac{q_m^{\pi,p}(x,y)p_m(z|x,y)q_{o|m+1}^{\pi,p}(u,v|z)}{n_m^k(x,y)\vee 1}}\cdot\sqrt{\frac{q_m^{\pi,p}(x,y)p_o(w|u,v)q_{o|m+1}^{\pi,p}(u,v|z)}{n_o^k(u,v)\vee 1}}$$

$$=\sum_{k=1}^{K}\sum_{\pi\in\Omega}\sum_{m=0}^{H-1}\sum_{x,y,z}\sum_{o=m+1}^{H}\sum_{u,v,w}\sqrt{\frac{\omega^k(\pi)q_m^{\pi,p}(x,y)p_m(z|x,y)q_{o|m+1}^{\pi,p}(u,v|z)}{n_o^k(u,v)\vee 1}}\cdot\sqrt{\frac{\omega^k(\pi)q_m^{\pi,p}(x,y)p_o(w|u,v)q_{o|m+1}^{\pi,p}(u,v|z)}{n_m^k(x,y)\vee 1}}$$

$$\leq\sum_{m=0}^{H-1}\sum_{o=m+1}^{H}\sqrt{\sum_{k=1}^{K}\sum_{\pi\in\Omega}\sum_{x,y,z}\sum_{u,v,w}\frac{\omega^k(\pi)q_m^{\pi,p}(x,y)p_m(z|x,y)q_{o|m+1}^{\pi,p}(u,v|z)}{n_o^k(u,v)\vee 1}}$$

$$\cdot\sqrt{\sum_{k=1}^{K}\sum_{\pi\in\Omega}\sum_{x,y,z}\sum_{u,v,w}\frac{\omega^k(\pi)q_m^{\pi,p}(x,y)p_o(w|u,v)q_{o|m+1}^{\pi,p}(u,v|z)}{n_m^k(x,y)\vee 1}}$$

$$\leq\sum_{m=0}^{H-1}\sum_{o=m+1}^{H}\sqrt{\sum_{k=1}^{K}\sum_{\pi\in\Omega}\sum_{u,v,w}\frac{\omega^k(\pi)q_o^{\pi,p}(u,v)}{n_o^k(u,v)\vee 1}}\cdot\sqrt{\sum_{k=1}^{K}\sum_{\pi\in\Omega}\sum_{x,y,z}\frac{\omega^k(\pi)q_m^{\pi,p}(x,y)}{n_m^k(x,y)\vee 1}}$$

$$=S\sum_{m=0}^{H-1}\sum_{o=m+1}^{H}\sqrt{\sum_{k=1}^{K}\sum_{u,v}\frac{\widehat{q}_o^k(u,v)}{n_o^k(u,v)\vee 1}}\cdot\sqrt{\sum_{k=1}^{K}\sum_{x,y,z}\frac{\widehat{q}_m(x,y)}{n_m^k(x,y)\vee 1}}$$

$$=\mathcal{O}\left(S^2H^2A\ln K\right)\tag{16}$$

where the third step uses Cauchy-Schwarz inequality; the fourth step follows from the properties of conditional occupancy measure $\sum_{x,y,z}q_m^{\pi,p}(x,y)p_m(z|x,y)q_{o|m+1}^{\pi,p}(u,v|z)=q_m^{\pi,p}(u,v)$; the last step applies Lemma A.6 with probability at least $1-2\delta$.

Combining Equations (13) to (16) into Eq. (11), we have the following inequality holds with probability at least $1-4\delta$ under the event $p\in\bigcap_k\mathcal{P}^k$ that

$$\sum_{k=1}^{K}\left\langle q^k,c^k-\widehat{c}^k\right\rangle=\mathcal{O}\left(\gamma HSAK+H^2S\sqrt{AK\log\iota}+S^3H^3A\ln K\iota\right).\tag{17}$$

With slightly abuse of notations, we use $\bar{p}^k(\pi)$ to denote the transition function that yields $b^k(\pi)$ associated with $\pi$ and confidence set $\mathcal{P}^k$, that is, $\bar{p}^k(\pi)=\arg\max_{p'\in\mathcal{P}^k}\left\|q^{\pi,p'}-q^{\pi,\bar{p}^k}\right\|_1$. Thus, for $\sum_{k=1}^{K}\left\langle\omega^k,b^k\right\rangle$, we have the following inequality holds with probability at least $1-2\delta$ that

$$\sum_{k=1}^{K}\left\langle\omega^k,b^k\right\rangle=\sum_{k=1}^{K}\sum_{\pi\in\Omega}\omega^k(\pi)\left\|q^{\pi,\bar{p}^k(\pi)}-q^{\pi,\bar{p}^k}\right\|_1$$

$$\leq\sum_{k=1}^{K}\sum_{\pi\in\Omega}\omega^k(\pi)\left(\left\|q^{\pi,\bar{p}^k(\pi)}-q^{\pi,p}\right\|_1+\left\|q^{\pi,p}-q^{\pi,\bar{p}^k}\right\|_1\right)$$

$$\leq H\sum_{k=1}^{K}\sum_{\pi\in\Omega}\omega^k(\pi)\sum_{h=1}^{H}q_h^{\pi,p}(s,a)\cdot\left(\left\|\bar{p}_h^k(\cdot|s,a)-p_h(\cdot|s,a)\right\|_1+\left\|\bar{p}_h^k(\pi)(\cdot|s,a)-p_h(\cdot|s,a)\right\|_1\right)$$

$$\leq H\sum_{k=1}^{K}\sum_{\pi\in\Omega}\omega^k(\pi)\sum_{h=1}^{H}q_h^{\pi,p}(s,a)\cdot\left(\sum_{s'}\epsilon_h^k(s'|s,a)\right)$$

$$\leq\mathcal{O}\left(H\sum_{k=1}^{K}\sum_{h=1}^{H}\widehat{q}_h^k(s,a)\sqrt{\frac{S\iota}{n_h^k(s,a)\vee 1}}\right)$$

$$\leq\mathcal{O}\left(H^2S\sqrt{AK\iota}\right)\tag{18}$$

where the second step follows from the triangle inequality for $\ell_1$ norms; the third step comes from Lemma B.1 and B.2 of [36]; the forth step uses the property of $\epsilon^k$ defined in Eq. (12); the fifth step follows from the fact that $\sum_{\pi \in \Omega} \omega^k(\pi) q^{\pi,p} = \widehat{q}^k$; the final step follows from the same argument as in Eq. (14).

Combining Equations (17) and (18) into Eq. (10) concludes the proof. $\qquad\square$

### A.4 Bound on the Regret with respect to the Loss Estimators (REG in Eq. (7))

**Lemma A.4** (REG). *With probability at least $1 - 32\delta$, Algorithm 4 ensures that*

$$\text{REG} = \mathcal{O}\left(\frac{HS\ln(A)}{\eta} + \eta H^2 \left(SAK + D\right) + \frac{\eta}{\gamma} \cdot H^2 \left(d_{max} + 1\right)\iota\right).$$

*Proof.* Let $\{\widetilde{\omega}^{k+1}\}_{k=1}^K$ be the sequence of probability distributions with both received and un-received loss estimators prior to episode $k + 1$, that is,

$$\widetilde{\omega}^{k+1}(\pi) \propto \omega^1(\pi) \cdot \exp\left(-\eta \left(\sum_{j=1}^{k} \widehat{\ell^j}(\pi) - \sum_{j=1}^{k} b^j(\pi)\right)\right), \forall \pi \in \mathcal{A}^{\mathcal{S} \times [H]}.$$

On the other hand, according to the fact that $b^j(\pi') \leq 2H$, we add a constant $2H$ uniformly to the loss vector $\widehat{\ell}^k - b^k$ and construct $m^k(\pi) = \widehat{\ell}^k(\pi) - b^k(\pi) + 2H$ to ensure the positiveness for any $\pi$. Clearly, adding the constant uniformly will not change the outcomes of our algorithm.

With the help of these notations, we are able to decompose REG into two parts as:

$$\text{REG} = \underbrace{\sum_{k=1}^K \left\langle \widetilde{\omega}^{k+1} - \omega^\star, \widehat{\ell}^k - b^k \right\rangle}_{\text{CHEATING REGRET}} + \underbrace{\sum_{k=1}^K \left\langle \omega^k - \widetilde{\omega}^{k+1}, \widehat{\ell}^k - b^k \right\rangle}_{\text{DRIFT}}$$

where CHEATING-REGRET is bounded in [15] that

$$\sum_{k=1}^K \left\langle \widetilde{\omega}^{k+1} - \omega^\star, m^k \right\rangle \leq \frac{\ln|\Omega|}{\eta} = \frac{\ln\left|\mathcal{A}^{\mathcal{S} \times [H]}\right|}{\eta} = \frac{HS\ln(A)}{\eta}. \tag{19}$$

For DRIFT, we first rewrite it as

$$\sum_{k=1}^K \left\langle \omega^k - \widetilde{\omega}^{k+1}, \widehat{\ell}^k - b^k \right\rangle = \sum_{k=1}^K \left\langle \omega^k - \widetilde{\omega}^{k+1}, 2H + \widehat{\ell}^k - b^k \right\rangle$$

$$= \sum_{k=1}^K \sum_{\pi \in \Omega} \omega^k(\pi) \left(2H + \widehat{\ell}^k(\pi) - b^k(\pi)\right) \cdot \left(1 - \frac{\widetilde{\omega}^{k+1}(\pi)}{\omega^k(\pi)}\right)$$

$$= \sum_{k=1}^K \sum_{\pi \in \Omega} \omega^k(\pi) m^k(\pi) \cdot \left(1 - \frac{\widetilde{\omega}^{k+1}(\pi)}{\omega^k(\pi)}\right) \tag{20}$$

where the second step follows from the fact that $\sum_{\pi \in \Omega} \widetilde{\omega}^{k+1}(\pi) = \sum_{\pi \in \Omega} \omega^k(\pi) = 1$. Then, we consider the ratio between $\omega^k(\pi)$ and $\widetilde{\omega}^{k+1}(\pi)$:

$$\frac{\widetilde{\omega}^{k+1}(\pi)}{\omega^k(\pi)} = \frac{\exp\left(-\eta \sum_{j=1}^{k}\left(\widehat{\ell^j}(\pi) - b^j(\pi)\right)\right)}{\sum_{\pi' \in \Omega} \exp\left(-\eta \sum_{j=1}^{k}\left(\widehat{\ell^j}(\pi') - b^j(\pi')\right)\right)} \cdot \frac{\sum_{\pi' \in \Omega} \exp\left(-\eta \sum_{j:j+d^j<k} \widehat{\ell^j}(\pi') + \eta \sum_{j=1}^{k-1} b^j(\pi')\right)}{\exp\left(-\eta \sum_{j:j+d^j<k} \widehat{\ell^j}(\pi) + \eta \sum_{j=1}^{k-1} b^j(\pi)\right)}$$

$$= \frac{\exp\left(-\eta \sum_{j=1}^{k}\left(\widehat{\ell^j}(\pi) - b^j(\pi)\right) - \eta 2H\right)}{\sum_{\pi' \in \Omega} \exp\left(-\eta \sum_{j=1}^{k}\left(\widehat{\ell^j}(\pi') - b^j(\pi')\right) - \eta 2H\right)} \cdot \frac{\sum_{\pi' \in \Omega} \exp\left(-\eta \sum_{j:j+d^j<k} \widehat{\ell^j}(\pi') + \eta \sum_{j=1}^{k-1} b^j(\pi')\right)}{\exp\left(-\eta \sum_{j:j+d^j<k} \widehat{\ell^j}(\pi) + \eta \sum_{j=1}^{k-1} b^j(\pi)\right)}$$

$$= \frac{\sum_{\pi' \in \Omega} \exp\left(-\eta \sum_{j:j+d^j<k} \widehat{\ell^j}(\pi') + \eta \sum_{j=1}^{k-1} b^j(\pi')\right)}{\sum_{\pi' \in \Omega} \exp\left(-\eta \sum_{j=1}^{k}\left(\widehat{\ell^j}(\pi') - b^j(\pi')\right) - \eta 2H\right)} \cdot \frac{\exp\left(-\eta \sum_{j=1}^{k}\left(\widehat{\ell^j}(\pi) - b^j(\pi)\right) - \eta 2H\right)}{\exp\left(-\eta \sum_{j:j+d^j<k} \widehat{\ell^j}(\pi) + \eta \sum_{j=1}^{k-1} b^j(\pi)\right)} \tag{21}$$

where the second step follows from multiplying denominator and nominator together by $\exp(-\eta 2H)$. Note that $b^k(\pi) \le 2H$ and $\widehat{\ell}^k(\pi) \ge 0$ for any $\pi$ and $k$, we thus have the following inequality holds that

$$\sum_{j=1}^{k} \left( \widehat{\ell^j}(\pi') - b^j(\pi') \right) + 2H = \sum_{j=1}^{k} \widehat{\ell^j}(\pi') - \sum_{j=1}^{k-1} b^j(\pi') + 2H - b^k(\pi') \ge \sum_{j=1, j+d^j < k}^{k-1} \widehat{\ell^j}(\pi') - \sum_{j=1}^{k-1} b^j(\pi')$$

which indicates that the first fraction is lower bounded by 1.

Therefore, the ratio $\widetilde{\omega}^{k+1}(\pi)/\omega^k(\pi)$ for any policy $\pi \in \Omega$ can be further bounded by

$$\widetilde{\omega}^{k+1}(\pi)/\omega^k(\pi) \ge \exp\left(-\eta \left( \widehat{\ell}^k(\pi) + \sum_{j=1, j+d^j \ge k}^{k-1} \widehat{\ell}^k(\pi) + \left(2H - b^k(\pi)\right) \right)\right)$$

$$\ge 1 - \eta \left( m^k(\pi) + \sum_{j=1, j+d^j \ge k}^{k-1} \widehat{\ell}^k(\pi) \right),$$

where the last step uses $1 + x \le e^x$ for any $x \in \mathbb{R}$.

Plugging this inequality back to Eq. (20), we have DRIFT bounded and then decomposed into two parts as

$$\sum_{k=1}^{K} \sum_{\pi \in \Omega} \omega^k(\pi) m^k(\pi) \left( 1 - \frac{\widetilde{\omega}^{k+1}(\pi)}{\omega^k(\pi)} \right) \le \eta \sum_{k=1}^{K} \sum_{\pi \in \Omega} \omega^k(\pi) m^k(\pi) \left( m^k(\pi) + \sum_{j=1, j+d^j \ge k}^{k-1} \widehat{\ell}^k(\pi) \right)$$

$$= \eta \sum_{k=1}^{K} \sum_{\pi \in \Omega} \omega^k(\pi) m^k(\pi)^2 + \eta \sum_{k=1}^{K} \sum_{\pi \in \Omega} \omega^k(\pi) m^k(\pi) \sum_{j=1, j+d^j \ge k}^{k-1} \widehat{\ell^j}(\pi) \quad (22)$$

where the first part associates with the regret incurred without the delayed feedback and can be controlled by standard arguments as:

$$\eta \sum_{k=1}^{K} \sum_{\pi \in \Omega} \omega^k(\pi) m^k(\pi)^2 = \eta \sum_{k=1}^{K} \sum_{\pi \in \Omega} \omega^k(\pi) \left( \sum_{h=1}^{H} \sum_{s,a} q_h^{\pi, \bar{p}^k}(s,a) \widehat{c}_h^k(s,a) + 2H - b^k(\pi) \right)^2$$

$$\le 2\eta \sum_{k=1}^{K} \sum_{\pi \in \Omega} \omega^k(\pi) \left[ \left( \sum_{h=1}^{H} \sum_{s,a} q_h^{\pi, \bar{p}^k}(s,a) \widehat{c}_h^k(s,a) \right)^2 + 4H^2 \right]$$

$$\le 8\eta H^2 K + 2\eta H \sum_{k=1}^{K} \sum_{\pi \in \Omega} \omega^k(\pi) \sum_{h=1}^{H} \left( \sum_{s,a} q_h^{\pi, \bar{p}^k}(s,a) \cdot \widehat{c}_h^k(s,a) \right)^2$$

$$= 8\eta H^2 K + 2\eta H \sum_{k=1}^{K} \sum_{h=1}^{H} \sum_{s,a} \sum_{\pi \in \Omega} \omega^k(\pi) q_h^{\pi, \bar{p}^k}(s,a)^2 \cdot \widehat{c}_h^k(s,a)^2$$

$$\le 8\eta H^2 K + 2\eta H \sum_{k=1}^{K} \sum_{h=1}^{H} \sum_{s,a} \widehat{c}_h^k(s,a)^2 \left( \sum_{\pi \in \Omega} \omega^k(\pi) q_h^{\pi, \bar{p}^k}(s,a)^2 \right)$$

$$\le 8\eta H^2 K + 2\eta H \sum_{k=1}^{K} \sum_{h=1}^{H} \sum_{s,a} \widehat{c}_h^k(s,a)$$

where the second step follows from the fact that $(x+y)^2 \le x^2 + y^2$; the third step uses Cauchy-Schwartz inequality; the forth step follows from the fact $\mathbb{I}\{s_h^k = s, a_h^k = a\} \mathbb{I}\{s_h^k = s', a_h^k = a'\} = 0$ for all $(s,a), (s',a') \in \mathcal{S} \times \mathcal{A}$ such that $(s,a) \ne (s',a')$; the final step uses the fact that $u_h^k(s,a) \ge \sum_{\pi \in \Omega} \omega^k(\pi) q_h^{\pi, \bar{p}^k}(s,a)$ and the definition of loss estimator $\widehat{c}^k$.

Moreover, with Lemma A.7, we can show that the following inequality hold with probability at least $1 - 9\delta$ that

$$8\eta H^2 K + 2\eta H \sum_{k=1}^{K} \sum_{h=1}^{H} \sum_{s,a} \widehat{c}_h^k(s,a) = \mathcal{O}\left( \eta H^2 SAK + \frac{\eta H^2}{\gamma} \iota \right). \quad (23)$$

Similarly, for some part of the second term of Equation (22), we have

$$\eta \sum_{k=1}^{K} \sum_{\pi \in \Omega} \omega^k(\pi) \left(2H - b^k(\pi)\right) \sum_{j=1, j+d^j \geq k}^{k-1} \widehat{c}^k(\pi) \leq 2\eta H \sum_{k=1}^{K} \sum_{j=1, j+d^j \geq k}^{k-1} \sum_{\pi \in \Omega} \omega^k(\pi) \left(\sum_{h=1}^{H} \sum_{s,a} q_h^{\pi, \bar{p}^j}(s,a) \widehat{c}_h^j(s,a)\right)$$

$$= 2\eta H \sum_{k=1}^{K} \sum_{j=1, j+d^j \geq k}^{k-1} \sum_{\pi \in \Omega} \sum_{h=1}^{H} \sum_{s,a} \omega^k(\pi) q_h^{\pi, \bar{p}^j}(s,a) \widehat{c}_h^j(s,a)$$

$$\leq \mathcal{O}\left(\frac{\eta}{\gamma} H^2 d_{max} \iota\right) + 2\eta H \sum_{k=1}^{K} \sum_{j=1, j+d^j \geq k}^{k-1} \sum_{\pi \in \Omega} \sum_{h=1}^{H} \sum_{s,a} \omega^k(\pi) q_h^{\pi, \bar{p}^j}(s,a)$$

$$= \mathcal{O}\left(\frac{\eta}{\gamma} H^2 d_{max} \iota + \eta H^2 D\right) \tag{24}$$

where the third step uses Lemma A.7 under the event that $p \in \cap_k \mathcal{P}^k$, which holds with probability at least $1 - 9\delta$.

On the other hand, the rest of the second part can be be bounded with respect to the conditional independence between loss estimators $\widehat{c}^k$ and $\widehat{c}_j$ for any $j < k$ satisfying $j + d^j \geq k$:

$$\eta \sum_{k=1}^{K} \sum_{\pi \in \Omega} \omega^k(\pi) \widehat{c}^k(\pi) \sum_{j=1, j+d^j \geq k}^{k-1} \widehat{c}^k(\pi)$$

$$\leq \eta \sum_{k=1}^{K} \sum_{j=1, j+d^j \geq k}^{k-1} \sum_{\pi \in \Omega} \omega^k(\pi) \left(\sum_{h=1}^{H} \sum_{s,a} q_h^{\pi, \bar{p}^k}(s,a) \widehat{c}_h^k(s,a)\right) \left(\sum_{h=1}^{H} \sum_{s,a} q_h^{\pi, \bar{p}^j}(s,a) \widehat{c}_h^j(s,a)\right)$$

$$= \eta \sum_{k=1}^{K} \sum_{j=1, j+d^j \geq k}^{k-1} \sum_{h=1}^{H} \sum_{s,a} \sum_{h'=1}^{H} \sum_{s',a'} \widehat{c}_h^k(s,a) \widehat{c}_{h'}^j(s',a') \left(\sum_{\pi \in \Omega} \omega^k(\pi) \cdot q_h^{\pi, \bar{p}^k}(s,a) q_{h'}^{\pi, \bar{p}^j}(s',a')\right)$$

where the first step uses the definition of loss estimators. Similarly, we have the following inequality holds with probability at least $1 - 12\delta$ that

$$\eta \sum_{k=1}^{K} \sum_{j=1, j+d^j \geq k}^{k-1} \sum_{h=1}^{H} \sum_{s,a} \sum_{h'=1}^{H} \sum_{s',a'} \widehat{c}_h^k(s,a) \widehat{c}_{h'}^j(s',a') \left(\sum_{\pi \in \Omega} \omega^k(\pi) \cdot q_h^{\pi, \bar{p}^k}(s,a) q_{h'}^{\pi, \bar{p}^j}(s',a')\right)$$

$$\leq \eta \sum_{k=1}^{K} \sum_{j=1, j+d^j \geq k}^{k-1} \sum_{h=1}^{H} \sum_{s,a} \sum_{h'=1}^{H} \sum_{s',a'} \widehat{c}_h^k(s,a) \left(\sum_{\pi \in \Omega} \omega^k(\pi) \cdot q_h^{\pi, \bar{p}^k}(s,a) q_{h'}^{\pi, \bar{p}^j}(s',a')\right) + \mathcal{O}\left(\frac{\eta}{\gamma} H^2 d_{max} \iota\right)$$

$$\leq \eta \sum_{k=1}^{K} \sum_{j=1, j+d^j \geq k}^{k-1} \sum_{h=1}^{H} \sum_{s,a} \sum_{h'=1}^{H} \sum_{s',a'} \sum_{\pi \in \Omega} \omega^k(\pi) \cdot q_h^{\pi, \bar{p}^k}(s,a) q_{h'}^{\pi, \bar{p}^j}(s',a') + \mathcal{O}\left(\frac{\eta}{\gamma} H^2 d_{max} \iota\right)$$

$$= \eta \sum_{k=1}^{K} \sum_{j=1, j+d^j \geq k}^{k-1} \sum_{\pi \in \Omega} \omega^k(\pi) \left(\sum_{h=1}^{H} \sum_{s,a} q_h^{\pi, \bar{p}^k}(s,a)\right) \left(\sum_{h=1}^{H} \sum_{s,a} q_h^{\pi, \bar{p}^j}(s,a)\right) + \mathcal{O}\left(\frac{\eta}{\gamma} H^2 d_{max} \iota\right)$$

$$= \mathcal{O}\left(\frac{\eta}{\gamma} H^2 d_{max} \iota\right) + \eta H^2 \sum_{k=1}^{K} \sum_{j=1, j+d^j \geq k}^{k-1} 1 = \mathcal{O}\left(\frac{\eta}{\gamma} H^2 d_{max} \iota\right) + \eta H^2 \sum_{j=1}^{K} \sum_{k=1, k>j, k \leq j+d^j}^{K} 1$$

$$= \mathcal{O}\left(\eta H^2 D + \frac{\eta}{\gamma} H^2 d_{max} \iota\right) \tag{25}$$

where the first and second step apply Lemma A.7 twice under the event that $p \in \cap_k \mathcal{P}^k$, based on the fact that $q_{h'}^{\pi, \bar{p}^j}(s',a') \leq 1$ and $\sum_{\pi \in \Omega} \omega^k(\pi) \cdot q_h^{\pi, \bar{p}^k}(s,a) q_{h'}^{\pi, \bar{p}^j}(s',a') \leq \sum_{\pi \in \Omega} \omega^k(\pi) \cdot q_h^{\pi, \bar{p}^k}(s,a) \leq u_h^k(s,a)$.

Combining Equations (23) to (25) yields the following bound of DRIFT with probability at least $1 - 30\delta$ under the event $p \in \cap_{k=1}^{K} \mathcal{P}^k$:

$$\text{DRIFT} = \mathcal{O}\left(\eta H^2 \left(D + H^2 SAK\right) + \frac{\eta}{\gamma} H^2 \left(d_{max} + 1\right) \iota\right). \tag{26}$$

Finally, combining the bounds for CHEATING-REGRET and DRIFT in Equations (19) and (26) concludes the proof. $\quad\square$

## A.5 Supplementary Lemmas

In this section, we list the supplementary lemmas which directly attained from the previous work [22].

**Lemma A.5** (Lemma 4 of [22]). *With probability at least $1 - 6\delta$, for any collection of transition functions $\{p_k^{s_h}\}_{s \in \mathcal{S}, h \in [H]}$ such that $p_k^{s_h}$ belongs to the confidence set $\mathcal{P}^k$ defined by Algorithm 5 for all every $(s, h) \in \mathcal{S} \times [H]$, we have*

$$\sum_{k=1}^{K} \sum_{s,h} \left| q^{\pi^k, p_k^{s_h}}(s_h) - q^{\pi^k, p}(s_h) \right| = \mathcal{O}\left( H^2 S \sqrt{AK \log\left(\frac{HSAK}{\delta}\right)} + H^3 S^3 A \log^3\left(\frac{HSAK}{\delta}\right) \right).$$

**Lemma A.6** (Lemma 10 of [22]). *With probability at least $1 - 2\delta$, we have for all $h \in [H]$,*

$$\sum_{k=1}^{K} \sum_{s \in \mathcal{S}, a \in \mathcal{A}} \frac{q_h^{\pi^k, p}(s, a)}{\sqrt{n_h^k(s, a) \vee 1}} = \mathcal{O}\left( \sqrt{SAK} + SA \log K + \log\left(\frac{H}{\delta}\right) \right),$$

*and*

$$\sum_{k=1}^{K} \sum_{s \in \mathcal{S}, a \in \mathcal{A}} \frac{q_h^{\pi^k, p}(s, a)}{n_h^k(s, a) \vee 1} = \mathcal{O}\left( SA \log K + \log\left(\frac{H}{\delta}\right) \right)$$

*where $p$ here is the true transition function, and $q_h^{\pi^k, p}(s, a)$ denotes the probability of visiting state-action pair $(s, a)$ at step $h$ via the policy $\pi^k$ for episode $k$.*

**Lemma A.7** (Lemma 11 of [22]). *For any sequence of functions $\alpha_1, \alpha_2, \ldots \alpha_K$ such that $\alpha_k \in [0, 2\gamma]^{\mathcal{S} \times \mathcal{A}}$ is $\mathcal{F}_k$-measurable for all $k$, with probability at least $1 - \delta$ we have for every $h \in [H]$ that*

$$\sum_{k=1}^{K} \sum_{s,a} \alpha_k(s, a) \left( \widehat{c}_h^k(s, a) - \frac{q_h^{\pi^k, p}(s, a)}{u_h^k(s, a)} \cdot c_h^k(s, a) \right) \leq \mathcal{O}\left( \log\frac{H}{\delta} \right)$$

*where $q_h^{\pi^k, p}(s, a)$ is the true probability of visiting state-action pair $(s, a)$ at step $h$ in episode $k$, and $u_h^k(s, a)$ defined in Algorithm 4 is the upper occupancy bound of this probability.*

**Lemma A.8** (Lemma 14 of [22]). *For any policy $\pi^\star$, with probability at least $1 - 6\delta$, Algorithm 4 ensures that*

$$\sum_{k=1}^{K} \left\langle q^{\pi^\star, p}, \widehat{c}^k - c^k \right\rangle = \mathcal{O}\left( \frac{H}{\gamma} \log\left(\frac{HSA}{\delta}\right) \right).$$

---
**Algorithm 6** Delayed UOB-FTRL with Normal Loss Estimator
---
**Input:** State space $\mathcal{S}$, Action space $\mathcal{A}$, Horizon $H$, Number of episodes $K$, Learning rate $\eta > 0$, Exploration parameter $\gamma > 0$, Confidence parameter $\delta > 0$.

**Initialization:** Set $\pi_h^1(a \mid s) = \frac{1}{A}, q_h^1(s,a,s') = \frac{1}{S^2 A}, n_h^1(s,a) = 0, n_h^1(s,a,s')$ for every $(s,a,s',h) \in \mathcal{S} \times \mathcal{A} \times \mathcal{S} \times [H]$ and $\mathcal{P}^1$ be the set of all transition functions.

**for** $k = 1,2,...,K$ **do**

    Play episode $k$ with policy $\pi^k$ and observe trajectory $\{(s_h^k, a_h^k)\}_{h=1}^H$.

    Define confidence set $\mathcal{P}^{k+1}$ by Algorithm 5.

    **for** $j : j + d^j = k$ **do**

        Observe feedback $\{c_h^j(s_h^j, a_h^j)\}_{h=1}^H$.

        Compute upper occupancy bound $u_h^j(s,a) = \max_{p' \in \mathcal{P}^j} q_h^{p',\pi^j}(s,a)$.

        Compute loss estimator $\widehat{c}_h^j(s,a) = \frac{c_h^j(s,a)\mathbb{I}\{s_h^j=s,a_h^j=a\}}{u_h^j(s,a)+\gamma}$ for every $(s,a,h) \in \mathcal{S} \times \mathcal{A} \times [H]$.

    **end for**

    Update occupancy measure:

$$q^{k+1} = \arg \min_{q \in \cap_{j=1}^{k+1}\Delta(\mathcal{M},j)} \left\langle q, \sum_{j:j+d^j \leq k} \widehat{c}^j \right\rangle + \phi(q),$$

    where $\phi(q) = \frac{1}{\eta}\sum_{h,s,a,s'} q_h(s,a,s')\log q_h(s,a,s')$ is the Shannon entropy regularizer, and $\Delta(\mathcal{M},k) = \{q^{\pi,p'} \mid \pi \in (\Delta_{\mathcal{A}})^{\mathcal{S}\times[H]}, p' \in \mathcal{P}^k\}$.

    Update policy: $\pi_h^{k+1}(a \mid s) = \frac{\sum_{s'} q_h^{k+1}(s,a,s')}{\sum_{a'}\sum_{s'} q_h^{k+1}(s,a',s')}$ for every $(s,a,h) \in \mathcal{S} \times \mathcal{A} \times [H]$.

**end for**
---

## B  FTRL with normal loss estimator

In this section, we show that applying the FTRL framework with normal loss estimators and fixed amount Shannon entropy can achieve $\widetilde{\mathcal{O}}\left(\sqrt{K} + \sqrt{D}\right)$ expected regret (ignoring dependence on other parameters). We propose Algorithm 6 which based on this simple idea and Theorem B.1 below shows that our algorithm essentially achieves this goal.

As one may noticed that, compared with Algorithm 8 which uses the Online Mirror Descent framework, Algorithm 6 uses $\cap_{j=1}^k \Delta(\mathcal{M},j)$, the set of occupancy measures associated with transition functions that belong to all confidence sets prior to episode $k$, as the decision space to compute $q^k$. This setup is necessary to adopt the FTRL framework for ensuring that a shrinking sequence of decision sets, which is critical to analyze the penalty term as in Lemma B.7. Please see the proof of Lemma B.7 for more details. On the other hand, the unknown underlying transition $p$ belongs to all the confidence sets with high probability, which ensures that the intersection of confidence sets is nonempty with high probability.

**Theorem B.1.** *With confidence parameter* $\delta = \frac{1}{H^2 S^2 A^2 K^5}$, *learning rate* $\eta = \sqrt{\frac{H \log \frac{HSAK}{\delta}}{HSAK+(HSA)^2 D}}$ *and exploration parameter* $\gamma = \sqrt{\frac{\log \frac{HSAK}{\delta}}{SAK}}$, *Algorithm 6 ensures that*

$$\mathbb{E}[R_K] = O\left(H^2 S\sqrt{AK\log(HSAK)} + HSA\sqrt{HD\log(HSAK)} + H^4 S^2 A^2 \log^2(HSAK)\right).$$

### B.1  Proof of the Main Theorem

We first decompose the regret into four terms according to the work of [22]:

$$R_K = \underbrace{\sum_{k=1}^K \left\langle q^{\pi^k} - q^k, c^k \right\rangle}_{\text{EST}} + \underbrace{\sum_{k=1}^K \left\langle q^k, c^k - \widehat{c}^k \right\rangle}_{\text{BIAS}_1} + \underbrace{\sum_{k=1}^K \left\langle q^k - q^*, \widehat{c}^k \right\rangle}_{\text{REG}} + \underbrace{\sum_{k=1}^K \left\langle q^\star, \widehat{c}^k - c^k \right\rangle}_{\text{BIAS}_2}, \tag{27}$$

where $q^k$ is the computed occupancy measure of episode $k$; $q^{\pi^k}$ is the underlying occupancy measure associated with the unknown transition $p$ and policy $\pi^k$; $q^\star$ is the occupancy measure of the optimal policy $\pi^\star$ in hindsight .

Then, with the help of Lemma 4, 6 and 14 of [22], we have the following lemma for EST, BIAS$_1$ and BIAS$_2$.

**Lemma B.2.** *with probability at least $1 - 9\delta$, Algorithm 6 ensures that*

$$\text{EST} = \mathcal{O}\left(H^2 S \sqrt{AK \log\left(\frac{HSAK}{\delta}\right)} + H^4 S^2 A^2 \log^2\left(\frac{HSAK}{\delta}\right)\right),$$

$$\text{BIAS}_1 = \mathcal{O}\left(H^2 S \sqrt{AK \log\left(\frac{HSAK}{\delta}\right)} + \gamma HSAK\right),$$

$$\text{BIAS}_2 = \mathcal{O}\left(\frac{H}{\gamma} \log\left(\frac{HSA}{\delta}\right)\right).$$

*Proof.* Without loss of generality, we convert our MDP setting to that of [22] by setting $\mathcal{X} = \mathcal{S} \times [H]$ and $L = H$. Then, by direct application of Lemma 4, 6 and 14 of [22] (which are combined together in the proof of Theorem 3), we arrive at the high-probability bounds of these terms. Note that, the double epoch scheduling and larger confidence sets of transition functions only changes the constant of regret bound, which is hidden in $\mathcal{O}(\cdot)$ operator. □

Based on the high-probability bound, we have the following corollary for the expected bound of these terms.

**Corollary B.3.** *Algorithm 6 ensures that $\mathbb{E}[\text{EST} + \text{BIAS}_1 + \text{BIAS}_2]$ is bounded at most $\mathcal{O}\left(H^4 S^2 A^2 \log^2\left(\frac{HSAK}{\delta}\right)\right)$ plus:*

$$\mathcal{O}\left(H^2 S \sqrt{AK \log\left(\frac{HSAK}{\delta}\right)} + \gamma HSAK + \frac{H}{\gamma} \log\left(\frac{HSA}{\delta}\right) + HK\delta\right).$$

Then, we prove the following lemma for the expected bound of REG with the help a unique novel analysis, and defer the complete proof to to Appendix B.2.

**Lemma B.4.** *Algorithm 6 ensures that $\mathbb{E}[\text{REG}]$ is bounded by:*

$$\mathcal{O}\left(\frac{H \ln\left(S^2 A\right)}{\eta} + \eta\left(HSAK + (HSA)^2 D\right) + \frac{H^2 S^2 A^2 K^3}{\gamma^2}\delta\right).$$

With the help of above lemmas, we are ready to prove the Theorem B.1.

*Proof of Theorem B.1.* Combining the expected bound of $\text{EST} + \text{BIAS}_1 + \text{BIAS}_2$ in Corollary B.3 and that of REG in Lemma B.4, we are able to show that the expected regret $\mathbb{E}[R_K]$ is bounded by

$$\mathcal{O}\left(H^2 S \sqrt{AK \log\left(\frac{HSAK}{\delta}\right)} + \gamma HSAK + \frac{H}{\gamma} \log\left(\frac{HSA}{\delta}\right) + \frac{H \ln\left(S^2 A\right)}{\eta} + \eta\left(HSAK + (HSA)^2 D\right)\right)$$

$$+ \mathcal{O}\left(\frac{H^2 S^2 A^2 K^3}{\gamma^2}\delta + H^4 S^2 A^2 \log^2\left(\frac{HSAK}{\delta}\right)\right).$$

Finally, selecting a small enough confidence parameter $\delta = \frac{1}{H^2 S^2 A^2 K^5}$ and picking up the learning rate $\eta = \sqrt{\frac{H \log \frac{HSAK}{\delta}}{HSAK + (HSA)^2 D}}$ and the exploration parameter $\gamma = \sqrt{\frac{\log \frac{HSAK}{\delta}}{SAK}}$ ensure that

$$\mathbb{E}[R_K] = O\left(H^2 S \sqrt{AK \log(HSAK)} + HSA \sqrt{HD \log(HSAK)} + H^4 S^2 A^2 \log^2(HSAK)\right). \qquad □$$

## B.2 Bound on the Regret with respect to the Loss Estimators (REG in Eq. (27))

In this part, we focus on REG defined in Eq (27) with delayed feedback of losses, and prove Lemma B.4 through the introduced key steps in Section 4. To this end, we will use the following decomposition of REG in this section:

$$\text{REG} = \sum_{k=1}^{K} \left\langle q^k - q^\star, \widehat{c}^k \right\rangle = \sum_{k=1}^{K} \Phi_k(q^k) + \left\langle q^k, \widehat{c}^k \right\rangle - \Phi'_k(\widehat{q}^k) \qquad (\text{STABILITY})$$

$$+ \sum_{k=1}^{K} \Phi'_k(\widehat{q}^k) - \Phi_k(q^k) - \left( \Phi^C_k(\widetilde{q}'_k) - \Phi^B_k(\widetilde{q}_k) \right) \quad (\text{DELAY-CAUSED DRIFT})$$

$$+ \sum_{k=1}^{K} \Phi^C_k(\widetilde{q}'_k) - \Phi^B_k(\widetilde{q}_k) - \left\langle q^\star, \widehat{c}^k \right\rangle \qquad (\text{PENALTY})$$

where the functions $\Phi_k, \Phi'_k, \Phi^B_k, \Phi^C_k$ and the occupancy measures $q^k, \widehat{q}^k, \widetilde{q}_k, \widetilde{q}'_k$ are defined as

$$\Phi_k(q) = \left\langle q, \widehat{L}^{\text{obs}}_k \right\rangle + \phi(q), \qquad\qquad q^k = \underset{q\in\cap_{j=1}^{k}\Delta(\mathcal{M},j)}{\arg\min} \Phi_k(q),$$

$$\Phi'_k(q) = \left\langle q, \widehat{L}^{\text{obs}}_k + \widehat{c}^k \right\rangle + \phi(q), \qquad\qquad \widehat{q}^k = \underset{q\in\cap_{j=1}^{k}\Delta(\mathcal{M},j)}{\arg\min} \Phi'_k(q),$$

$$\Phi^B_k(q) = \left\langle q, \widehat{L}_k \right\rangle + \phi(q), \qquad\qquad \widetilde{q}_k = \underset{q\in\cap_{j=1}^{k}\Delta(\mathcal{M},j)}{\arg\min} \Phi^B_k(q),$$

$$\Phi^C_k(q) = \left\langle q, \widehat{L}_k + \widehat{c}^k \right\rangle + \phi(q), \qquad\qquad \widetilde{q}'_k = \underset{q\in\cap_{j=1}^{k}\Delta(\mathcal{M},j)}{\arg\min} \Phi^C_k(q).$$

with $\widehat{L}_k = \sum_{j=1}^{k-1} \widehat{c}^j$ being the un-delayed cumulative loss estimator prior to episode $k$, and $\widehat{L}^{\text{obs}}_k = \sum_{j=1, j+d^j<k}^{k-1} \widehat{c}^j$ being the received cumulative loss estimator.

On the other hand, with the help of $F^\star_k(x) = -\min_{q\in\cap_{j=1}^{k}\Delta(\mathcal{M},j)} \{\phi(x) - \langle x, q\rangle\}$, the convex conjugate with respect to $\phi(\cdot)$, these functions and occupancy measures ensures that

$$\Phi_k(q^k) = -F^\star_k\left(-\widehat{L}^{\text{obs}}_k\right), \Phi'_k(\widehat{q}^k) = -F^\star_k\left(-\widehat{L}^{\text{obs}}_k - \widehat{c}^k\right), \Phi^B_k(\widetilde{q}_k) = -F^\star_k\left(-\widehat{L}_k\right), \Phi^C_k(\widetilde{q}'_k) = -F^\star_k\left(-\widehat{L}_k - \widehat{c}^k\right).$$

In addition, according to the property of convex conjugates, these occupancy measures are able to be presented as the gradient of the convex conjugate with different inputs as

$$q^k = \nabla F^\star_k\left(-\widehat{L}^{\text{obs}}_k\right), \widehat{q}^k = \nabla F^\star_k\left(-\widehat{L}^{\text{obs}}_k - \widehat{c}^k\right), \widetilde{q}_k = \nabla F^\star_k\left(-\widehat{L}_k\right), \widetilde{q}'_k = \nabla F^\star_k\left(-\widehat{L}_k - \widehat{c}^k\right).$$

For notational convenience, we denote $\widehat{\Delta}_k = \widehat{L}_k - \widehat{L}^{\text{obs}}_k$ as the summation of un-received loss estimators prior to episode $k$, that is, $\widehat{\Delta}_k = \sum_{j=1, j+d^j\geq k}^{k-1} \widehat{c}_j$. Thus, $\Phi^B_k(\widetilde{q}_k)$ and $\Phi^C_k(\widetilde{q}'_k) = -F^\star_k\left(-\widehat{L}_k - \widehat{c}^k\right)$ can be represented as

$$\Phi^B_k(\widetilde{q}'_k) = -F^\star_k\left(-\widehat{L}^{\text{obs}}_k - \widehat{\Delta}_k\right), \Phi^C_k(\widetilde{q}'_k) = -F^\star_k\left(-\widehat{L}^{\text{obs}}_k - \widehat{\Delta}_k - \widehat{c}^k\right).$$

With the help of these definitions, we are now ready to bound the terms STABILTY, DELAY-CAUSED DRFIT and PENALTY in following lemmas.

**Lemma B.5.** *(Stability) With fixed learning rate $\eta > 0$ and exploration $\gamma > 0$, Algorithm 6 ensures that*

$$\sum_{k=1}^{K} \Phi_k(q^k) + \left\langle q^k, \widehat{c}^k \right\rangle - \Phi'_k(\widehat{q}^k) \leq \eta \sum_{k=1}^{K} \sum_{h,s,a} q^k_h(s,a)\widehat{c}^k_h(s,a)^2.$$

*Proof.* Let $D_k(u,v) = \phi(u) - \phi(v) - \langle u - v, \nabla\phi(v)\rangle$ be the Bregman divergence with the convex regularizer $\phi$. Then,

$$\Phi_k(q^k) = \left\langle q^k, \widehat{L}^{\text{obs}}_k \right\rangle + \phi(q^k) = \left\langle \widehat{q}^k, \widehat{L}^{\text{obs}}_k \right\rangle + \phi(\widehat{q}^k) - \left( \left\langle \widehat{q}^k - q^k, \widehat{L}^{\text{obs}}_k \right\rangle + \phi(\widehat{q}^k) - \phi(q^k) \right)$$

$$\leq \left\langle \widehat{q}^k, \widehat{L}^{\text{obs}}_k \right\rangle + \phi(\widehat{q}^k) - \left( -\left\langle \widehat{q}^k - q^k, \nabla\phi(q^k) \right\rangle + \phi(\widehat{q}^k) - \phi(q^k) \right)$$

$$= \left\langle \widehat{q}^k, \widehat{L}^{\text{obs}}_k \right\rangle + \phi(\widehat{q}^k) - D_k(\widehat{q}^k, q^k) = \Phi'_k(\widehat{q}^k) - \left\langle \widehat{q}^k, \widehat{c}^k \right\rangle - D_k(\widehat{q}^k, q^k),$$

where the third step follows from the first order optimality of $q^k$ with respect to $\Phi_k$, in other words, $\left\langle \widehat{q}^k - q^k, \widehat{L}_k^{\mathrm{obs}} + \nabla\phi(q^k) \right\rangle \geq 0$. Rearranging terms and adding $\langle q^k, \widehat{c}^k \rangle$ on both sides give us the following inequality:

$$\Phi_k(q^k) + \langle q^k, \widehat{c}^k \rangle - \Phi_k'(\widehat{q}^k) \leq \langle q^k - \widehat{q}^k, \widehat{c}^k \rangle - D_k(\widehat{q}^k, q^k).$$

To bound the right hand side term, we relax the constraints and taking the maximum as:

$$\langle q^k - \widehat{q}^k, \widehat{c}^k \rangle - D_k(\widehat{q}^k, q^k) \leq \max_{q \in \mathbb{R}_{\geq 0}^{S \times \mathcal{A} \times [H] \times S}} \langle q^k - q, \widehat{c}^k \rangle - D_k(q, q^k) = \langle q^k - \xi^k, \widehat{c}^k \rangle - D_k(\xi^k, q^k),$$

where $\xi_k$ denotes the maximizer point. Setting the gradient to zero gives the equality that $\nabla\phi(q^k) - \nabla\phi(\xi^k) = \widehat{c}^k$. By direct calculation, one can verify that $\xi_h^k(s, a, s') = q_h^k(s, a, s') \cdot \exp\left(-\eta \widehat{c}_h^k(s, a)\right)$ for all transition tuples. Therefore, we have the following inequality that

$$\langle q^k - \xi^k, \widehat{c}^k \rangle - D_k(\xi^k, q^k) = \langle q^k - \xi^k, \widehat{c}^k \rangle - \phi(\xi^k) + \phi(q^k) - \langle q^k - \xi^k, \nabla\phi(q^k) \rangle = D_k(q^k, \xi^k)$$

$$= \frac{1}{\eta} \sum_{h=1}^{H} \sum_{s,a,s'} \left( q_h^k(s, a, s') \ln\left(\frac{q_h^k(s, a, s')}{\xi_h^k(s, a, s')}\right) - q_h^k(s, a, s') + \xi_h^k(s, a, s') \right)$$

$$= \frac{1}{\eta} \sum_{h=1}^{H} \sum_{s,a,s'} q_h^k(s, a, s') \left(\eta \widehat{c}_h^k(s, a) - 1 + \exp\left(-\eta \widehat{c}_h^k(s, a)\right)\right)$$

$$\leq \eta \sum_{h=1}^{H} \sum_{s,a,s'} q_h^k(s, a, s') \widehat{c}_h^k(s, a)^2 = \eta \sum_{h=1}^{H} \sum_{s,a} q_h^k(s, a) \widehat{c}_h^k(s, a)^2,$$

where the second step uses $\nabla\phi(q^k) - \nabla\phi(\xi_k) = \widehat{c}^k$; the forth step follows from the fact that $e^{-x} \leq 1 - x + x^2$ for any $x \geq 0$. Finally, taking the summation over all episodes finishes the proof. $\square$

**Lemma B.6.** *(Delay-caused Drift) Algorithm 6 guarantees that*

$$\sum_{k=1}^{K} \Phi_k'(\widehat{q}^k) - \Phi_k(q^k) - \left(\Phi_k^C(\widetilde{q}_k') - \Phi_k^B(\widetilde{q}_k)\right) \leq 2\eta \sum_{k=1}^{K} \left(\sum_{h=1}^{H} \sum_{s,a} \widehat{c}_h^k(s, a)\right) \cdot \left(\sum_{h=1}^{H} \sum_{s,a} \widehat{\Delta}_h^k(s, a)\right).$$

*Proof.* With the help of the convex conjugate $F_k^\star(\cdot)$, we have the following inequality holds for some $\theta \in [0, 1]$ that:

$$\Phi_k'(\widehat{q}^k) - \Phi_k(q^k) - \left(\Phi_k^C(\widetilde{q}_k') - \Phi_k^B(\widetilde{q}_k)\right) = -F_k^\star(-\widehat{L}_k^{\mathrm{obs}} - \widehat{c}^k) + F_k^\star(-\widehat{L}_k^{\mathrm{obs}}) - \left(-F_k^\star(-\widehat{L}_k - \widehat{c}^k) + F_k^\star(-\widehat{L}_k)\right)$$

$$= \int_0^1 \left\langle \widehat{c}^k, \nabla F_k^\star(-\widehat{L}_k^{\mathrm{obs}} - x\widehat{c}^k) \right\rangle dx - \int_0^1 \left\langle \widehat{c}^k, \nabla F_k^\star(-\widehat{L}_k - x\widehat{c}^k) \right\rangle dx$$

$$= \int_0^1 \left\langle \widehat{c}^k, \nabla F_k^\star(-\widehat{L}_k^{\mathrm{obs}} - x\widehat{c}^k) - \nabla F_k^\star(-\widehat{L}_k - x\widehat{c}^k) \right\rangle dx$$

$$= \left\langle \widehat{c}^k, \nabla F_k^\star(-\widehat{L}_k^{\mathrm{obs}} - \theta\widehat{c}^k) - \nabla F_k^\star(-\widehat{L}_k - \theta\widehat{c}^k) \right\rangle,$$

where the second step uses Newton-Leibniz theorem; the forth step uses the mean value theorem. To analyze the right hand side, we define the functions $W$ and $W'$ as

$$W(q) = \left\langle q, \widehat{L}_k^{\mathrm{obs}} + \theta\widehat{c}^k \right\rangle + \phi(q) \quad ; \quad W'(q) = \left\langle q, \widehat{L}_k + \theta\widehat{c}^k \right\rangle + \phi(q),$$

and denote their minimizer occupancy measures within the decision set $\cap_{j=1}^k \Delta(\mathcal{M}, j)$ by $u$ and $v$. According to the properties of convex conjugate, we have $u = \nabla F_k^\star(-\widehat{L}_k^{\mathrm{obs}} - \theta\widehat{c}^k)$ and $v = \nabla F_k^\star(-\widehat{L}_k - \theta\widehat{c}^k)$.

To analyze $\langle u - v, \widehat{c}^k \rangle$, we first lower bound $W(u) + \left\langle u, \widehat{\Delta}_k \right\rangle - W'(v)$ as

$$W(u) + \left\langle u, \widehat{\Delta}_k \right\rangle - W'(v) = W'(u) - W'(v) = \langle \nabla W'(v), u - v \rangle + \frac{1}{2} \|u - v\|_{\nabla^2\phi(\xi)}^2 \geq \frac{1}{2} \|u - v\|_{\nabla^2\phi(\xi)}^2,$$

where the second step applies Taylor's expansion with $\xi$ being an intermediate point between $u$ and $v$; the last step uses the first order optimality condition of $v$. On the other hand, we can upper $W(u) + \left\langle u, \widehat{\Delta}_k \right\rangle - W'(v)$ as

$$W(u) + \left\langle u, \widehat{L}_k - \widehat{L}_k^{\text{obs}} \right\rangle - W'(v) = W(u) - W(v) + \left\langle u - v, \widehat{L}_k - \widehat{L}_k^{\text{obs}} \right\rangle \leq \left\langle u - v, \widehat{L}_k - \widehat{L}_k^{\text{obs}} \right\rangle$$

$$\leq \|u - v\|_{\nabla^2 \phi(\xi)} \left\| \widehat{L}_k - \widehat{L}_k^{\text{obs}} \right\|_{\nabla^{-2} \phi(\xi)},$$

where the second step uses the optimality of $u$, and the last step comes from Hölder's inequality. Combining the lower bound and upper bound, we arrives at the following inequality

$$\|u - v\|_{\nabla^2 \phi(\xi)} \leq 2 \left\| \widehat{L}_k - \widehat{L}_k^{\text{obs}} \right\|_{\nabla^{-2} \phi(\xi)}.$$

Therefore, we can upper bound the term $\left\langle \widehat{c}^k, u - v \right\rangle$ with the help of Hölder's inequality again as

$$\left\langle \widehat{c}^k, u - v \right\rangle \leq \left\| \widehat{c}^k \right\|_{\nabla^{-2} \phi(\xi)} \|u - v\|_{\nabla^2 \phi(\xi)} \leq 2 \left\| \widehat{c}^k \right\|_{\nabla^{-2} \phi(\xi)} \left\| \widehat{L}_k - \widehat{L}_k^{\text{obs}} \right\|_{\nabla^{-2} \phi(\xi)}.$$

By direct calculation, one can verify the following:

$$2 \left\| \widehat{c}^k \right\|_{\nabla^{-2} \phi(\xi)} \cdot \left\| \widehat{\Delta}_k \right\|_{\nabla^{-2} \phi(\xi)} = 2 \sqrt{\eta \sum_{h=1}^{H} \sum_{s,a,s'} \widehat{c}_h^k(s,a)^2 \xi(s,a,s')} \cdot \sqrt{\eta \sum_{h=1}^{H} \sum_{s,a,s'} \widehat{\Delta}_h^k(s,a)^2 \xi(s,a,s')}$$

$$\leq 2\eta \sqrt{\sum_{h=1}^{H} \sum_{s,a} \widehat{c}_h^k(s,a)^2} \cdot \sqrt{\sum_{h=1}^{H} \sum_{s,a} \widehat{\Delta}_h^k(s,a)^2}$$

$$\leq 2\eta \left( \sum_{h=1}^{H} \sum_{s,a} \widehat{c}_h^k(s,a) \right) \cdot \left( \sum_{h=1}^{H} \sum_{s,a} \widehat{\Delta}_h^k(s,a) \right),$$

where the second step follows from the fact that $\xi$ is a valid occupancy measure and $\sum_{s'} \xi(s,a,s') = \xi(s,a) \leq 1$ holds for all state-action pairs. Taking the summation over all episodes concludes the proof. $\qquad\square$

**Lemma B.7.** *(Penalty) With the shrinking decision set sequence that $\cap_{j=1}^{k+1} \Delta(\mathcal{M}, j) \subset \cap_{j=1}^{k} \Delta(\mathcal{M}, j)$ for $k = 1, \ldots K - 1$, Algorithm 6 ensures that*

$$\sum_{k=1}^{K} \Phi_k^C(\widetilde{q}_k') - \Phi_k^B(\widetilde{q}_k) - \left\langle q^\star, \widehat{c}^k \right\rangle \leq \frac{H \ln \left( S^2 A \right)}{\eta}.$$

*Proof.* First, we observe that

$$\Phi_k^C(\widetilde{q}_k') = \min_{q \in \cap_{j=1}^{k} \Delta(\mathcal{M}, j)} \left\langle q, \widehat{L}_k + \widehat{c}^k \right\rangle + \phi(q) \leq \min_{q \in \cap_{j=1}^{k+1} \Delta(\mathcal{M}, j)} \left\langle q, \widehat{L}_k + \widehat{c}^k \right\rangle + \phi(q)$$

$$= \min_{q \in \cap_{j=1}^{k+1} \Delta(\mathcal{M}, j)} \left\langle q, \widehat{L}_{k+1} \right\rangle + \phi(q) = \Phi_{k+1}^B(\widetilde{q}_{k+1}),$$

where the second step follows from the fact that $\mathcal{P}^{k+1} \subset \mathcal{P}^k$ by the definition. Therefore, we have the following inequality:

$$\sum_{k=1}^{K} \Phi_k^C(\widetilde{q}_k') - \Phi_k^B(\widetilde{q}_k) - \left\langle q^\star, \widehat{c}^k \right\rangle = \Phi_K^C(\widetilde{q}_K') - \Phi_1^B(\widetilde{q}_1) - \left\langle q^\star, \widehat{L}_{K+1} \right\rangle + \sum_{k=1}^{K-1} \Phi_k^C(\widetilde{q}_k') - \Phi_{k+1}^B(\widetilde{q}_{k+1})$$

$$\leq \Phi_K^C(\widetilde{q}_K') - \Phi_1^B(\widetilde{q}_1) - \left\langle q^\star, \widehat{L}_{K+1} \right\rangle \leq \phi(q^\star) - \phi(\widetilde{q}_1) \leq \frac{H \ln \left( S^2 A \right)}{\eta},$$

where the third step follows from the optimality of $\widetilde{q}_K'$ and the last steps follows the standard argument of Shannon entropy (such as, Lemma 12 of [22]). $\qquad\square$

We are now ready to prove Lemma B.4 by combining the results of Lemmas B.5 to B.7 and taking the expectation.

*Proof of Lemma B.4.* By combining Lemmas B.5 to B.7, we have REG bounded as

$$\text{REG} \leq \frac{H \ln (S^2 A)}{\eta} + \eta \sum_{k=1}^{K} \sum_{h=1}^{H} \sum_{s,a} q_h^k(s,a) \widehat{c}_h^k(s,a)^2 + 2\eta \sum_{k=1}^{K} \sum_{h=1}^{H} \sum_{s,a} \sum_{h'=1}^{H} \sum_{s',a'} \widehat{c}_h^k(s,a) \widehat{\Delta}_{h'}^k(s',a').$$

To analyze the expectation, we use the indicator $Z_k = \mathbb{I}\{p \notin \mathcal{P}_k\}$ to denote the event that the true transition function $p$ is not included in the confidence set of episode $k$. Clearly, one can verify that $q_h^k(s,a) \leq Z_k + u_h^k(s,a)$ and $q_h^{\pi^k}(s,a) \leq Z_k + u_h^k(s,a)$ due to the definition of upper occupancy bound $u_k$ and the property of occupancy measures. Therefore, we are able to bound $\mathbb{E}[\text{REG}]$ by

$$\frac{H \ln (S^2 A)}{\eta} + \eta \mathbb{E} \left[ \sum_{k=1}^{K} \sum_{h=1}^{H} \sum_{s,a} q_h^k(s,a) \widehat{c}_h^k(s,a)^2 + 2 \sum_{h=1}^{H} \sum_{s,a} \sum_{h'=1}^{H} \sum_{s',a'} \widehat{c}_h^k(s,a) \widehat{\Delta}_h^k(s,a) \right]$$

$$\leq \frac{H \ln (S^2 A)}{\eta} + \eta \mathbb{E} \left[ \sum_{k=1}^{K} \mathbb{E}_k \left[ \sum_{h=1}^{H} \sum_{s,a} \widehat{c}_h^k(s,a) + 2\eta \sum_{h=1}^{H} \sum_{s,a} \sum_{h'=1}^{H} \sum_{s',a'} \widehat{c}_h^k(s,a) \widehat{\Delta}_h^k(s,a) \right] \right]$$

$$\leq \frac{H \ln (S^2 A)}{\eta} + \eta \mathbb{E} \left[ \sum_{k=1}^{K} \sum_{h=1}^{H} \sum_{s,a} \frac{q_h^{\pi^k}(s,a)}{u_h^k(s,a) + \gamma} + 2 \sum_{j=1,j+d^j \geq k}^{k-1} \sum_{h'=1}^{H} \sum_{s',a'} \frac{q_h^{\pi^k}(s,a)}{u_h^k(s,a) + \gamma} \frac{q_{h'}^{\pi^j}(s',a')}{u_{h'}^j(s',a') + \gamma} \right]$$

$$\leq \frac{H \ln (S^2 A)}{\eta} + \eta \left( HSAK + 2(HSA)^2 D \right) + \frac{HSAK + 4(HSA)^2 D}{\gamma^2} \cdot \mathbb{E} \left[ \sum_{k=1}^{K} Z_k \right],$$

where the first step uses the fact that $q_h^k(s,a) \leq u_h^k(s,a)$ for any state-action pair; the second step uses the definition of loss estimators; the third step follows from the fact that $q_h^{\pi^k}(s,a) \leq Z_k + u_h^k(s,a)$.

According to Lemma 2 of [22], we have the expectation of $\mathbb{E}\left[ \sum_{k=1}^{K} Z_k \right]$ bounded by $4K\delta$, and the following upper bound of $\mathbb{E}[\text{REG}]$:

$$\mathcal{O} \left( \frac{H \ln (S^2 A)}{\eta} + \eta \left( HSAK + (HSA)^2 D \right) + \frac{H^2 S^2 A^2 K^3}{\gamma^2} \delta \right).$$

$\square$

**Algorithm 7** Delayed O-REPS with delay-adapted estimator and known transition

---

**Input:** State space $\mathcal{S}$, Action space $\mathcal{A}$, Horizon $H$, Number of episodes $K$, Transition function $p$, Learning rate $\eta > 0$, Exploration parameter $\gamma > 0$.

**Initialization:** Set $\pi_h^1(a \mid s) = \frac{1}{A}$, $q_h^1(s,a) = \frac{1}{SA}$ for every $(s,a,h) \in \mathcal{S} \times \mathcal{A} \times [H]$.

**for** $k = 1, 2, ..., K$ **do**

    Play episode $k$ with policy $\pi^k$ and observe trajectory $\{(s_h^k, a_h^k)\}_{h=1}^H$.

    **for** $j : j + d^j = k$ **do**

        Observe feedback $\{c_h^j(s_h^j, a_h^j)\}_{h=1}^H$.

        Compute loss estimator $\hat{c}_h^j(s,a) = \frac{c_h^j(s,a)\mathbb{I}\{s_h^j=s,a_h^j=a\}}{\max\{q_h^j(s,a),q_h^k(s,a)\}+\gamma}$ for every $(s,a,h) \in \mathcal{S} \times \mathcal{A} \times [H]$.

    **end for**

    Update occupancy measure:

$$q^{k+1} = \arg\min_{q \in \Delta(\mathcal{M})} \eta \left\langle q, \sum_{j:j+d^j=k} \hat{c}^j \right\rangle + \mathrm{KL}(q \parallel q^k), \tag{28}$$

    where $\mathrm{KL}(q \parallel q') = \sum_{h,s,a} q_h(s,a) \ln \frac{q_h(s,a)}{q_h'(s,a)} + q_h'(s,a) - q_h(s,a)$.

    Update policy: $\pi_h^{k+1}(a \mid s) = \frac{q_h^{k+1}(s,a)}{\sum_{a'} q_h^{k+1}(s,a')}$ for every $(s,a,h) \in \mathcal{S} \times \mathcal{A} \times [H]$.

**end for**

---

## C    Delayed O-REPS with delay-adapted estimator

Explicitly solving this optimization problem in Eq. (28), we get [51]:

$$q_h^{k+1}(s,a) = \frac{q_h^k(s,a)e^{B_h^k(s,a|v^k)}}{Z_h^k(v^k)},$$

for:

$$B_h^k(s,a \mid v) = v_h(s) - \eta \sum_{j:j+d^j=k} \hat{c}_h^j(s,a) - \sum_{s'} p_h(s' \mid s,a)v_{h+1}(s')$$

$$Z_h^k(v) = \sum_{s,a} q_h^k(s,a)e^{B_h^k(s,a|v)}$$

$$v^k = \arg\min_v \sum_h \log Z_h^k(v).$$

These different formulations will be helpful in the regret analysis.

**Theorem C.1.** *Running O-REPS with the delay-adapted estimator, $\eta = \gamma = \min\{\sqrt{\frac{\log \frac{HSA}{\delta}}{SAK}}, \sqrt{\frac{\log \frac{HSA}{\delta}}{\sqrt{HSAD}}}\}$ guarantees, with probability $1 - \delta$,*

$$R_K = O\left(H\sqrt{SAK \log \frac{HSA}{\delta}} + (HSA)^{1/4} \cdot H\sqrt{D \log \frac{HSA}{\delta}} + H^{3/2}d_{max} \log \frac{H}{\delta}\right).$$

## C.1 The good event

Let $\tilde{\mathcal{H}}^k$ be the history of episodes $\{j : j + d^j < k\}$. Define the following events:

$$E^c = \left\{ \sum_{k=1}^K \langle \mathbb{E}[\hat{c}^k \mid \widetilde{\mathcal{H}}^{k+d^k}] - \hat{c}^k, q^k \rangle \leq 4H \sqrt{K \log \frac{10}{\delta}} \right\}$$

$$E^{\hat{c}} = \left\{ \sum_{k=1}^K \langle |q^k - q^{k+d^k}|, \hat{c}^k \rangle \leq 4 \sum_{k=1}^K \langle |q^k - q^{k+d^k}|, c^k \rangle + \frac{40H \log \frac{10H}{\delta}}{\gamma} \right\}$$

$$E^d = \left\{ \sum_{k,h,s,a} |\mathcal{F}^{k+d^k}| \hat{c}_h^k(s,a) \leq \sum_{k,h,s,a} |\mathcal{F}^{k+d^k}| c_h^k(s,a) + \frac{10H d_{max} \log \frac{10H}{\delta}}{\gamma} \right\}$$

$$E^{sq} = \left\{ \sum_{k=1}^K \sum_{i=1}^K \mathbb{I}\{k \leq i + d^i < k + d^k\} \sum_{h,s,a} \sqrt{q_h^{i+d^i}(s,a)}(\hat{c}_h^i(s,a) - 4c_h^i(s,a)) \leq \frac{10H d_{max} \log \frac{10H}{\delta}}{\gamma} \right\}$$

$$E^{\star} = \left\{ \sum_{k=1}^K \langle \hat{c}^k - c^k, q^{\star} \rangle \leq \frac{H \log \frac{10HSA}{\delta}}{\gamma} \right\}$$

The good event is the intersection of the above events. The following lemma establishes that the good event holds with high probability.

**Lemma C.2** (The Good Event). *Let $\mathbb{G} = E^c \cap E^{\hat{c}} \cap E^d \cap E^{sq} \cap E^{\star}$ be the good event. It holds that $\Pr[\mathbb{G}] \geq 1 - \delta$.*

*Proof.* We show that each of the events $\neg E^c, \neg E^{\hat{c}}, \neg E^d, \neg E^{sq}, \neg E^{\star}$ occur with probability at most $\delta/5$. Then, by a union bound we obtain the statement.

- $\Pr[\neg E^c] < \delta/5$ by Azuma inequality since it is a martingale with respect to the filtration $\{\tilde{\mathcal{H}}^{1+d^1}, \tilde{\mathcal{H}}^{2+d^2}, \dots\}$ where the differences are bounded by $H$.

- $\Pr[\neg E^{\hat{c}}] < \delta/5$ by [11, Lemma E.2] since $\langle |q^k - q^{k+d^k}|, \hat{c}^k \rangle \leq H/\gamma$, and $\mathbb{E}[\langle |q^k - q^{k+d^k}|, \hat{c}^k \rangle \mid \tilde{\mathcal{H}}^{i+d^i}] \leq \langle |q^k - q^{k+d^k}|, c^k \rangle$.

- $\Pr[\neg E^d] < \delta/5$ by [22, Lemma 11].

- $\Pr[\neg E^{sq}] < \delta/5$ by [11, Lemma E.2] in the following way. Denote $Y_i = \sum_k \mathbb{I}\{k \leq i + d^i < k + d^k\} \sum_{h,s,a} \sqrt{q_h^{i+d^i}(s,a)}\hat{c}_h^i(s,a)$ and notice that $Y_i \leq H d_{max}/\gamma$, and that:

$$\mathbb{E}[Y_i \mid \widetilde{\mathcal{H}}^{i+d^i}] \leq \sum_k \mathbb{I}\{k \leq i + d^i < k + d^k\} \sum_{h,s,a} \sqrt{q_h^{i+d^i}(s,a)} c_h^i(s,a).$$

- $\Pr[\neg E^{\star}] < \delta/5$ by Lemma A.8. $\qquad\square$

## C.2 Proof of the Main Theorem

*Proof of Theorem C.1.* By Lemma C.2, the good event holds with probability $1 - \delta$. We now analyze the regret under the assumption that the good event holds. We decompose the regret as follows:

$$R_K = \sum_{k=1}^K \langle q^k - q^{\star}, c^k \rangle$$

$$= \underbrace{\sum_{k=1}^K \langle q^k, c^k - \hat{c}^k \rangle}_{\text{BIAS}_1} + \underbrace{\sum_{k=1}^K \langle q^{\star}, \hat{c}^k - c^k \rangle}_{\text{BIAS}_2} + \underbrace{\sum_{k=1}^K \langle q^k - q^{k+d^k}, \hat{c}^k \rangle}_{\text{DRIFT}} + \underbrace{\sum_{k=1}^K \langle q^{k+d^k} - q^{\star}, \hat{c}^k \rangle}_{\text{REG}}. \tag{29}$$

$\text{BIAS}_2$ is bounded under event $E^\star$ by $O(\frac{H \log \frac{HSA}{\delta}}{\gamma})$, $\text{REG}$ is bounded in Lemma C.3 by $O(\frac{H \log(HSA)}{\eta} + \eta HSAK + \frac{\eta}{\gamma}d_{max} \log \frac{H}{\delta})$, $\text{DRIFT}$ is bounded in Lemma C.4 by $O(\eta\sqrt{H^3SA}(D+K) + \frac{\eta}{\gamma}H^{3/2}d_{max}\log\frac{H}{\delta} + \frac{H\log\frac{H}{\delta}}{\gamma})$, and $\text{BIAS}_1$ is bounded in Lemma C.5 by $O(H\sqrt{K\log\frac{1}{\delta}} + \gamma HSAK + \eta\sqrt{H^3SA}(D+K) + \frac{\eta}{\gamma}H^{3/2}d_{max}\log\frac{H}{\delta})$. Putting everything together:

$$R_K = O\left(H\sqrt{K\log\frac{1}{\delta}} + (\eta+\gamma)HSAK + (\frac{1}{\eta}+\frac{1}{\gamma})H\log\frac{HSA}{\delta} + \eta\sqrt{H^3SA}(D+K) + \frac{\eta}{\gamma}H^{3/2}d_{max}\log\frac{H}{\delta}\right),$$

and plugging in the definitions of $\eta$ and $\gamma$ finishes the proof. $\square$

### C.3 Bound on the Regret with respect to the Loss Estimators and Future Policies (REG in Eq. (29))

**Lemma C.3** (REG Term). *Under the good event,*

$$\sum_{k=1}^{K}\langle q^{k+d^k} - q^\star, \hat{c}^k\rangle = O\left(\frac{H\log(HSA)}{\eta} + \eta HSAK + \frac{\eta}{\gamma}Hd_{max}\log\frac{H}{\delta}\right).$$

*Proof.* Let $\tilde{q}_h^{k+1}(s,a) = q_h^k(s,a)e^{-\eta\sum_{j:j+d^j=k}\hat{c}_h^j(s,a)}$. Taking the log,

$$\eta\sum_{j:j+d^j=k}\hat{c}_h^j(s,a) = \log q_h^k(s,a) - \log\tilde{q}_h^{k+1}(s,a).$$

Hence for any $q$

$$\eta\left\langle\sum_{j:j+d^j=k}\hat{c}_h^j, q^k - q^\star\right\rangle = \langle\log q^k - \log\tilde{q}^{k+1}, q^k - q^\star\rangle = \text{KL}(q^\star \parallel q^k) - \text{KL}(q^\star \parallel \tilde{q}^{k+1}) + \text{KL}(q^k \parallel \tilde{q}^{k+1})$$

$$\leq \text{KL}(q^\star \parallel q^k) - \text{KL}(q^\star \parallel q^{k+1}) - \text{KL}(q^{k+1} \parallel \tilde{q}^{k+1}) + \text{KL}(q^k \parallel \tilde{q}^{k+1})$$
$$\leq \text{KL}(q^\star \parallel q^k) - \text{KL}(q^\star \parallel q^{k+1}) + \text{KL}(q^k \parallel \tilde{q}^{k+1}),$$

where the second equality follows directly the definition of KL, the first inequality is by [50, Lemma 1.2], and the second inequality is since the KL is non-negative. Now, the last term is bounded as follows:

$$\text{KL}(q^k \parallel \tilde{q}^{k+1}) \leq \text{KL}(q^k \parallel \tilde{q}^{k+1}) + \text{KL}(\tilde{q}^{k+1} \parallel q^k)$$

$$= \sum_h\sum_{s,a}\tilde{q}_h^{k+1}(s,a)\log\frac{\tilde{q}_h^{k+1}(s,a)}{q_h^k(s,a)} + \sum_h\sum_{s,a}q_h^k(s,a)\log\frac{q_h^k(s,a)}{\tilde{q}_h^{k+1}(s,a)}$$

$$= \langle q^k - \tilde{q}^{k+1}, \log q^k - \log\tilde{q}^{k+1}\rangle = \eta\left\langle q^k - \tilde{q}^{k+1}, \sum_{j:j+d^j=k}\hat{c}^j\right\rangle.$$

We get that

$$\eta\left\langle\sum_{j:j+d^j=k}\hat{c}^j, q^k - q^\star\right\rangle \leq \text{KL}(q^\star \parallel q^k) - \text{KL}(q^\star \parallel q^{k+1}) + \eta\left\langle q^k - \tilde{q}^{k+1}, \sum_{j:j+d^j=k}\hat{c}^j\right\rangle.$$

Summing over $k$ and dividing by $\eta$, we get

$$\underbrace{\sum_{k=1}^{K}\sum_{j:j+d^j=k}\langle\hat{c}^j, q^k - q^\star\rangle}_{(*)} \leq \frac{\text{KL}(q^\star \parallel q^1) - \text{KL}(q^\star \parallel q^{K+1})}{\eta} + \sum_{k=1}^{K}\left\langle q^k - \tilde{q}^{k+1}, \sum_{j:j+d^j=k}\hat{c}^j\right\rangle$$

$$\leq \frac{\text{KL}(q^\star \parallel q^1)}{\eta} + \sum_{k=1}^{K}\left\langle q^k - \tilde{q}^{k+1}, \sum_{j:j+d^j=k}\hat{c}^j\right\rangle$$

$$\leq \frac{2H\log(SA)}{\eta} + \underbrace{\sum_{k=1}^{K}\left\langle q^k - \tilde{q}^{k+1}, \sum_{j:j+d^j=k}\hat{c}^j\right\rangle}_{(**)},$$

where the last inequality is a standard argument (see [50, 16]). We now rearrange $(*)$ and $(**)$:

$$(*) = \sum_{k=1}^{K} \sum_{j=1}^{K} \mathbb{I}\{j + d^j = k\}\langle \hat{c}^j, q^k - q^\star \rangle = \sum_{j=1}^{K} \sum_{k=1}^{K} \mathbb{I}\{j + d^j = k\}\langle \hat{c}^j, q^k - q^\star \rangle$$

$$= \sum_{j=1}^{K} \langle \hat{c}^j, q^{j+d^j} - q^\star \rangle = \sum_{k=1}^{K} \langle \hat{c}^k, q^{k+d^k} - q^\star \rangle.$$

In a similar way,

$$(**) = \sum_{k=1}^{K} \sum_{j:j+d^j=k} \langle q^k - \tilde{q}^{k+1}, \hat{c}^j \rangle = \sum_{k=1}^{K} \sum_{j=1}^{K} \mathbb{I}\{j + d^j = k\}\langle q^k - \tilde{q}^{k+1}, \hat{c}^j \rangle$$

$$= \sum_{j=1}^{K} \sum_{k=1}^{K} \mathbb{I}\{j + d^j = k\}\langle q^k - \tilde{q}^{k+1}, \hat{c}^j \rangle = \sum_{k=1}^{K} \langle q^{k+d^k} - \tilde{q}^{k+d^k+1}, \hat{c}^k \rangle.$$

This gives us,

$$\sum_{k=1}^{K} \langle \hat{c}^k, q^{k+d^k} - q^\star \rangle \le \frac{2H\log(SA)}{\eta} + \sum_{k=1}^{K} \langle q^{k+d^k} - \tilde{q}^{k+d^k+1}, \hat{c}^k \rangle.$$

It remains to bound the second term on the right hand side:

$$\sum_{k} \langle q^{k+d^k} - \tilde{q}^{k+d^k+1}, \hat{c}^k \rangle = \sum_{k,h,s,a} \hat{c}_h^k(s,a)(q_h^{k+d^k}(s,a) - \tilde{q}_h^{k+d^k+1}(s,a))$$

$$= \sum_{k,h,s,a} \hat{c}_h^k(s,a) \left( q_h^{k+d^k}(s,a) - q_h^{k+d^k}(s,a)e^{-\eta \sum_{j:j+d^j=k+d^k} \hat{c}_h^j(s,a)} \right)$$

$$= \sum_{k,h,s,a} q_h^{k+d^k}(s,a)\hat{c}_h^k(s,a) \left( 1 - e^{-\eta \sum_{j:j+d^j=k+d^k} \hat{c}_h^j(s,a)} \right)$$

$$\le \eta \sum_{k,h,s,a} q_h^{k+d^k}(s,a)\hat{c}_h^k(s,a) \left( \sum_{j:j+d^j=k+d^k} \hat{c}_h^j(s,a) \right) \qquad (1 - e^{-x} \le x)$$

$$= \eta \sum_{k,h,s,a} q_h^{k+d^k}(s,a) \frac{\mathbb{I}\{s_h^k = s, a_h^k = a\}c_h^k(s,a)}{\max\{q_h^k(s,a), q_h^{k+d^k}(s,a)\} + \gamma} \left( \sum_{j:j+d^j=k+d^k} \hat{c}_h^j(s,a) \right)$$

$$\le \eta \sum_{k,h,s,a} \sum_{j:j+d^j=k+d^k} \hat{c}_h^j(s,a) = \eta \sum_{k,h,s,a} \sum_{j} \mathbb{I}\{j + d^j = k + d^k\}\hat{c}_h^j(s,a)$$

$$= \eta \sum_{j,h,s,a} \hat{c}_h^j(s,a) \sum_{k} \mathbb{I}\{j + d^j = k + d^k\} \le \eta \sum_{k,h,s,a} |\mathcal{F}^{k+d^k}|\hat{c}_h^k(s,a).$$

Finally, by event $E^d$,

$$\sum_{k,h,s,a} |\mathcal{F}^{k+d^k}|\hat{c}_h^k(s,a) = O\left( \sum_{k,h,s,a} |\mathcal{F}^{k+d^k}|c_h^k(s,a) + \frac{Hd_{max}\log\frac{H}{\delta}}{\gamma} \right) = O\left( \eta HSAK + \frac{Hd_{max}\log\frac{H}{\delta}}{\gamma} \right). \qquad \square$$

### C.4 Bound on the Delay-caused Drift (DRIFT in Eq. (29))

**Lemma C.4** (DRIFT term). *Under the good event,*

$$\sum_{k=1}^{K} \langle q^k - q^{k+d^k}, \hat{c}^k \rangle = O\left( \eta\sqrt{H^3SA}(D + K) + \frac{\eta}{\gamma}H^{3/2}d_{max}\log\frac{H}{\delta} + \frac{H\log\frac{H}{\delta}}{\gamma} \right).$$

*Proof.* By event $E^{\hat{c}}$ we have:

$$\sum_{k=1}^{K} \langle \hat{c}^k, q^k - q^{k+d^k} \rangle \le \sum_{k=1}^{K} \langle \hat{c}^k, |q^k - q^{k+d^k}| \rangle = O\left( \sum_{k=1}^{K} \langle c^k, |q^k - q^{k+d^k}| \rangle + \frac{H\log\frac{H}{\delta}}{\gamma} \right).$$

Now, by Pinsker inequality and Jensen inequality:

$$\sum_{k=1}^{K} \langle c^k, |q^k - q^{k+d^k}| \rangle \le \sum_{k=1}^{K} \sum_{j=k}^{k+d^k-1} \sum_{h,s,a} |q_h^j(s,a) - q_h^{j+1}(s,a)| = \sum_{k=1}^{K} \sum_{j=k}^{k+d^k-1} \sum_{h} \|q_h^j - q_h^{j+1}\|_1$$

$$\le \sum_{k=1}^{K} \sum_{j=k}^{k+d^k-1} \sum_{h} \sqrt{2\mathrm{KL}(q_h^j \| q_h^{j+1})} \le \sum_{k=1}^{K} \sum_{j=k}^{k+d^k-1} \sqrt{2H \sum_{h} \mathrm{KL}(q_h^j \| q_h^{j+1})}$$

$$\le \sum_{k=1}^{K} \sum_{j=k}^{k+d^k-1} \sqrt{H \sum_{h} \sum_{s,a} q_h^j(s,a) \Big( \eta \sum_{i:i+d^i=j} \hat{c}_h^i(s,a) \Big)^2}$$

$$\le \eta\sqrt{H} \sum_{k=1}^{K} \sum_{j=k}^{k+d^k-1} \sum_{i:i+d^i=j} \sum_{h,s,a} \sqrt{q_h^j(s,a)} \hat{c}_h^i(s,a),$$

where the last inequality is by $\|x\|_2 \le \|x\|_1$, and the one before is by Lemma C.6. Finally, we rearrange as follows:

$$\sum_{k=1}^{K} \sum_{j=k}^{k+d^k-1} \sum_{i:i+d^i=j} \sum_{h,s,a} \sqrt{q_h^j(s,a)} \hat{c}_h^i(s,a) = \sum_{k,j,i} \mathbb{I}\{k \le j < k+d^k, i+d^i=j\} \sum_{h,s,a} \sqrt{q_h^j(s,a)} \hat{c}_h^i(s,a)$$

$$= \sum_{k,j,i} \mathbb{I}\{k \le j < k+d^k, i+d^i=j\} \sum_{h,s,a} \sqrt{q_h^{i+d^i}(s,a)} \hat{c}_h^i(s,a)$$

$$= \sum_{k,i} \mathbb{I}\{k \le i+d^i < k+d^k\} \sum_{h,s,a} \sqrt{q_h^{i+d^i}(s,a)} \hat{c}_h^i(s,a)$$

$$= O\left( \sum_{k,i} \mathbb{I}\{k \le i+d^i < k+d^k\} \sum_{h,s,a} \sqrt{q_h^{i+d^i}(s,a)} c_h^i(s,a) + \frac{Hd_{max} \log \frac{H}{\delta}}{\gamma} \right),$$

where the last relation is by event $E^{sq}$. To finish the proof we use Lemma C.7:

$$\sum_{k,i} \mathbb{I}\{k \le i+d^i < k+d^k\} \sum_{h,s,a} \sqrt{q_h^{i+d^i}(s,a)} c_h^i(s,a) \le \sqrt{HSA} \sum_{k,i} \mathbb{I}\{k \le i+d^i < k+d^k\} \sqrt{\sum_{h,s,a} q_h^{i+d^i}(s,a)}$$

$$= H\sqrt{SA} \sum_{k,i} \mathbb{I}\{k \le i+d^i < k+d^k\} \le H\sqrt{SA}(D+K). \qquad \square$$

## C.5  Bound on the Bias of the Delay-adapted Estimator ($\mathrm{BIAS}_1$ in Eq. (29))

**Lemma C.5** ($\mathrm{BIAS}_1$). *Under the good event,*

$$\sum_{k=1}^{K} \langle c^k - \hat{c}^k, q^k \rangle = O\left( H\sqrt{K \log \frac{1}{\delta}} + \gamma HSAK + \eta\sqrt{H^3 SA}(D+K) + \frac{\eta}{\gamma} H^{3/2} d_{max} \log \frac{H}{\delta} \right).$$

*Proof.* Decompose $\mathrm{BIAS}_1$ as follows:

$$\sum_{k=1}^{K} \langle c^k - \hat{c}^k, q^k \rangle = \sum_{k=1}^{K} \langle c^k - \mathbb{E}\left[ \hat{c}^k \mid \tilde{\mathcal{H}}^{k+d^k} \right], q^k \rangle + \sum_{k=1}^{K} \langle \mathbb{E}\left[ \hat{c}^k \mid \tilde{\mathcal{H}}^{k+d^k} \right] - \hat{c}^k, q^k \rangle.$$

The second term is bounded by $O(H\sqrt{K\log\frac{1}{\delta}})$ under event $E^c$. The first term is bounded as follows:

$$\sum_{k=1}^{K}\langle c^k - \mathbb{E}[\hat{c}^k \mid \tilde{\mathcal{H}}^{k+d^k}], q^k\rangle = \sum_{k,h,s,a} q_h^k(s,a)c_h^k(s,a)\left(1 - \frac{\mathbb{E}\left[\mathbb{I}\{s_h^k = s, a_h^k = a\} \mid \tilde{\mathcal{H}}^{k+d^k}\right]}{\max\{q_h^{k+d^k}(s,a), q_h^k(s,a)\} + \gamma}\right)$$

$$= \sum_{k,h,s,a} q_h^k(s,a)c_h^k(s,a)\left(1 - \frac{q_h^k(s,a)}{\max\{q_h^{k+d^k}(s,a), q_h^k(s,a)\} + \gamma}\right)$$

$$= \sum_{k,h,s,a} \frac{q_h^k(s,a)}{\max\{q_h^{k+d^k}(s,a), q_h^k(s,a)\} + \gamma}(\max\{q_h^{k+d^k}(s,a), q_h^k(s,a)\} - q_h^k(s,a) + \gamma)$$

$$\leq \sum_{k,h,s,a} (\max\{q_h^{k+d^k}(s,a), q_h^k(s,a)\} - q_h^k(s,a)) + \gamma HSAK$$

$$\leq \sum_{k,h,s,a} |q_h^{k+d^k}(s,a) - q_h^k(s,a)| + \gamma HSAK$$

$$\leq \eta\sqrt{H^3SA}(D+K) + \frac{\eta}{\gamma}H^{3/2}d_{\max} + \gamma HSAK.$$

where the first equality uses the fact that $q^k$ and $q^{k+d^k}$ are determined by the history $\tilde{\mathcal{H}}^{k+d^k}$, the second equality is since the $k$-th episode is not part of the history $\tilde{\mathcal{H}}^{k+d^k}$ as $k \notin \{j : j + d^j < k + d^k\}$, and the last inequality is as in the proof of Lemma C.4. $\qquad\square$

### C.6 Auxiliary lemmas

**Lemma C.6.** $\sum_h KL(q_h^k \parallel q_h^{k+1}) \leq \frac{\eta^2}{2}\sum_{h,s,a} q_h^k(s,a)(\sum_{j:j+d^j=k} \hat{c}_h^j(s,a))^2$.

*Proof.* We start with expanding $\mathrm{KL}(q_h^k \parallel q_h^{k+1})$ as follows:

$$\sum_h \mathrm{KL}(q_h^k \parallel q_h^{k+1}) = \sum_{h,s,a} q_h^k(s,a)\log\frac{q_h^k(s,a)}{q_h^{k+1}(s,a)} = \sum_{h,s,a} q_h^k(s,a)\log\frac{Z_h^k(v^k)q_h^k(s,a)}{q_h^k(s,a)e^{B_h^k(s,a|v^k)}}$$

$$= \sum_{h,s,a} q_h^k(s,a)\log Z_h^k(v^k) - \sum_{h,s,a} q_h^k(s,a)B_h^k(s,a \mid v^k)$$

$$= \sum_h \log Z_h^k(v^k) - \sum_{h,s,a} q_h^k(s,a)B_h^k(s,a \mid v^k). \tag{30}$$

For the first term in Eq. (30), by definition of $v^k$ and $Z_h^k$:

$$\sum_h \log Z_h^k(v^k) \leq \sum_h \log Z_h^k(0) = \sum_h \log\left(\sum_{s,a} q_h^k(s,a)e^{B_h^k(s,a|0)}\right) = \sum_h \log\left(\sum_{s,a} q_h^k(s,a)e^{-\eta\sum_{j:j+d^j=k}\hat{c}_h^j(s,a)}\right)$$

$$\leq \sum_h \log\left(\sum_{s,a} q_h^k(s,a)\left(1 - \eta\sum_{j:j+d^j=k}\hat{c}_h^j(s,a) + \frac{1}{2}\left(\eta\sum_{j:j+d^j=k}\hat{c}_h^j(s,a)\right)^2\right)\right)$$

$$= \sum_h \log\left(1 - \eta\sum_{s,a}\sum_{j:j+d^j=k} q_h^k(s,a)\hat{c}_h^j(s,a) + \frac{\eta^2}{2}\sum_{s,a} q_h^k(s,a)\left(\sum_{j:j+d^j=k}\hat{c}_h^j(s,a)\right)^2\right)$$

$$\leq \sum_h\left(-\eta\sum_{s,a}\sum_{j:j+d^j=k} q_h^k(s,a)\hat{c}_h^j(s,a) + \frac{\eta^2}{2}\sum_{s,a} q_h^k(s,a)\left(\sum_{j:j+d^j=k}\hat{c}_h^j(s,a)\right)^2\right)$$

$$= -\eta\sum_{h,s,a}\sum_{j:j+d^j=k} q_h^k(s,a)\hat{c}_h^j(s,a) + \frac{\eta^2}{2}\sum_{h,s,a} q_h^k(s,a)\left(\sum_{j:j+d^j=k}\hat{c}_h^j(s,a)\right)^2,$$

where the second inequality is by $e^s \le 1 + s + s^2/2$ for $s \le 0$, and the third inequality is by $\log(1 + s) \le s$ for all $s$. The second term in Eq. (30) can be written as follows:

$$\sum_{h,s,a} q_h^k(s,a)B_h^k(s,a \mid v^k) = \sum_{h,s,a} q_h^k(s,a)v_h^k(s) - \eta \sum_{h,s,a} \sum_{j:j+d^j=k} q_h^k(s,a)\hat{c}_h^j(s,a)$$
$$- \sum_{h,s,a,s'} q_h^k(s,a)p_h(s' \mid s,a)v_{h+1}^k(s').$$

So now, by occupancy measure constraints:

$$\sum_{h,s,a,s'} q_h^k(s,a)p_h(s' \mid s,a)v_{h+1}^k(s') = \sum_{h,s'} v_{h+1}^k(s') \sum_{s,a} q_h^k(s,a)p_h(s' \mid s,a) = \sum_{h,s',a'} q_{h+1}^k(s',a')v_{h+1}^k(s'),$$

which forms a telescopic sum, so by $v_0^k(s) = v_{H+1}^k(s) = 0$, we have:

$$\sum_{h,s,a} q_h^k(s,a)B_h^k(s,a \mid v^k) = -\eta \sum_{h,s,a} \sum_{j:j+d^j=k} q_h^k(s,a)\hat{c}_h^j(s,a). \qquad \square$$

**Lemma C.7** ([41]). $\sum_{k=1}^{K} \sum_{i=1}^{K} \mathbb{I}\{k \le i + d^i < k + d^k\} \le D + K.$

*Proof.*

$$\sum_{k=1}^{K} \sum_{i=1}^{K} \mathbb{I}\{k \le i + d^i < k + d^k\} = \sum_{k=1}^{K} \sum_{i=1}^{K} \mathbb{I}\{k \le i + d^i < k + d^k\}$$

$$= \sum_{k=1}^{K} \sum_{i=1}^{k} \mathbb{I}\{k \le i + d^i < k + d^k\} + \sum_{k=1}^{K} \sum_{i=k+1}^{K} \mathbb{I}\{k \le i + d^i < k + d^k\}$$

$$= \sum_{k=1}^{K} \sum_{i=1}^{k} \mathbb{I}\{k \le i + d^i\} - \sum_{k=1}^{K} \sum_{i=1}^{k} \mathbb{I}\{k \le i + d^i, i + d^i \ge k + d^k\} + \sum_{k=1}^{K} \sum_{i=k+1}^{K} \mathbb{I}\{k \le i + d^i < k + d^k\}$$

$$= \sum_{k=1}^{K} \sum_{i=1}^{K} \mathbb{I}\{i \le k \le i + d^i\} - \sum_{k=1}^{K} \sum_{i=1}^{k} \mathbb{I}\{k + d^k \le i + d^i\} + \sum_{k=1}^{K} \sum_{i=1}^{K} \mathbb{I}\{i \ge k + 1, k \le i + d^i < k + d^k\}$$

$$= \sum_{i=1}^{K} \underbrace{\sum_{k=1}^{K} \mathbb{I}\{i \le k \le i + d^i\}}_{=d^i+1} - \sum_{k=1}^{K} \sum_{i=1}^{K} \mathbb{I}\{i \le k, k + d^k \le i + d^i\} + \sum_{k=1}^{K} \sum_{i=1}^{K} \mathbb{I}\{i \ge k + 1, k \le i + d^i < k + d^k\}$$

$$\le D + K - \sum_{k=1}^{K} \sum_{i=1}^{K} \mathbb{I}\{i \le k, k + d^k \le i + d^i\} + \sum_{k=1}^{K} \sum_{i=1}^{K} \mathbb{I}\{k \le i, i + d^i \le k + d^k\} \le D + K. \qquad \square$$

**Algorithm 8** Delayed UOB-REPS with delay-adapted estimator

---

**Input:** State space $\mathcal{S}$, Action space $\mathcal{A}$, Horizon $H$, Number of episodes $K$, Learning rate $\eta > 0$, Exploration parameter $\gamma > 0$, Confidence parameter $\delta > 0$.

**Initialization:** Set $\pi_h^1(a \mid s) = \frac{1}{A}$, $q_h^1(s,a,s') = \frac{1}{S^2 A}$, $m_h^1(s,a) = 0$, $m_h^1(s,a,s')$ for every $(s,a,s',h) \in \mathcal{S} \times \mathcal{A} \times \mathcal{S} \times [H]$.

**for** $k = 1,2,...,K$ **do**

    Play episode $k$ with policy $\pi^k$ and observe delayed trajectory feedback $\{(s_h^j, a_h^j)\}_{h=1}^H$ for all $j$ such that $j + d^j = k$

    Update confidence set $\mathcal{P}^{k+1}$ by Algorithm 9.

    **for** $j : j + d^j = k$ **do**

        Observe feedback $\{c_h^j(s_h^j, a_h^j)\}_{h=1}^H$.

        Compute $u_h^j(s,a) = \max_{p' \in \mathcal{P}^j} q_h^{p',\pi^j}(s,a)$ and $u_h^k(s,a) = \max_{p' \in \mathcal{P}^k} q_h^{p',\pi^k}(s,a)$.

        Compute loss estimator $\hat{c}_h^j(s,a) = \frac{c_h^j(s,a)\mathbb{I}\{s_h^j=s, a_h^j=a\}}{\max\{u_h^j(s,a), u_h^k(s,a)\} + \gamma}$ for every $(s,a,h) \in \mathcal{S} \times \mathcal{A} \times [H]$.

    **end for**

    Update occupancy measure:

$$q^{k+1} = \arg \min_{q \in \Delta(\mathcal{M}, k+1)} \eta \left\langle q, \sum_{j:j+d^j=k} \hat{c}^j \right\rangle + \mathrm{KL}(q \parallel q^k), \tag{31}$$

    where $\mathrm{KL}(q \parallel q') = \sum_{h,s,a,s'} q_h(s,a,s') \ln \frac{q_h(s,a,s')}{q_h'(s,a,s')} + q_h'(s,a,s') - q_h(s,a,s')$ and $\Delta(\mathcal{M}, k+1) = \{q^{\pi,p'} \mid \pi \in (\Delta_\mathcal{A})^{\mathcal{S} \times [H]}, p' \in \mathcal{P}^{k+1}\}$.

    Update policy: $\pi_h^{k+1}(a \mid s) = \frac{\sum_{s'} q_h^{k+1}(s,a,s')}{\sum_{a'} \sum_{s'} q_h^{k+1}(s,a',s')}$ for every $(s,a,h) \in \mathcal{S} \times \mathcal{A} \times [H]$.

**end for**

---

**Algorithm 9** Update confidence set with delayed trajectory feedback

---

**Input:** trajectories $\{(s_h^j, a_h^j)\}_{h \in [H], j:j+d^j=k}$.

Update visit counters: $m_h^{k+1}(s,a) \leftarrow m_h^k(s,a) + \sum_{j:j+d^j=k} \mathbb{I}\{s_h^j = s, a_h^j = a\}$,

$m_h^{k+1}(s,a,s') \leftarrow m_h^k(s,a,s') + \sum_{j:j+d^j=k} \mathbb{I}\{s_h^j = s, a_h^j = a, s_{h+1}^j = s'\}$ for every $h, s, s'$ and $a$.

Compute empirical transitions function $\bar{p}^{k+1}$: $\bar{p}_h^{k+1}(s' \mid s,a) = \frac{m_h^{k+1}(s,a,s')}{m_h^{k+1}(s,a) \vee 1}$     $\forall (s,a,s',h)$.

Define confidence sets $\mathcal{P}^{k+1}$ such that $p' \in \mathcal{P}^{k+1}$ if and only if, for every $(s,a,s',h)$, $p'$ ensures $\sum_{s'} p_h'(s'|s,a) = 1$ and:

$$\left| p_h'(s'|s,a) - \bar{p}_h^{k+1}(s'|s,a) \right| \leq \sqrt{\frac{16 \bar{p}_h^{k+1}(s'|s,a) \log \frac{10HSAK}{\delta}}{m_h^{k+1}(s,a) \vee 1}} + \frac{10 \log \frac{10HSAK}{\delta}}{m_h^{k+1}(s,a) \vee 1}.$$

---

## D Delayed UOB-REPS with delay-adapted estimator

**Remark D.1.** *Note that the confidence set at time $k$ in Algorithm 8 is constructed using only the trajectories from rounds $j$ such that $j + d^j < k$ (a.k.a delayed trajectory feedback [28]). The main reason for that is that our analysis requires that $\pi^k$ would be completely determined by the history from rounds $j$ such that $j + d^j < k$. This is specifically crucial for the analysis of $\mathrm{BIAS}_1$ (see Lemma C.5) and in some of the concentration bounds. This means that our algorithm performs under the weaker assumption of delayed trajectory feedback, but this also comes at the price of an additional additive term in the regret of order $H^3 S^2 A d_{max}$. In order to eliminate the dependency in $d_{max}$ one can use the skipping technique of [41]. In this case the regret would scale as $\tilde{O}(H^2 S \sqrt{AD})$, under the worst case.*

Explicitly solving this optimization problem in Eq. (31), we get [36]:

$$q_h^{k+1}(s,a,s') = \frac{q_h^k(s,a,s') e^{B_h^k(s,a,s' | v^{\mu^k}, e^{\mu^k}, \beta^k)}}{Z_h^k(v^{\mu^k}, e^{\mu^k}, \beta^k)},$$

for:

$$B_h^k(s, a, s' \mid v, e) = e_h(s, a, s') + v_h(s, a, s') - \eta \sum_{j: j + d^j = k} \hat{c}_h^j(s, a) - \sum_{s''} \bar{p}_h^k(s'' \mid s, a) v_{h+1}(s, a, s'')$$

$$v_h^\mu(s, a, s') = \mu_h^-(s, a, s') - \mu_h^+(s, a, s')$$

$$e_h^{\mu, \beta}(s, a, s') = \beta_{h+1}(s') - \beta_h(s) + \sum_{s''} (\mu_h^-(s, a, s'') + \mu_h^+(s, a, s'')) r_h^k(s'' \mid s, a)$$

$$r_h^k(s' \mid s, a) = \sqrt{\frac{16 \bar{p}_h^k(s' \mid s, a) \log \frac{10 HSAK}{\delta}}{m_h^k(s, a) \vee 1}} + \frac{10 \log \frac{10 HSAK}{\delta}}{m_h^k(s, a) \vee 1}$$

$$Z_h^k(v, e) = \sum_{s, a, s'} q_h^k(s, a, s') e^{B_h^k(s, a, s' \mid v, e)}$$

$$\mu^k, \beta^k = \arg \min_{\beta, \mu \geq 0} \sum_{h=1}^H \log Z_h^k(v^\mu, e^{\mu, \beta}).$$

**Theorem D.2.** *Running UOB-REPS with the delay-adapted estimator,* $\eta = \gamma = \min\{\sqrt{\frac{\log \frac{KHSA}{\delta}}{SAK}}, \sqrt{\frac{\log \frac{KHSA}{\delta}}{\sqrt{HSAD}}}\}$ *guarantees, with probability* $1 - \delta$,

$$R_K = O\left( H^2 S \sqrt{AK \log \frac{KHSA}{\delta}} + (HSA)^{1/4} \cdot H \sqrt{D \log \frac{KHSA}{\delta}} \right.$$
$$\left. + H^3 S^2 A d_{max} \log \frac{KHSA}{\delta} + H^3 S^3 A \log^3 \frac{KHSA}{\delta} \right).$$

## D.1 The good event

Let $\tilde{\mathcal{H}}^k$ be the history of episodes $\{j : j + d^j < k\}$, $\epsilon_h^k(s' \mid s, a) = 16\sqrt{\frac{p_h(s'|s,a)\iota}{n_h^k(s,a)\vee 1}} + \frac{200\iota}{n_h^k(s,a)\vee 1}$ and $\iota = \log\frac{HSAK}{\delta}$. Define the following events:

$$E^p = \left\{ \forall k, s', s, a, h : \left| p_h(s' \mid s, a) - \bar{p}_h^k(s' \mid s, a) \right| \leq 4\sqrt{\frac{\bar{p}_h^k(s' \mid s, a)\log\frac{10HSAK}{\delta}}{m_h^k(s,a) \vee 1}} + 10\frac{\log\frac{10HSAK}{\delta}}{m_h^k(s,a) \vee 1} \right\}$$

$$E^{on1} = \left\{ \sum_{k,h,s,a} \left( q_h^{\pi^k}(s,a) - \mathbb{I}\{s_h^{k,v} = s, a_h^{k,v} = a\} \right) \min\{2, \epsilon_h^k(s,a)\} \leq 10\sqrt{K \log\frac{30KHSA}{\delta}} \right\}$$

$$E^{on2} = \left\{ \sum_{k,h,s,a} q_h^{\pi^k}(s,a)\epsilon_h^k(s,a) \leq 2 \sum_{k,h,s,a} \mathbb{I}\{s_h^{k,v} = s, a_h^{k,v} = a\}\epsilon_h^k(s,a) + 100HS \log^2\frac{30KHSA}{\delta} \right\}$$

$$E^{on3} = \left\{ \sum_{k,s,a,h} \frac{q_h^{\pi^k}(s,a)}{n_h^k(s,a)} \leq 2 \sum_{k,s,a,h} \frac{\mathbb{I}\{s_h^{k,v} = s, a_h^{k,v} = a\}}{n_h^k(s,a)} + H \log\frac{m}{\delta} \right\}$$

$$E^c = \left\{ \sum_{k=1}^K \langle \mathbb{E}[\hat{c}^k \mid \tilde{\mathcal{H}}^{k+d^k}] - \hat{c}^k, q^k \rangle \leq 4H\sqrt{K \log\frac{10}{\delta}} \right\}$$

$$E^{\hat{c}} = \left\{ \sum_{k=1}^K \langle |q^k - q^{k+d^k}|, \hat{c}^k \rangle \leq 4 \sum_{k=1}^K \langle |q^k - q^{k+d^k}|, c^k \rangle + \frac{40H \log\frac{10H}{\delta}}{\gamma} \right\}$$

$$E^d = \left\{ \sum_{k,h,s,a} |\mathcal{F}^{k+d^k}|\hat{c}_h^k(s,a) \leq \sum_{k,h,s,a} |\mathcal{F}^{k+d^k}|c_h^k(s,a) + \frac{10Hd_{max} \log\frac{10H}{\delta}}{\gamma} \right\}$$

$$E^{sq} = \left\{ \sum_{k=1}^K \sum_{i=1}^K \mathbb{I}\{k \leq i + d^i < k + d^k\} \sum_{h,s,a} \sqrt{q_h^{i+d^i}(s,a)}(\hat{c}_h^i(s,a) - 4c_h^i(s,a)) \leq \frac{10Hd_{max} \log\frac{10H}{\delta}}{\gamma} \right\}$$

$$E^\star = \left\{ \sum_{k=1}^K \langle \hat{c}^k - c^k, q^\star \rangle \leq \frac{H \log\frac{10HSA}{\delta}}{\gamma} \right\}$$

The good event is the intersection of the above events. The following lemma establishes that the good event holds with high probability.

**Lemma D.3** (The Good Event). *Let* $\mathbb{G} = E^p \cap E^{on1} \cap E^{on2} \cap E^{on3} \cap E^c \cap E^{\hat{c}} \cap E^d \cap E^{sq} \cap E^\star$ *be the good event. It holds that* $\Pr[\mathbb{G}] \geq 1 - \delta$.

*Proof.* Similar to the proof of Lemma C.2. Events $E^p$, $E^{on1}$, $E^{on2}$ and $E^{on3}$ are standard (see, e.g., [22, 27]). ☐

## D.2 Proof of the Main Theorem

*Proof of Theorem D.2.* By Lemma D.3, the good event holds with probability $1 - \delta$. We now analyze the regret under the assumption that the good event holds. We decompose the regret as follows:

$$R_K = \sum_{k=1}^K \langle q^{\pi^k} - q, c^k \rangle$$

$$= \underbrace{\sum_{k=1}^K \langle q^{\pi^k} - q^k, c^k \rangle}_{\text{EST}} + \underbrace{\sum_{k=1}^K \langle q^k, c^k - \hat{c}^k \rangle}_{\text{BIAS}_1} + \underbrace{\sum_{k=1}^K \langle q^\star, \hat{c}^k - c^k \rangle}_{\text{BIAS}_2} + \underbrace{\sum_{k=1}^K \langle q^k - q^{k+d^k}, \hat{c}^k \rangle}_{\text{DRIFT}} + \underbrace{\sum_{k=1}^K \langle q^{k+d^k} - q^\star, \hat{c}^k \rangle}_{\text{REG}}. \quad (32)$$

BIAS$_2$ is bounded under event $E^\star$ by $O(\frac{H\iota}{\gamma})$, EST is bounded in Lemma D.4 by $O(H^2S\sqrt{AK\iota} + H^2S^2A\iota^2 + H^2SAd_{max})$, REG is bounded in Lemma D.5 by $O(\frac{H\iota}{\eta} + \eta HSAK + \frac{\eta}{\gamma}Hd_{max}\iota)$, DRIFT is bounded in Lemma D.6 by

$O(\eta\sqrt{H^3SA}(D+K) + \frac{\eta}{\gamma}H^{3/2}d_{max}\iota + \frac{H\iota}{\gamma})$, and $\textsc{Bias}_1$ is bounded in Lemma D.7 by $O(H^2S\sqrt{AK\iota} + H^3S^3A\iota^3 + \gamma HSAK + \eta\sqrt{H^3SA}(D+K) + \frac{\eta}{\gamma}H^{3/2}d_{max}\iota) + H^3S^2Ad_{max}$. Putting everything together:

$$R_K = O\Big(H^2S\sqrt{AK\iota} + H^3S^3A\iota^3 + (\eta+\gamma)HSAK$$
$$+ (\frac{1}{\eta} + \frac{1}{\gamma})H\iota + \eta\sqrt{H^3SA}(D+K) + \frac{\eta}{\gamma}H^{3/2}d_{max}\iota + H^3S^2Ad_{max}\Big),$$

and plugging in the definitions of $\eta$ and $\gamma$ finishes the proof. $\square$

### D.3 Bound on the Transition Estimation Error (EST in Eq. (32))

**Lemma D.4** (EST Term). *Under the good event,*

$$\sum_{k=1}^{K}\langle q^{\pi^k} - q^k, c^k\rangle = O\left(H^2S\sqrt{AK\iota} + H^2S^2A\iota^2 + H^2SAd_{max}\right).$$

*Proof.* Let $q^k = q^{\pi^k,p^k}$. By the value difference lemma [40]:

$$\sum_{k=1}^{K}\langle q^{\pi^k} - q^k, c^k\rangle = \sum_{k,h,s,a} q_h^{\pi^k}(s,a) \sum_{s'}\left(p_h^k(s'\mid s,a) - p_h(s'\mid s,a)\right) V_{h+1}^{\pi^k,p^k}(s')$$

$$\leq H\sum_{k,h,s,a} q_h^{\pi^k}(s,a)\|p_h^k(\cdot\mid s,a) - p_h(\cdot\mid s,a)\|_1$$

$$= O(H^2S\sqrt{AK\iota} + H^2S^2A\iota^2 + H^2SAd_{max}),$$

where the second inequality is by event $E^p$ and the last is by [28, lemma 5]. $\square$

### D.4 Bound on the Regret with respect to the Loss Estimators and Future Policies (REG in Eq. (32))

**Lemma D.5** (REG Term). *Under the good event,*

$$\sum_{k=1}^{K}\langle q^{k+d^k} - q^\star, \hat{c}^k\rangle = O\left(\frac{H\iota}{\eta} + \eta HSAK + \frac{\eta}{\gamma}Hd_{max}\iota\right).$$

*Proof.* Let $\tilde{q}_h^{k+1}(s,a,s') = q_h^k(s,a,s')e^{-\eta\sum_{j:j+d^j=k}\hat{c}_h^j(s,a)}$. Taking the log,

$$\eta\sum_{j:j+d^j=k}\hat{c}_h^j(s,a) = \log q_h^k(s,a,s') - \log\tilde{q}_h^{k+1}(s,a,s').$$

Hence,

$$\eta\left\langle\sum_{j:j+d^j=k}\hat{c}_h^j, q^k - q\right\rangle = \langle\log q^k - \log\tilde{q}^{k+1}, q^k - q^\star\rangle = \mathrm{KL}(q^\star\|q^k) - \mathrm{KL}(q^\star\|\tilde{q}^{k+1}) + \mathrm{KL}(q^k\|\tilde{q}^{k+1})$$

$$\leq \mathrm{KL}(q^\star\|q^k) - \mathrm{KL}(q^\star\|q^{k+1}) - \mathrm{KL}(q^{k+1}\|\tilde{q}^{k+1}) + \mathrm{KL}(q^k\|\tilde{q}^{k+1})$$

$$\leq \mathrm{KL}(q^\star\|q^k) - \mathrm{KL}(q^\star\|q^{k+1}) + \mathrm{KL}(q^k\|\tilde{q}^{k+1}),$$

where the second equality follows directly the definition of KL, the first inequality is by [50, Lemma 1.2], and the second inequality is since the KL is non-negative. Now, the last term is bounded as follows:

$$\mathrm{KL}(q^k\|\tilde{q}^{k+1}) \leq \mathrm{KL}(q^k\|\tilde{q}^{k+1}) + \mathrm{KL}(\tilde{q}^{k+1}\|q^k)$$

$$= \sum_h\sum_{s,a,s'}\tilde{q}_h^{k+1}(s,a,s')\log\frac{\tilde{q}_h^{k+1}(s,a,s')}{q_h^k(s,a,s')} + \sum_h\sum_{s,a,s'}q_h^k(s,a,s')\log\frac{q_h^k(s,a,s')}{\tilde{q}_h^{k+1}(s,a,s')}$$

$$= \langle q^k - \tilde{q}^{k+1}, \log q^k - \log\tilde{q}^{k+1}\rangle = \eta\left\langle q^k - \tilde{q}^{k+1}, \sum_{j:j+d^j=k}\hat{c}^j\right\rangle.$$

We get that

$$\eta \left\langle \sum_{j:j+d^j=k} \hat{c}^j, q^k - q^\star \right\rangle \le \mathrm{KL}(q^\star \parallel q^k) - \mathrm{KL}(q^\star \parallel q^{k+1}) + \eta \left\langle q^k - \tilde{q}^{k+1}, \sum_{j:j+d^j=k} \hat{c}^j \right\rangle.$$

Summing over $k$ and dividing by $\eta$, we get

$$\underbrace{\sum_{k=1}^{K} \sum_{j:j+d^j=k} \langle \hat{c}^j, q^k - q^\star \rangle}_{(*)} \le \frac{\mathrm{KL}(q^\star \parallel q^1) - \mathrm{KL}(q^\star \parallel q^{K+1})}{\eta} + \sum_{k=1}^{K} \left\langle q^k - \tilde{q}^{k+1}, \sum_{j:j+d^j=k} \hat{c}^j \right\rangle$$

$$\le \frac{\mathrm{KL}(q^\star \parallel q^1)}{\eta} + \sum_{k=1}^{K} \left\langle q^k - \tilde{q}^{k+1}, \sum_{j:j+d^j=k} \hat{c}^j \right\rangle$$

$$\le \frac{4H \log(SA)}{\eta} + \underbrace{\sum_{k=1}^{K} \left\langle q^k - \tilde{q}^{k+1}, \sum_{j:j+d^j=k} \hat{c}^j \right\rangle}_{(**)},$$

where the last inequality is a standard argument (see [50, 16]). We now rearrange $(*)$ and $(**)$:

$$(*) = \sum_{k=1}^{K} \sum_{j=1}^{K} \mathbb{I}\{j + d^j = k\} \langle \hat{c}^j, q^k - q^\star \rangle = \sum_{j=1}^{K} \sum_{k=1}^{K} \mathbb{I}\{j + d^j = k\} \langle \hat{c}^j, q^k - q^\star \rangle$$

$$= \sum_{j=1}^{K} \langle \hat{c}^j, q^{j+d^j} - q^\star \rangle = \sum_{k=1}^{K} \langle \hat{c}^k, q^{k+d^k} - q^\star \rangle.$$

In a similar way,

$$(**) = \sum_{k=1}^{K} \sum_{j:j+d^j=k} \langle q^k - \tilde{q}^{k+1}, \hat{c}^j \rangle = \sum_{k=1}^{K} \sum_{j=1}^{K} \mathbb{I}\{j + d^j = k\} \langle q^k - \tilde{q}^{k+1}, \hat{c}^j \rangle$$

$$= \sum_{j=1}^{K} \sum_{k=1}^{K} \mathbb{I}\{j + d^j = k\} \langle q^k - \tilde{q}^{k+1}, \hat{c}^j \rangle = \sum_{k=1}^{K} \langle q^{k+d^k} - \tilde{q}^{k+d^k+1}, \hat{c}^k \rangle.$$

This gives us,

$$\sum_{k=1}^{K} \langle \hat{c}^k, q^{k+d^k} - q^\star \rangle \le \frac{4H \log(SA)}{\eta} + \sum_{k=1}^{K} \langle q^{k+d^k} - \tilde{q}^{k+d^k+1}, \hat{c}^k \rangle.$$

It remains to bound the second term on the right hand side:

$$\sum_k \langle q^{k+d^k} - \tilde{q}^{k+d^k+1}, \hat{c}^k \rangle = \sum_{k,h,s,a,s'} \hat{c}_h^k(s,a)(q_h^{k+d^k}(s,a,s') - \tilde{q}_h^{k+d^k+1}(s,a,s'))$$

$$= \sum_{k,h,s,a,s'} \hat{c}_h^k(s,a) \left( q_h^{k+d^k}(s,a,s') - q_h^{k+d^k}(s,a,s')e^{-\eta \sum_{j:j+d^j=k+d^k} \hat{c}_h^j(s,a)} \right)$$

$$= \sum_{k,h,s,a,s'} q_h^{k+d^k}(s,a,s')\hat{c}_h^k(s,a) \left( 1 - e^{-\eta \sum_{j:j+d^j=k+d^k} \hat{c}_h^j(s,a)} \right)$$

$$\le \eta \sum_{k,h,s,a} q_h^{k+d^k}(s,a)\hat{c}_h^k(s,a) \left( \sum_{j:j+d^j=k+d^k} \hat{c}_h^j(s,a) \right) \qquad (1 - e^{-x} \le x)$$

$$= \eta \sum_{k,h,s,a} q_h^{k+d^k}(s,a) \frac{\mathbb{I}\{s_h^k = s, a_h^k = a\}c_h^k(s,a)}{\max\{u_h^k(s,a), u_h^{k+d^k}(s,a)\} + \gamma} \left( \sum_{j:j+d^j=k+d^k} \hat{c}_h^j(s,a) \right)$$

$$\le \eta \sum_{k,h,s,a} \sum_{j:j+d^j=k+d^k} \hat{c}_h^j(s,a) = \eta \sum_{k,h,s,a} \sum_j \mathbb{I}\{j + d^j = k + d^k\}\hat{c}_h^j(s,a)$$

$$= \eta \sum_{j,h,s,a} \hat{c}_h^j(s,a) \sum_k \mathbb{I}\{j + d^j = k + d^k\} \le \eta \sum_{k,h,s,a} |\mathcal{F}^{k+d^k}|\hat{c}_h^k(s,a),$$

where the second inequality is since $u_h^{k+d^k}(s,a) \ge q_h^{k+d^k}(s,a)$ under the good event. Finally, by event $E^d$,

$$\sum_{k,h,s,a} |\mathcal{F}^{k+d^k}|\hat{c}_h^k(s,a) = O\left( \sum_{k,h,s,a} |\mathcal{F}^{k+d^k}|c_h^k(s,a) + \frac{Hd_{max}\iota}{\gamma} \right) = O\left( \eta HSAK + \frac{Hd_{max}\iota}{\gamma} \right). \qquad \square$$

### D.5 Bound on the Delay-caused Drift (DRIFT in Eq. (32))

**Lemma D.6** (DRIFT term). *Under the good event,*

$$\sum_{k=1}^K \langle q^k - q^{k+d^k}, \hat{c}^k \rangle = O\left( \eta\sqrt{H^3 SA}(D + K) + \frac{\eta}{\gamma}H^{3/2}d_{max}\iota + \frac{H\iota}{\gamma} \right).$$

*Proof.* By event $E^{\hat{c}}$ we have:

$$\sum_{k=1}^K \langle \hat{c}^k, q^k - q^{k+d^k} \rangle \le \sum_{k=1}^K \langle \hat{c}^k, |q^k - q^{k+d^k}| \rangle = O\left( \sum_{k=1}^K \langle c^k, |q^k - q^{k+d^k}| \rangle + \frac{H\iota}{\gamma} \right).$$

Now, by Pinsker inequality and Jensen inequality:

$$\sum_{k=1}^K \langle c^k, |q^k - q^{k+d^k}| \rangle \le \sum_{k=1}^K \sum_{j=k}^{k+d^k-1} \sum_{h,s,a,s'} |q_h^j(s,a,s') - q_h^{j+1}(s,a,s')| = \sum_{k=1}^K \sum_{j=k}^{k+d^k-1} \sum_h \|q_h^j - q_h^{j+1}\|_1$$

$$\le \sum_{k=1}^K \sum_{j=k}^{k+d^k-1} \sum_h \sqrt{2\text{KL}(q_h^j \| q_h^{j+1})} \le \sum_{k=1}^K \sum_{j=k}^{k+d^k-1} \sqrt{2H \sum_h \text{KL}(q_h^j \| q_h^{j+1})}$$

$$\le \sum_{k=1}^K \sum_{j=k}^{k+d^k-1} \sqrt{H \sum_h \sum_{s,a,s'} q_h^j(s,a,s') \left( \eta \sum_{i:i+d^i=j} \hat{c}_h^i(s,a) \right)^2}$$

$$\le \eta\sqrt{H} \sum_{k=1}^K \sum_{j=k}^{k+d^k-1} \sum_{i:i+d^i=j} \sum_{h,s,a} \sqrt{q_h^j(s,a)}\hat{c}_h^i(s,a),$$

where the last inequality is by $\|x\|_2 \leq \|x\|_1$, and the one before is by Lemma D.8. Finally, we rearrange as follows:

$$\sum_{k=1}^{K} \sum_{j=k}^{k+d^k-1} \sum_{i:i+d^i=j} \sum_{h,s,a} \sqrt{q_h^j(s,a)} \hat{c}_h^i(s,a) = \sum_{k,j,i} \mathbb{I}\{k \leq j < k+d^k, i+d^i = j\} \sum_{h,s,a} \sqrt{q_h^j(s,a)} \hat{c}_h^i(s,a)$$

$$= \sum_{k,j,i} \mathbb{I}\{k \leq j < k+d^k, i+d^i = j\} \sum_{h,s,a} \sqrt{q_h^{i+d^i}(s,a)} \hat{c}_h^i(s,a)$$

$$= \sum_{k,i} \mathbb{I}\{k \leq i+d^i < k+d^k\} \sum_{h,s,a} \sqrt{q_h^{i+d^i}(s,a)} \hat{c}_h^i(s,a)$$

$$= O\left( \sum_{k,i} \mathbb{I}\{k \leq i+d^i < k+d^k\} \sum_{h,s,a} \sqrt{q_h^{i+d^i}(s,a)} c_h^i(s,a) + \frac{H d_{max}\iota}{\gamma} \right),$$

where the last relation is by event $E^{sq}$. To finish the proof we use Lemma C.7:

$$\sum_{k,i} \mathbb{I}\{k \leq i+d^i < k+d^k\} \sum_{h,s,a} \sqrt{q_h^{i+d^i}(s,a)} c_h^i(s,a) \leq \sqrt{HSA} \sum_{k,i} \mathbb{I}\{k \leq i+d^i < k+d^k\} \sqrt{\sum_{h,s,a} q_h^{i+d^i}(s,a)}$$

$$= H\sqrt{SA} \sum_{k,i} \mathbb{I}\{k \leq i+d^i < k+d^k\} \leq H\sqrt{SA}(D+K). \qquad \square$$

### D.6 Bound on the Bias of the Delay-adapted Estimator (BIAS$_1$ in Eq. (32))

**Lemma D.7** (BIAS$_1$ Term). *Under the good event,*

$$\sum_{k=1}^{K} \langle c^k - \hat{c}^k, q^k \rangle = O\left( H^2 S\sqrt{AK\iota} + H^3 S^3 A\iota^3 + \gamma HSAK + \eta \sqrt{H^3 SA}(D+K) + \frac{\eta}{\gamma} H^{3/2} d_{max}\iota + H^3 S^2 A d_{max} \right).$$

*Proof.* Decompose BIAS$_1$ as follows:

$$\sum_{k=1}^{K} \langle c^k - \hat{c}^k, q^k \rangle = \sum_{k=1}^{K} \langle c^k - \mathbb{E}\left[\hat{c}^k \mid \tilde{\mathcal{H}}^{k+d^k}\right], q^k \rangle + \sum_{k=1}^{K} \langle \mathbb{E}\left[\hat{c}^k \mid \tilde{\mathcal{H}}^{k+d^k}\right] - \hat{c}^k, q^k \rangle.$$

The second term is bounded by $O(H\sqrt{K\iota})$ under event $E^c$. The first term is bounded as follows:

$$\sum_{k=1}^{K} \langle c^k - \mathbb{E}[\hat{c}^k \mid \tilde{\mathcal{H}}^{k+d^k}], q^k \rangle = \sum_{k,h,s,a,s'} q_h^k(s,a,s') c_h^k(s,a) \left( 1 - \frac{\mathbb{E}\left[\mathbb{I}\{s_h^k = s, a_h^k = a\} \mid \tilde{\mathcal{H}}^{k+d^k}\right]}{\max\{u_h^k(s,a), u_h^{k+d^k}(s,a)\} + \gamma} \right)$$

$$= \sum_{k,h,s,a} q_h^k(s,a) c_h^k(s,a) \left( 1 - \frac{q_h^{\pi^k}(s,a)}{\max\{u_h^k(s,a), u_h^{k+d^k}(s,a)\} + \gamma} \right)$$

$$= \sum_{k,h,s,a} \frac{q_h^k(s,a)}{\max\{u_h^k(s,a), u_h^{k+d^k}(s,a)\} + \gamma} (\max\{u_h^k(s,a), u_h^{k+d^k}(s,a)\} - q_h^{\pi^k}(s,a) + \gamma)$$

$$\leq \sum_{k,h,s,a} (\max\{u_h^k(s,a), u_h^{k+d^k}(s,a)\} - q_h^{\pi^k}(s,a)) + \gamma HSAK$$

$$\leq \sum_{k,h,s,a} |\max\{u_h^k(s,a), u_h^{k+d^k}(s,a)\} - q_h^{\pi^k}(s,a)| + \gamma HSAK.$$

where the first equality uses the fact that $u^k$ and $u^{k+d^k}$ is determined by the history $\tilde{\mathcal{H}}^{k+d^k}$, the second equality is since the $k$-th episode is not part of the history $\tilde{\mathcal{H}}^{k+d^k}$ as $k \notin \{j : j+d^j < k+d^k\}$, and the first inequality is since $u_h^k(s,a) \geq q_h^k(s,a)$ under the good event. Finally, we bound:

$$\sum_{k,h,s,a} |\max\{u_h^k(s,a), u_h^{k+d^k}(s,a)\} - q_h^{\pi^k}(s,a)| \leq \sum_{k,h,s,a} |u_h^k(s,a) - q_h^{\pi^k}(s,a)| + \sum_{k,h,s,a} |u_h^{k+d^k}(s,a) - q_h^{\pi^k}(s,a)|.$$

The first term is bounded in Lemma D.12 by $O(H^2 S \sqrt{AK\iota} + H^3 S^3 A \iota^3 + H^3 S^2 A d_{max})$, and for the second term:

$$\sum_{k,h,s,a} |u_h^{k+d^k}(s,a) - q_h^{\pi^k}(s,a)| \leq \sum_{k,h,s,a} |u_h^{k+d^k}(s,a) - q_h^{\pi^{k+d^k}}(s,a)|$$
$$+ \sum_{k,h,s,a} |q_h^{\pi^{k+d^k}}(s,a) - q_h^{\pi^k}(s,a)|,$$

where again the first term is bounded in Lemma D.12. Finally,

$$\sum_{k,h,s,a} |q_h^{\pi^{k+d^k}}(s,a) - q_h^{\pi^k}(s,a)| \leq \sum_{k,h,s,a} |q_h^{\pi^{k+d^k}}(s,a) - q_h^{k+d^k}(s,a)| + \sum_{k,h,s,a} |q_h^k(s,a) - q_h^{\pi^k}(s,a)|$$
$$+ \sum_{k,h,s,a} |q_h^{k+d^k}(s,a) - q_h^k(s,a)|,$$

where the first two terms are bounded similarly to Lemma D.4 and the last term is bounded similarly to Lemma D.6. $\qquad\square$

### D.7 Auxiliary lemmas

**Lemma D.8.** $\sum_h KL(q_h^k \parallel q_h^{k+1}) \leq \frac{\eta^2}{2} \sum_{h,s,a,s'} q_h^k(s,a,s')(\sum_{j:j+d^j=k} \hat{c}_h^j(s,a))^2.$

*Proof.* We start with expanding $KL(q_h^k \parallel q_h^{k+1})$ as follows:

$$\sum_h KL(q_h^k \parallel q_h^{k+1}) = \sum_h \sum_{s,a,s'} q_h^k(s,a,s') \log \frac{q_h^k(s,a,s')}{q_h^{k+1}(s,a,s')} = \sum_h \sum_{s,a,s'} q_h^k(s,a,s') \log \frac{Z_h^k(v^{\mu^k}, e^{\mu^k,\beta^k})}{e^{B_h^k(s,a,s'|v^{\mu^k}, e^{\mu^k,\beta^k})}}$$

$$= \underbrace{\sum_h \log Z_h^k(v^{\mu^k}, e^{\mu^k,\beta^k})}_{(A)} - \underbrace{\sum_h \sum_{s,a,s'} q_h^k(s,a,s') B_h^k(s,a,s' \mid v^{\mu^k}, e^{\mu^k,\beta^k})}_{(B)}.$$

By definition of $\mu^k, \beta^k$, term $(A)$ can be bounded by

$$(A) \leq \sum_h \log Z_h^k(0,0) = \sum_h \log(\sum_{s,a,s'} q_h^k(s,a,s') e^{B_h^k(s,a,s'|0,0)}) = \sum_h \log(\sum_{s,a,s'} q_h^k(s,a,s') e^{-\eta \sum_{j:j+d^j=k} \hat{c}_h^j(s,a)})$$

$$\leq \sum_h \log \left( \sum_{s,a,s'} q_h^k(s,a,s') \left( 1 - \eta \sum_{j:j+d^j=k} \hat{c}_h^j(s,a) + \frac{(\eta \sum_{j:j+d^j=k} \hat{c}_h^j(s,a))^2}{2} \right) \right)$$

$$= \sum_h \log \left( 1 - \eta \sum_{s,a,s'} \sum_{j:j+d^j=k} q_h^k(s,a,s') \hat{c}_h^j(s,a) + \sum_{s,a,s'} q_h^k(s,a,s') \frac{(\eta \sum_{j:j+d^j=k} \hat{c}_h^j(s,a))^2}{2} \right)$$

$$\leq -\eta \sum_h \sum_{s,a,s'} \sum_{j:j+d^j=k} q_h^k(s,a,s') \hat{c}_h^j(s,a) + \sum_h \sum_{s,a,s'} q_h^k(s,a,s') \frac{(\eta \sum_{j:j+d^j=k} \hat{c}_h^j(s,a))^2}{2},$$

where the second inequality is by $e^s \leq 1 + s + s^2/2$ for $s \leq 0$, and the third inequality is by $\log(1+s) \leq s$ for all $s$. Term $(B)$ can be rewritten as

$$(B) = \sum_h \sum_{s,a,s'} q_h^k(s,a,s')(e_h^{\mu^k,\beta^k}(s,a,s') + v_h^{\mu^k}(s,a,s') - \eta \sum_{j:j+d^j=k} \hat{c}_h^j(s,a) - \sum_{s''} \bar{p}_h^k(s'' \mid s,a) v_{h+1}^{\mu^k}(s,a,s''))$$

$$= \sum_h \sum_{s,a,s'} q_h^k(s,a,s') e_h^{\mu^k,\beta^k}(s,a,s') + \sum_h \sum_{s,a,s'} q_h^k(s,a,s') v_h^{\mu^k}(s,a,s')$$

$$- \eta \sum_h \sum_{s,a,s'} \sum_{j:j+d^j=k} q_h^k(s,a,s') \hat{c}_h^j(s,a) - \sum_h \sum_{s,a,s'} \sum_{s''} q_h^k(s,a,s') \bar{p}_h^k(s'' \mid s,a) v_{h+1}^{\mu^k}(s,a,s'')$$

$$= \sum_h \sum_{s,a,s'} q_h^k(s,a,s') e_h^{\mu^k,\beta^k}(s,a,s') + \sum_h \sum_{s,a,s'} q_h^k(s,a,s') v_h^{\mu^k}(s,a,s')$$

$$- \eta \sum_h \sum_{s,a,s'} \sum_{j:j+d^j=k} q_h^k(s,a,s') \hat{c}_h^j(s,a) - \sum_h \sum_{s,a} \sum_{s''} q_h^k(s,a) \bar{p}_h^k(s'' \mid s,a) v_{h+1}^{\mu^k}(s,a,s'').$$

Notice that:

$$\sum_{h,s,a,s''} q_h^k(s,a)\bar{p}_h^k(s'' \mid s,a)v_{h+1}^{\mu^k}(s,a,s'')$$

$$= \sum_{h,s,a,s''} q_h^k(s,a)p_h^k(s'' \mid s,a)v_{h+1}^{\mu^k}(s,a,s'') + \sum_{h,s,a,s''} q_h^k(s,a)(\bar{p}_h^k(s'' \mid s,a) - p_h^k(s'' \mid s,a))v_{h+1}^{\mu^k}(s,a,s'')$$

$$= \sum_{h,s,a,s''} q_{h+1}^k(s,a,s'')v_{h+1}^{\mu^k}(s,a,s'') + \sum_{h,s,a,s''} q_h^k(s,a)(\bar{p}_h^k(s'' \mid s,a) - p_h^k(s'' \mid s,a))v_{h+1}^{\mu^k}(s,a,s''),$$

and therefore:

$$(B) = \sum_h \sum_{s,a,s'} q_h^k(s,a,s')e_h^{\mu^k,\beta^k}(s,a,s') - \eta \sum_h \sum_{s,a,s'} \sum_{j:j+d^j=k} q_h^k(s,a,s')\hat{c}_h^j(s,a)$$

$$- \sum_{h,s,a,s''} q_h^k(s,a)(\bar{p}_h^k(s'' \mid s,a) - p_h^k(s'' \mid s,a))v_{h+1}^{\mu^k}(s,a,s'').$$

Overall we get:

$$\sum_h \mathrm{KL}(q_h^k \parallel q_h^{k+1}) \le \sum_{h,s,a,s'} q_h^k(s,a,s')\frac{(\eta \sum_{j:j+d^j=k} \hat{c}_h^j(s,a))^2}{2} - \sum_{h,s,a,s'} q_h^k(s,a,s')e_h^{\mu^k,\beta^k}(s,a,s')$$

$$+ \sum_{h,s,a,s'} q_h^k(s,a)(\bar{p}_h^k(s' \mid s,a) - p_h^k(s' \mid s,a))v_{h+1}^{\mu^k}(s,a,s').$$

To finish the proof we show that:

$$\sum_{h,s,a,s'} q_h^k(s,a)(\bar{p}_h^k(s' \mid s,a) - p_h^k(s' \mid s,a))v_{h+1}^{\mu^k}(s,a,s'') \le \sum_{h,s,a,s'} q_h^k(s,a,s')e_h^{\mu^k,\beta^k}(s,a,s').$$

By definition of $v^{\mu^k}$ and $\epsilon^k$, and since $\mu \ge 0$, we have:

$$\sum_{h,s,a,s'} q_h^k(s,a)(\bar{p}_h^k(s' \mid s,a) - p_h^k(s' \mid s,a))v_{h+1}^{\mu^k}(s,a,s'')$$

$$= \sum_{h,s,a,s'} q_h^k(s,a)(\bar{p}_h^k(s' \mid s,a) - p_h^k(s' \mid s,a))(\mu_{h+1}^{k,-}(s,a,s') - \mu_{h+1}^{k,+}(s,a,s'))$$

$$\le \sum_{h,s,a,s'} q_h^k(s,a)|\bar{p}_h^k(s' \mid s,a) - p_h^k(s' \mid s,a)|(\mu_{h+1}^{k,-}(s,a,s') + \mu_{h+1}^{k,+}(s,a,s'))$$

$$\le \sum_{h,s,a,s'} q_h^k(s,a)\epsilon_h^k(s' \mid s,a)(\mu_{h+1}^{k,-}(s,a,s') + \mu_{h+1}^{k,+}(s,a,s')),$$

so to finish the proof it suffices to show that $\sum_{h,s,a,s'} \beta_{h+1}^k(s') - \beta_h^k(s) = 0$. Indeed, this follows as the sum is telescopic and $\beta_{H+1}^k = \beta_0^k = 0$. $\square$

**Lemma D.9** (Lemma 8 of [22]; see also Lemma B.13 of in [11]). *Under the good event we have,*

$$\forall (k,s,a,s',h): \qquad |p_h(s'|s,a) - \hat{p}_h^k(s'|s,a)| \le \tilde{\epsilon}_h^k(s' \mid s,a).$$

*where* $\tilde{\epsilon}_h^k(s' \mid s,a) = 8\sqrt{\frac{p_h(s'|s,a)\iota}{m_h^k(s,a)\vee 1}} + \frac{100\iota}{m_h^k(s,a)\vee 1}$

**Lemma D.10.** *Under the good event we have, for any* $(k,s,a,s',h)$ *such that* $m_h^k(s,a) \ge d_{max}$,

$$|p_h(s'|s,a) - \hat{p}_h^k(s'|s,a)| \le \epsilon_h^k(s' \mid s,a).$$

*where* $\epsilon_h^k(s' \mid s,a) = 16\sqrt{\frac{p_h(s'|s,a)\iota}{n_h^k(s,a)\vee 1}} + \frac{200\iota}{n_h^k(s,a)\vee 1}$.

*Proof.* Note that if $m_h^k(s,a) \ge d_{max}$ then,

$$\frac{1}{m_h^k(s,a)\vee 1} = \frac{1}{n_h^k(s,a)\vee 1}\frac{n_h^k(s,a)\vee 1}{m_h^k(s,a)\vee 1} = \frac{1}{n_h^k(s,a)\vee 1}\left(1 + \frac{n_h^k(s,a)\vee 1 - m_h^k(s,a)\vee 1}{m_h^k(s,a)\vee 1}\right)$$

$$\le \frac{1}{n_h^k(s,a)\vee 1}\left(1 + \frac{d_{max}}{m_h^k(s,a)\vee 1}\right) \le \frac{2}{n_h^k(s,a)\vee 1}. \tag{33}$$

where the first inequality is since $n_h^k(s,a) - m_h^k(s,a) \le d_{max}$. We complete the proof by combining Lemma D.9 with the fact that given Eq. (33) $\tilde{\epsilon}_h^k(s' \mid s,a) \le \epsilon_h^k(s' \mid s,a)$. $\qquad\square$

**Lemma D.11** (Lemma E.4 of [27] adapted to delays; see also Lemma 4 of [22]). *With delayed trajectory feedback, under the good event,*

$$
\sum_{k=1}^{K} \sum_{h,s,a} |u_h^k(s,a) - q_h^{\pi^k}(s,a)| \lesssim H \sum_{k=1}^{K} \sum_{h=1}^{H} \sum_{s\in\mathcal{S}, a\in\mathcal{A}} \epsilon_h^k(s,a) q_h^{\pi^k}(s,a)
$$

$$
+ HS \sum_{k=1}^{K} \sum_{1 \le h < \tilde{h} \le H} \sum_{s\in\mathcal{S}, a\in\mathcal{A}, s'\in\mathcal{S}} \sum_{\tilde{s}\in\mathcal{S}, \tilde{a}\in\mathcal{A}} \epsilon_h^k(s' \mid s,a) q_h^{\pi^k}(s,a) \min\left\{2, \sum_{\tilde{s}'\in\mathcal{S}} \epsilon_{\tilde{h}}^k(\tilde{s}' \mid \tilde{s}, \tilde{a})\right\} q_{\tilde{h}}^{\pi^k}(\tilde{s}, \tilde{a} \mid s'; h+1)
$$

$$
+ H^3 S^2 A d_{max} \tag{34}
$$

*where $q_{\tilde{h}}^{\pi^k}(\tilde{s}, \tilde{a} \mid \tilde{s}'; h)$ be the probability to visit $(\tilde{s}, \tilde{a})$ in time $\tilde{h}$ given that we visited $\tilde{s}'$ in time $h$, and $\epsilon_h^k(s' \mid s,a) = 16\sqrt{\frac{p_h(s'\mid s,a)\iota}{n_h^k(s,a)\vee 1}} + \frac{200\iota}{n_h^k(s,a)\vee 1}$*

*Proof.* Let $\mathcal{K}_{h,s,a} = \{k : s_h^k = s, a_h^k = a, m_h^k(s,a) \le d_{max}\}$ and define $\mathbb{I}_{h,s,a,k} = \mathbb{I}\{k \in \mathcal{K}_{h,s,a}\}$ and $\bar{\mathbb{I}}_{h,s,a,k} = 1 - \mathbb{I}_{h,s,a,k}$. Let $q^{k,s,h}$ be the occupancy measure such that $q_h^{k,s,h}(s) = u_h^k(s)$, and let $p^{k,s,h}$ be the transition that corresponds to $q^{k,s,h}$. Let $\sigma_h(s)$ be the set of all trajectories that end in $s$ in time $h$, i.e., $\sigma_h(s) = \{s_1, a_1, \ldots, s_{h-1}, a_{h-1}, s_h\}$ where $s_h = s$. We have:

$$
u_h^k(s,a) = q_h^{k,s,h}(s,a) = \pi_h^k(a \mid s) \sum_{\sigma_h(s)} \prod_{h'=1}^{h-1} \pi_{h'}^k(a_{h'} \mid s_{h'}) p_{h'}^{k,s,h}(s_{h'+1} \mid s_{h'}, a_{h'})
$$

$$
q_h^{\pi^k}(s,a) = \pi_h^k(a \mid s) \sum_{\sigma_h(s)} \prod_{h'=1}^{h-1} \pi_{h'}^k(a_{h'} \mid s_{h'}) p_{h'}(s_{h'+1} \mid s_{h'}, a_{h'}).
$$

Then,

$$
|u_h^k(s,a) - q_h^{\pi^k}(s,a)| = \pi_h^k(a \mid s) \sum_{\sigma_h(s)} \prod_{h'=1}^{h-1} \pi_{h'}^k(a_{h'} \mid s_{h'}) \left| \prod_{h'=1}^{h-1} p_{h'}^{k,s,h}(s_{h'+1} \mid s_{h'}, a_{h'}) - \prod_{h'=1}^{h-1} p_{h'}(s_{h'+1} \mid s_{h'}, a_{h'}) \right|.
$$

We can rewrite the following term as,

$$
\left| \prod_{h'=1}^{h-1} p^{k,s,h}(s_{h'+1} \mid s_{h'}, a_{h'}) - \prod_{h'=1}^{h-1} p_{h'}(s_{h'+1} \mid s_{h'}, a_{h'}) \right|
$$

$$
= \left| \sum_{l=2}^{h-1} \prod_{h'=1}^{l-1} p_{h'}(s_{h'+1} \mid s_{h'}, a_{h'}) \prod_{h'=l}^{h-1} p_{h'}^{k,s,h}(s_{h'+1} \mid s_{h'}, a_{h'}) + \prod_{h'=1}^{h-1} p_{h'}^{k,s,h}(s_{h'+1} \mid s_{h'}, a_{h'}) \right.
$$

$$
\left. - \prod_{h'=1}^{h-1} p_{h'}(s_{h'+1} \mid s_{h'}, a_{h'}) - \sum_{l=2}^{h-1} \prod_{h'=1}^{l-1} p_{h'}(s_{h'+1} \mid s_{h'}, a_{h'}) \prod_{h'=l}^{h-1} p_{h'}^{k,s,h}(s_{h'+1} \mid s_{h'}, a_{h'}) \right|
$$

$$
= \left| \sum_{l=1}^{h-1} \prod_{h'=1}^{l-1} p_{h'}(s_{h'+1} \mid s_{h'}, a_{h'}) \prod_{h'=l}^{h-1} p_{h'}^{k,s,h}(s_{h'+1} \mid s_{h'}, a_{h'}) \right.
$$

$$
\left. - \sum_{l=2}^{h} \prod_{h'=1}^{l-1} p_{h'}(s_{h'+1} \mid s_{h'}, a_{h'}) \prod_{h'=l}^{h-1} p_{h'}^{k,s,h}(s_{h'+1} \mid s_{h'}, a_{h'}) \right|
$$

$$
= \left| \sum_{l=1}^{h-1} \prod_{h'=1}^{l-1} p_{h'}(s_{h'+1} \mid s_{h'}, a_{h'}) \prod_{h'=l}^{h-1} p_{h'}^{k,s,h}(s_{h'+1} \mid s_{h'}, a_{h'}) \right.
$$

$$
\left. - \sum_{l=1}^{h-1} \prod_{h'=1}^{l} p_{h'}(s_{h'+1} \mid s_{h'}, a_{h'}) \prod_{h'=l+1}^{h-1} p_{h'}^{k,s,h}(s_{h'+1} \mid s_{h'}, a_{h'}) \right|
$$

$$
= \sum_{l=1}^{h-1} \left| p_l^{k,s,h}(s_{l+1} \mid s_l, a_l) - p_l(s_{l+1} \mid s_l, a_l) \right| \prod_{h'=1}^{l-1} p_h(s_{h'+1} \mid s_{h'}, a_{h'}) \prod_{h'=l+1}^{h-1} p_{h'}^{k,s,h}(s_{h'+1} \mid s_{h'}, a_{h'}).
$$

Hence,

$$
|u_h^k(s,a) - q_h^{\pi^k}(s,a)|
$$

$$
\leq \pi_h^k(a \mid s) \sum_{\sigma_h(s)} \prod_{h'=1}^{h-1} \pi_{h'}^k(a_{h'} \mid s_{h'}) \sum_{l=1}^{h-1} \left| p_l^{k,s,h}(s_{l+1} \mid s_l, a_l) - p_l(s_{l+1} \mid s_l, a_l) \right|
$$

$$
\cdot \prod_{h'=1}^{l-1} p_{h'}(s_{h'+1} \mid s_{h'}, a_{h'}) \prod_{h'=l+1}^{h-1} p_{h'}^{k,s,h}(s_{h'+1} \mid s_{h'}, a_{h'})
$$

$$
\leq \sum_{l=1}^{h-1} \sum_{\sigma_h(s)} \left| p_l^{k,s,h}(s_{l+1} \mid s_l, a_l) - p_l(s_{l+1} \mid s_l, a_l) \right| \left( \pi_l^k(a_l \mid s_l) \prod_{h'=1}^{l-1} \pi_{h'}^k(a_{h'} \mid s_{h'}) p_{h'}(s_{h'+1} \mid s_{h'}, a_{h'}) \right)
$$

$$
\cdot \left( \pi_h^k(a \mid s) \prod_{h'=l+1}^{h-1} \pi_{h'}^k(a_{h'} \mid s_{h'}) p_{h'}^{k,s,h}(s_{h'+1} \mid s_{h'}, a_{h'}) \right)
$$

$$
= \sum_{l=1}^{h-1} \sum_{s_l \in \mathcal{S}, a_l \in \mathcal{A}, s_{l+1} \in \mathcal{S}} \left| p_l^{k,s,h}(s_{l+1} \mid s_l, a_l) - p_l(s_{l+1} \mid s_l, a_l) \right|
$$

$$
\cdot \left( \sum_{\sigma_l(s_l)} \pi_l^k(a_l \mid s_l) \prod_{h'=1}^{l-1} \pi_{h'}^k(a_{h'} \mid s_{h'}) p_{h'}(s_{h'+1} \mid s_{h'}, a_{h'}) \right)
$$

$$
\cdot \left( \sum_{a_{l+1} \in \mathcal{A}} \sum_{\{s_{h''} \in \mathcal{S}, a_{h''} \in \mathcal{A}\}_{h''=l+2}^{h-1}} \pi_h^k(a \mid s) \prod_{h'=l+1}^{h-1} \pi_{h'}^k(a_{h'} \mid s_{h'}) p_{h'}^{k,s,h}(s_{h'+1} \mid s_{h'}, a_{h'}) \right)
$$

$$
= \sum_{l=1}^{h-1} \sum_{s_l \in \mathcal{S}, a_l \in \mathcal{A}, s_{l+1} \in \mathcal{S}} \left| p_l^{k,s,h}(s_{l+1} \mid s_l, a_l) - p_l(s_{l+1} \mid s_l, a_l) \right| q_l^{\pi^k}(s_l, a_l) \cdot q_h^{k,s,h}(s, a \mid s_{l+1}), \tag{35}
$$

where we ease notation and denote $q_h^{k,s,h}(s,a \mid s_{l+1}) = q_h^{k,s,h}(s,a \mid s_{l+1}; l+1)$. Similarly, we can show that,

$$|q_h^{k,s,h}(s,a \mid s_{l+1}) - q_h^{\pi^k}(s,a \mid s_{l+1})|$$

$$\lesssim \sum_{h'=l+1}^{h-1} \sum_{s_{h'}\in\mathcal{S},a_{h'}\in\mathcal{A},s_{h'+1}\in\mathcal{S}} \left| p_{h'}^{k,s,h}(s_{h'+1} \mid s_{h'},a_{h'}) - p_{h'}(s_{h'+1} \mid s_{h'},a_{h'}) \right| q_{h'}^{\pi^k}(s_{h'},a_{h'} \mid s_{l+1}) q_{h'}^{k,s,h}(s,a \mid s_{h'+1}) \tag{36}$$

$$\le \pi_h^k(a \mid s) \sum_{h'=l+1}^{h-1} \sum_{s_{h'}\in\mathcal{S},a_{h'}\in\mathcal{A},s_{h'+1}\in\mathcal{S}} \left| p_{h'}^{k,s,h}(s_{h'+1} \mid s_{h'},a_{h'}) - p_{h'}(s_{h'+1} \mid s_{h'},a_{h'}) \right| q_{h'}^{\pi^k}(s_{h'},a_{h'} \mid s_{l+1}), \tag{37}$$

where the last is since $q_{h'}^{k,s,h}(s,a \mid s_{h'+1}) \le \pi_h^k(a \mid s)$. Decomposing Eq. (35) for episodes $k \in \mathcal{K}_{l,s_l,a_l}$ and $k \notin \mathcal{K}_{l,s_l,a_l}$

$$\sum_{h,s,a,k} |u_h^k(s,a) - q_h^{\pi^k}(s,a)|$$

$$\lesssim \underbrace{\sum_{h,s,a,k} \sum_{l=1}^{h-1} \sum_{s_l\in\mathcal{S},a_l\in\mathcal{A},s_{l+1}\in\mathcal{S}} \mathbb{I}_{l,s_l,a_l,k} \left| p_l^{k,s,h}(s_{l+1} \mid s_l,a_l) - p_l(s_{l+1} \mid s_l,a_l) \right| q_l^{\pi^k}(s_l,a_l) \cdot q_h^{k,s,h}(s,a \mid s_{l+1})}_{(i)}$$

$$+ \underbrace{\sum_{h,s,a,k} \sum_{l=1}^{h-1} \sum_{s_l\in\mathcal{S},a_l\in\mathcal{A},s_{l+1}\in\mathcal{S}} \bar{\mathbb{I}}_{l,s_l,a_l,k} \left| p_l^{k,s,h}(s_{l+1} \mid s_l,a_l) - p_l(s_{l+1} \mid s_l,a_l) \right| q_l^{\pi^k}(s_l,a_l) \cdot q_h^{k,s,h}(s,a \mid s_{l+1})}_{(ii)}$$

Now,

$$(i) = \sum_{h,s,a,k} \sum_{l=1}^{h-1} \sum_{s_l\in\mathcal{S},a_l\in\mathcal{A},s_{l+1}\in\mathcal{S}} \mathbb{I}_{l,s_l,a_l,k} \left| p_l^{k,s,h}(s_{l+1} \mid s_l,a_l) - p_l(s_{l+1} \mid s_l,a_l) \right| q_l^{\pi^k}(s_l,a_l) \cdot q_h^{k,s,h}(s,a \mid s_{l+1})$$

$$\le \sum_{h,s,a,k} \sum_{l=1}^{h-1} \sum_{s_l\in\mathcal{S},a_l\in\mathcal{A},s_{l+1}\in\mathcal{S}} \mathbb{I}_{l,s_l,a_l,k} \left| p_l^{k,s,h}(s_{l+1} \mid s_l,a_l) - p_l(s_{l+1} \mid s_l,a_l) \right| q_l^{\pi^k}(s_l,a_l) \cdot \pi_h^k(a \mid s)$$

$$= \sum_{h,s,k} \sum_{l=1}^{h-1} \sum_{s_l\in\mathcal{S},a_l\in\mathcal{A},s_{l+1}\in\mathcal{S}} \mathbb{I}_{l,s_l,a_l,k} \left| p_l^{k,s,h}(s_{l+1} \mid s_l,a_l) - p_l(s_{l+1} \mid s_l,a_l) \right| q_l^{\pi^k}(s_l,a_l)$$

$$= 2S \sum_{h,k} \sum_{l=1}^{h-1} \sum_{s_l\in\mathcal{S},a_l\in\mathcal{A}} \mathbb{I}_{l,s_l,a_l,k} q_l^{\pi^k}(s_l,a_l)$$

$$\le 2S \sum_{h} \sum_{l=1}^{h-1} \sum_{s_l\in\mathcal{S},a_l\in\mathcal{A}} \sum_{k} \mathbb{I}_{l,s_l,a_l,k}$$

$$\le 4S \sum_{h} \sum_{l=1}^{h-1} \sum_{s_l\in\mathcal{S},a_l\in\mathcal{A}} d_{max} \le 4H^2 S^2 A d_{max},$$

where the third inequality is since $|\mathcal{K}_{h,s,a}| \le 2d_{max}$ for any $h, s$ and $a$. For $(ii)$ we first use Eq. (37) to bound,

$$(ii) \leq \underbrace{\sum_{h,s,a,k} \sum_{l=1}^{h-1} \sum_{s_l \in \mathcal{S}, a_l \in \mathcal{A}, s_{l+1} \in \mathcal{S}} \bar{\bar{\mathbb{I}}}_{l,s_l,a_l,k} \left| p_l^{k,s,h}(s_{l+1} \mid s_l, a_l) - p_l(s_{l+1} \mid s_l, a_l) \right| q_l^{\pi^k}(s_l, a_l) \cdot q_h^{\pi^k}(s, a \mid s_{l+1})}_{(iii)}$$

$$+ \sum_{h,s,a,k} \sum_{l=1}^{h-1} \sum_{s_l \in \mathcal{S}, a_l \in \mathcal{A}, s_{l+1} \in \mathcal{S}} \bar{\bar{\mathbb{I}}}_{l,s_l,a_l,k} \left| p_l^{k,s,h}(s_{l+1} \mid s_l, a_l) - p_l(s_{l+1} \mid s_l, a_l) \right| q_l^{\pi^k}(s_l, a_l) \pi_h^k(a \mid s)$$

$$\underbrace{\cdot \left( \sum_{h'=l+1}^{h-1} \sum_{s_{h'} \in \mathcal{S}, a_{h'} \in \mathcal{A}, s_{h'+1} \in \mathcal{S}} \left| p_{h'}^{k,s,h}(s_{h'+1} \mid s_{h'}, a_{h'}) - p_{h'}(s_{h'+1} \mid s_{h'}, a_{h'}) \right| q_{h'}^{\pi^k}(s_{h'}, a_{h'} \mid s_{l+1}) \right)}_{(iv)}$$

Now using Lemma D.10,

$$(iii) \leq \sum_{k,h} \sum_{l=1}^{h-1} \sum_{s_l \in \mathcal{S}, a_l \in \mathcal{A}, s_{l+1} \in \mathcal{S}} \bar{\bar{\mathbb{I}}}_{l,s_l,a_l,k} \epsilon_l^k(s_{l+1} \mid s_l, a_l) q_l^{\pi^k}(s_l, a_l) \cdot \left( \sum_{s,a} q_h^{\pi^k}(s, a \mid s_{l+1}) \right)$$

$$\leq H \sum_{k=1}^{K} \sum_{1 \leq l \leq H} \sum_{s_l \in \mathcal{S}, a_l \in \mathcal{A}, s_{l+1} \in \mathcal{S}} \epsilon_l^k(s_{l+1} \mid s_l, a_l) q_l^{\pi^k}(s_l, a_l)$$

$$= H \sum_{k=1}^{K} \sum_{h=1}^{H} \sum_{s \in \mathcal{S}, a \in \mathcal{A}, s' \in \mathcal{S}} \epsilon_h^k(s' \mid s, a) q_h^{\pi^k}(s, a)$$

For $(iv)$ we again devide into $k \in \mathcal{K}_{h',s_{h'},a_{h'}}$ and $k \notin \mathcal{K}_{h',s_{h'},a_{h'}}$,

$$(iv) = \sum_{h,s,a,k} \sum_{l=1}^{h-1} \sum_{s_l \in \mathcal{S}, a_l \in \mathcal{A}, s_{l+1} \in \mathcal{S}} \bar{\bar{\mathbb{I}}}_{l,s_l,a_l,k} \left| p_l^{k,s,h}(s_{l+1} \mid s_l, a_l) - p_l(s_{l+1} \mid s_l, a_l) \right| q_l^{\pi^k}(s_l, a_l) \pi_h^k(a \mid s)$$

$$\cdot \left( \sum_{h'=l+1}^{h-1} \sum_{s_{h'} \in \mathcal{S}, a_{h'} \in \mathcal{A}, s_{h'+1} \in \mathcal{S}} \mathbb{I}_{h',s_{h'},a_{h'},k} \left| p_{h'}^{k,s,h}(s_{h'+1} \mid s_{h'}, a_{h'}) - p_{h'}(s_{h'+1} \mid s_{h'}, a_{h'}) \right| q_{h'}^{\pi^k}(s_{h'}, a_{h'} \mid s_{l+1}) \right)$$

$$+ \sum_{h,s,a,k} \sum_{l=1}^{h-1} \sum_{s_l \in \mathcal{S}, a_l \in \mathcal{A}, s_{l+1} \in \mathcal{S}} \bar{\bar{\mathbb{I}}}_{l,s_l,a_l,k} \left| p_l^{k,s,h}(s_{l+1} \mid s_l, a_l) - p_l(s_{l+1} \mid s_l, a_l) \right| q_l^{\pi^k}(s_l, a_l) \pi_h^k(a \mid s)$$

$$\cdot \left( \sum_{h'=l+1}^{h-1} \sum_{s_{h'} \in \mathcal{S}, a_{h'} \in \mathcal{A}, s_{h'+1} \in \mathcal{S}} \bar{\bar{\mathbb{I}}}_{h',s_{h'},a_{h'},k} \left| p_{h'}^{k,s,h}(s_{h'+1} \mid s_{h'}, a_{h'}) - p_{h'}(s_{h'+1} \mid s_{h'}, a_{h'}) \right| q_{h'}^{\pi^k}(s_{h'}, a_{h'} \mid s_{l+1}) \right)$$

$$(38)$$

The first term is bounded in a similar way to $(i)$ by,

$$\sum_{h,s,a,k} \sum_{l=1}^{h-1} \sum_{s_l \in \mathcal{S}, a_l \in \mathcal{A}, s_{l+1} \in \mathcal{S}} \bar{\bar{\mathbb{I}}}_{l,s_l,a_l,k} \left| p_l^{k,s,h}(s_{l+1} \mid s_l, a_l) - p_l(s_{l+1} \mid s_l, a_l) \right| q_l^{\pi^k}(s_l, a_l) \pi_h^k(a \mid s)$$

$$\cdot \left( \sum_{h'=l+1}^{h-1} \sum_{s_{h'} \in \mathcal{S}, a_{h'} \in \mathcal{A}, s_{h'+1} \in \mathcal{S}} \mathbb{I}_{h',s_{h'},a_{h'},k} \left| p_{h'}^{k,s,h}(s_{h'+1} \mid s_{h'}, a_{h'}) - p_{h'}(s_{h'+1} \mid s_{h'}, a_{h'}) \right| q_{h'}^{\pi^k}(s_{h'}, a_{h'} \mid s_{l+1}) \right)$$

$$\leq 2 \sum_{h,s,a,k} \sum_{l=1}^{h-1} \sum_{s_l \in \mathcal{S}, a_l \in \mathcal{A}, s_{l+1} \in \mathcal{S}} \bar{\bar{\mathbb{I}}}_{l,s_l,a_l,k} \left| p_l^{k,s,h}(s_{l+1} \mid s_l, a_l) - p_l(s_{l+1} \mid s_l, a_l) \right| q_l^{\pi^k}(s_l, a_l) \pi_h^k(a \mid s)$$

$$\cdot \left( \sum_{h'=l+1}^{h-1} \sum_{s_{h'} \in \mathcal{S}, a_{h'} \in \mathcal{A}} \mathbb{I}_{h',s_{h'},a_{h'},k} q_{h'}^{\pi^k}(s_{h'}, a_{h'} \mid s_{l+1}) \right)$$

$$\leq 2 \sum_{h,s,k} \sum_{l=1}^{h-1} \sum_{s_l \in \mathcal{S}, a_l \in \mathcal{A}, s_{l+1} \in \mathcal{S}} \bar{\bar{\mathbb{I}}}_{l,s_l,a_l,k} \left| p_l^{k,s,h}(s_{l+1} \mid s_l, a_l) - p_l(s_{l+1} \mid s_l, a_l) \right| q_l^{\pi^k}(s_l, a_l) \cdot \left( \sum_{h'=l+1}^{h-1} \sum_{s_{h'} \in \mathcal{S}, a_{h'} \in \mathcal{A}} \mathbb{I}_{h',s_{h'},a_{h'}} \right)$$

$$\leq 4 \sum_{h,s,k} \sum_{l=1}^{h-1} \sum_{s_l \in \mathcal{S}, a_l \in \mathcal{A}} \bar{\bar{\mathbb{I}}}_{l,s_l,a_l,k} q_l^{\pi^k}(s_l, a_l) \cdot \left( \sum_{h'=l+1}^{h-1} \sum_{s_{h'} \in \mathcal{S}, a_{h'} \in \mathcal{A}} \mathbb{I}_{h',s_{h'},a_{h'}} \right)$$

$$\leq 4 \sum_{h,s,k} \sum_{l=1}^{h-1} \sum_{h'=l+1}^{h-1} \sum_{s_{h'} \in \mathcal{S}, a_{h'} \in \mathcal{A}} \mathbb{I}_{h',s_{h'},a_{h'},k}$$

$$= 4S \sum_{h} \sum_{l=1}^{h-1} \sum_{h'=l+1}^{h-1} \sum_{s_{h'} \in \mathcal{S}, a_{h'} \in \mathcal{A}} \sum_{k} \mathbb{I}_{h',s_{h'},a_{h'},k}$$

$$\leq 8S \sum_{h} \sum_{l=1}^{h-1} \sum_{h'=l+1}^{h-1} \sum_{s_{h'} \in \mathcal{S}, a_{h'} \in \mathcal{A}} d_{max} = 8H^3 S^2 A d_{max}$$

where the last inequality is since $|\mathcal{K}_{h,s,a}| \leq 2d_{max}$ for any $h, s$ and $a$. Again, using Lemma D.10, the second term in Eq. (38) is bounded by,

$$\sum_{h,s,k} \sum_{l=1}^{h-1} \sum_{s_l \in \mathcal{S}, a_l \in \mathcal{A}, s_{l+1} \in \mathcal{S}} \epsilon_l^k(s_{l+1} \mid s_l, a_l) q_l^{\pi^k}(s_l, a_l) \sum_a \pi_h^k(a \mid s)$$

$$\cdot \left( \sum_{h'=l+1}^{h-1} \sum_{s_{h'} \in \mathcal{S}, a_{h'} \in \mathcal{A}} \min \left\{ 2, \sum_{s_{h'+1} \in \mathcal{S}} \epsilon_{h'}^k(s_{h'+1} \mid s_{h'}, a_{h'}) \right\} q_{h'}^{\pi^k}(s_{h'}, a_{h'} \mid s_{l+1}) \right)$$

$$= HS \sum_{k=1}^{K} \sum_{1 \leq l < h' \leq H} \sum_{s_l \in \mathcal{S}, a_l \in \mathcal{A}, s_{l+1} \in \mathcal{S}} \sum_{s_{h'} \in \mathcal{S}, a_{h'} \in \mathcal{A}} \epsilon_l^k(s_{l+1} \mid s_l, a_l) q_l^{\pi^k}(s_l, a_l)$$

$$\cdot \min \left\{ 2, \sum_{s_{h'+1} \in \mathcal{S}} \epsilon_{h'}^k(s_{h'+1} \mid s_{h'}, a_{h'}) \right\} q_{h'}^{\pi^k}(s_{h'}, a_{h'} \mid s_{l+1})$$

$$= HS \sum_{k=1}^{K} \sum_{1 \leq h < \tilde{h} \leq H} \sum_{s \in \mathcal{S}, a \in \mathcal{A}, s' \in \mathcal{S}} \sum_{\tilde{s} \in \mathcal{S}, \tilde{a} \in \mathcal{A}} \epsilon_h^k(s' \mid s, a) q_h^{\pi^k}(s, a) \min \left\{ 2, \sum_{\tilde{s}' \in \mathcal{S}} \epsilon_{\tilde{h}}^k(\tilde{s}' \mid \tilde{s}, \tilde{a}) \right\} q_{\tilde{h}}^{\pi^k}(\tilde{s}, \tilde{a} \mid s'; h+1).$$

Summing the different terms completes the proof. $\square$

**Lemma D.12** (Lemma 4 of [22] adapted to delays). *With delayed trajectory feedback, under the good event,*

$$\sum_{h,s,a,k} |u_h^k(s,a) - q_h^{\pi^k}(s,a)| \lesssim \sqrt{H^4 S^2 AK\iota} + H^3 S^3 A\iota^2 + H^3 S^2 Ad_{max}.$$

*Proof.* Given Lemma D.11, the proof proceeds exactly like the proof of [27, Lemma E.5]. □