# OpenReview forum: "Near-Optimal Regret for Adversarial MDP with Delayed Bandit Feedback"
_NeurIPS.cc/2022/Conference — NeurIPS 2022 Accept_

### Official Review · Reviewer_sRmT · 2022-07-07

**Rating:** 6
**Confidence:** 4
**Soundness:** 3 good
**Presentation:** 3 good
**Contribution:** 2 fair

**Summary:**

This paper studies online learning for adversarial MDP with delayed bandit feedback in the tabular cases. It proposes three algorithms to solve the problem. The first one called Delayed Hedge is adapted from Hedge with optimism, which can provide high probability guarantee but is computationally inefficient. The second one called Delayed UOB-FTRL is adapted from FTRL with optimism, which is more computationally efficient but can only provide expected guarantee. Delayed UOB-REPS with Delayed-adapted Estimator improves the estimators in Delayed UOB-FTRL and is able to provide high probability guarantee. All these algorithms achieve better regrets than existing algorithms.

**Questions:**

1. The paper claims that Delayed UOB-FTRL and Delayed UOB-REPS with Delayed-adapted Estimator are computationally efficient. However, the updates of $q^{k+1}$ still seems a little difficult since the constraint set is $\Delta(M,k+1)$, whose geometry is unclear from the paper (the paper just mentions it is a polytope with polynomial constraints). I wonder the authors can elaborate on this.
2. The paper only studies the tabular cases, and their algorithms seem tailored to the tabular setting. I wonder whether their algorithms can be further extended to function approximation case.


**Limitations:**

See weakness.

**Strengths And Weaknesses:**

Strength:
1. The paper proposes three algorithms for adversarial MDP with delayed bandit feedback and achieves better regrets than existing algorithms.
2. The paper is well written, and the proof sketch is convincing.

Weakness:
There is still a gap between the regret upper bounds of these new algorithms and the theoretical lower bound. So maybe “near-optimal” seems a little exaggerated.

---

> ### Author Response · Authors · 2022-08-01
> **Response to reviewer sRmT**
>
> > *""near-optimal" seems a little exaggerated"*
>
> If the reviewers think it is necessary, we are willing to change the wording . However, we'll note that there are many papers in the field that use the term *"near-optimal"* when there are gaps of similar order to ours in $H,S$ and $A$. Take for example the seminal UCRL2 paper [19] (see paper's reference list) titled *"Near-optimal regret bounds for reinforcement learning"*, which has the same $\sqrt{S}$ gap as in our non-delayed term (note that the gaps on the delay term are even smaller).
>
> With that being said, we believe that the important thing is that we are very clear regarding our sub-optimality gaps. In particular, we indicate those in Table 1, at the introduction in line 51, at the proof of Theorem 4.1 (Delayed UOB-FTRL) in lines 208-211, after Theorem 5.1 (Delayed UOB-REPS) in line 277, and as the first open question in the future work (line 319).
>
>
> > Efficiency of the occupancy measure update step.
>
> The policy update using FTRL or OMD on the (confidence) set of occupancy measures was shown to be computationally efficient in several papers before us (e.g., [49, 34]), and since then this fact was used in much of the adversarial MDPs literature (e.g., [21, 24, 23, 26]), as well as some adversarial MAB papers [50].
>
> The optimization problem that defines the update step is convex with $O(H S^2 A)$ *linear constraints* that define the confidence set of occupancy measures. Thus, the efficient update step can be done using standard tools from classic convex optimization. We'll discuss this point in the main text, and explicitly write the linear constraints in the appendix (meanwhile we refer the reviewer to the beginning of the appendix of [26] for the explicit constraints).
>
> > Delays in adversarial MDPs with function approximation.
>
> This is a great and important question for future work. REPS methods do not naturally extend to function approximation, so we believe that involved adjustments are required for that.
>
> We actually believe that the more promising path towards that would be via Policy Optimization methods which extend to function approximation more naturally. This is another great reason to study the open question we raised on achieving optimal regret for policy optimization under bandit delayed feedback. We'll add this question to our discussion on future work.

---

> > ### Comment · Reviewer_sRmT · 2022-08-08
> > **Thanks for the response.**
> >
> > Thanks for the response. I have read other reviews and responses and decide to maintain the score.

---

### Official Review · Reviewer_dP9N · 2022-07-11

**Rating:** 8
**Confidence:** 4
**Soundness:** 4 excellent
**Presentation:** 4 excellent
**Contribution:** 4 excellent

**Summary:**

This paper considers online learning in the adversarial MDP setting with unknown transitions, adversarially changing costs, and (unbounded) obliviously adversarial delayed bandit feedback (only see costs along the observed trajectory rather than the whole cost function), and proves the first near-optimal regret bounds which scale as $\sqrt{K + D}$, for $K$ the number of episodes and $D$ the sum of delays over all episodes. This regret bound improves over the previously best known $(K + D)^{2/3}$ regret bound for this setting. Prior work studied the adversarial case under full-information feedback, and in that setting, achieved a $\sqrt{K + D}$ upper bound. In the prior work, it was proved that if the costs are changed from adversarial to stochastic, under bandit feedback, one can achieve a $\sqrt{K + D}$ upper bound, but if the costs remain adversarial, the paper only proved an upper bound of $(K + D)^{2/3}$.

The paper presents three algorithms: First, a Hedge-based algorithm which is adapted to the adversarial MDP which is computationally inefficient but is a simple solution to validate that an improvement in regret is possible; an FTRL-based algorithm which attains the improved regret bound and is computationally efficient, but does not have its regret bound hold with high probability, and finally, a REPS-based algorithm which attains all desired categories -- computational efficiency, (tightest) regret bound of all, and a high probability guarantee. Furthermore, the first term of the regret bound differs from the corresponding term in the lower bound only by order $\sqrt{HS}$, and the second term differs from the corresponding term in the lower bound only by order $(HSA)^{1/4}$.


**Questions:**

* Could you clarify in the main paper that $P^1$ is taken to be the set of all transition functions? It's currently buried in the more detailed version of Delayed Hedge (Alg 4) in the appendix - at least it should show up in Algorithm 1. I also think it would be helpful to qualify the definition of Bernstein confidence sets just a little more in the main paper to the unfamiliar reader; the intuition was immediately clear upon looking at Algorithm 5 in the appendix -- maybe consider saying that you're applying the Bernstein bound specifically to the empirical transitions on line 98?

* Comparing Theorem B.1 and Theorem 4.1, there is a term with $H^4S^2A$ and polylog dependency on $K$ in Theorem B.1 but not Theorem 4.1 -- shouldn't the worse dependencies be used here for the $\log(K)$ term (which is hidden via the tilde)?

* In step 2 of line 524 in the proof of Lemma 2, why is the log term log(HSA/delta) instead of log(SA/delta)? e.g. why do you need to union bound over all horizons as well?

* In step 5 of Eq. 13 in Line 552, why does the S show up inside the square root? Could you clarify what properties of $\eps_k$ you're using?

* Could you comment further on intuition for why OMD is preferable to FTRL (in lines 228-230)?

* Could you define $\phi$ directly in Algorithm 2?

* Based on the lower bound given in Table 1 and the comment in Line 278, is it the case that one expects the lower bound in Table 1 can be improved? Or is it the case that even in the non-delayed setting, it is believed that better algorithms in terms of first term of the regret bound exist?

* It might make sense to clarify that you are considering costs bounded in the range $[0, 1]$ when you describe the paper's setting.

* Minor things:
   * Line 15 has a typo "an inherit" --> "an inherent"
   * Line 19 has a typo "but assume" --> "but assumes"
   * Line 514: maybe use a different variable name for the coefficient?
   * Line 532: and Holder's inequality
   * in Line 539, might be helpful to mention that p' is the maximizer rather than just allude to u_k
   * Line 552: typo "forth" --> "fourth"





**Limitations:**

Yes, the authors have adequately clarified their contribution, and have not overstated any part of the work. There are no immediate negative societal consequences of this work.

**Strengths And Weaknesses:**

* Strengths: The paper is well written and gives solid theoretical results. I found the structure of the paper quite good, the related work well-cited and referenced, and I particularly thought the description of the FTRL analysis in Section 4 and the REPS analysis in Section 5 were well-written and clarified the main contributions in those sections. Several of the technical contributions are novel and appear to be interesting, and possibly more broadly useful beyond the scope of this specific paper. The paper is also useful pedagogically -- this paper appears to be a great introduction to the delayed adversarial online MDP literature, and possibly also to the more general adversarial online MDP literature via its clear breakdown of standard methodology and paper pointers. The delay-adapted estimator of Section 5 is quite elegant.

* Weaknesses: One weakness is that the proofs in the appendix are sometimes difficult to read due to many steps of the proofs citing specific lemmas and proofs in other papers -- this leads to a lot of pointer tracing to get to the desired statements. Additionally, it might be helpful if the reasoning behind each step in the proofs was less terse.

---

> ### Author Response · Authors · 2022-08-01
> **Response to reviewer dP9N**
>
> > Referencing specific lemmas and proofs from other papers.
>
> We thanks the reviewer for this comment. In the final version we will restate the main lemmas that we use or give an appropriate description of their claim's idea.
>
> > *"Could you clarify in the main paper that $P^1$ is taken to be the set of all transition functions?"*
>
> Yes, we will indicate that in the algorithms.
>
> > *"there is a term with $H^4S^2A$ and polylog dependency on $K$ in Theorem B.1 but not Theorem 4.1."*
>
> This is a rather standard burn-in term (e.g., see [21]) and is usually of low order. We avoid it in the main paper to simplify the bounds and provide the full bounds in the appendix. We'll clearly state in the main text of the final version.
>
>
> > *"In step 2 of line 524... why do you need to union bound over all horizons as well?"*
>
> Note that the cost also depend on the horizon $h$, and thus we need the union bound to be taken on the horizon as well. In Lemma 14 of [21] the state also encodes the horizon and thus $X$ in their case corresponds to $HS$ in our case.
>
> > *"In step 5 of Eq. 13 in Line 552, why does the S show up inside the square root? Could you clarify what properties of $\epsilon^k$ you're using?"*
>
>
> The $S$ factor comes from the following application of the Cauchy-Schwarz inequality: For any $x_1,...,x_n: \sum_{i=1}^n \sqrt{x_i} \leq \sqrt{n \sum_{i=1}^n x_i}$.
> More specifically, by the definition of $\epsilon^k$ in Eq. (11) (which unfortunately has a small typo: $p_h^k$ should be $p_h$), we have
> $$\epsilon_m^k(z|x,y) \leq \mathcal{O}\left( \sqrt{ \frac{p_m(z|x,y)\iota}{n_m^k(x,y)\lor 1} } +  \frac{\iota}{n_m^k(x,y)\lor 1}  \right).$$
> Taking the summation over all the next states $z$ and using the application of Cauchy-Schwarz mentioned above, we have
> $$\sum_{z} \epsilon_m^k(z|x,y)  \leq \mathcal{O}\left(\sqrt{ \frac{S\sum_{z}p_m(z|x,y)\iota}{n_m^k(x,y)\lor 1}} + \frac{S\iota}{n_m^k(x,y)\lor 1} \right)= \mathcal{O}\left(\sqrt{ \frac{S\iota}{n_m^k(x,y)\lor 1}} + \frac{S\iota}{n_m^k(x,y)\lor 1} \right),$$
> where the last step is by $\sum_{z}p_m(z|x,y)=1$.
> This is exactly the step 5 of Eq. (13).
>
> > *"Could you comment further on intuition for why OMD is preferable to FTRL (in lines 228-230)?"*
>
> OMD updates $q^{k+1}$ in terms of $q^k$ somewhat more explicitly (see the beginning of section D). At least technically, this more explicit relation between consecutive occupancy measure allows us to bound the KL-divergence between them. It is possible that FTRL also satisfies Lemma D.7 (or a similar claim), but unfortunately at this point we don't know if that's true.
>
>
> > *"Could you define $\phi$ directly in Algorithm 2?"*
>
> Yes, we'll do that in the later version. The specific definition of the regularizer $\phi$ in Algorithm 2 can be found in the Appendix, which is exactly the Shannon entropy regularizer $\frac{1}{\eta}\sum_{h,s,a,s'}q_h(s,a,s')\log q_h(s,a,s')$.
>
>
> > *"is it believed that better algorithms in terms of first term of the regret bound exist?"*
>
> Our belief is that the lower bound can be improved. This belief is based on the fact that in the reward free setting (see the paper of Jin et al., (2020) titled ``Reward-Free Exploration for Reinforcement Learning") the extra dependency in S is necessary, and the intuitive connection between having adversarial rewards (in our setting) and being able to find a good policy for any given reward function (in their setting).
>
>
> > *"clarify that you are considering costs bounded in the range $[0, 1]$"*
>
> This appears in line 72, but we'll clarify that also in an earlier point.
>
> We thank the reviewer for spotting typos and for the suggestions. All will be fixed in the final version.

---

> > ### Comment · Reviewer_dP9N · 2022-08-07
> > **Post-Rebuttal**
> >
> > Thanks for the clarifications. I read the other reviews and maintain my evaluation.

---

### Official Review · Reviewer_nrUe · 2022-07-11

**Rating:** 5
**Confidence:** 4
**Soundness:** 3 good
**Presentation:** 3 good
**Contribution:** 3 good

**Summary:**

This work focused on the adversarial MDPs with delayed and bandit feedback. Compared with previous results, the author proposes novel algorithms and improves the regret bound of $(K+D)^{2/3}$ to $\sqrt{K+D}$, which matches the lower bound on the factors K and D.

**Questions:**

1. For the previous work Lancewicki et al.[2022] , it can still obtain a $\sqrt{D+K}$ regret with full feedback. In this work, the author focuses on the bandit feedback and only improves it in this case. However, the distinction is not clear in the introduction or preliminaries.
Therefore, it is better if the author can also mention the full feedback and have some discussion about the difference between these settings.

Lancewicki T, Rosenberg A, Mansour Y. Learning adversarial Markov decision processes with delayed feedback. (AAAI 2022)

2. In the related work part, the author mentions recent advances in regret guarantees for adversarial MDPs. More recent advanced works have focused on using the function approximation techniques in Adversarial MDPs. For instance, He et al. [2022] improve the result in [5]
with a factor of $H$. It is better if the author can mention them or have some discussion about them.

 He J, Zhou D, Gu Q. Nearly Optimal Regret for Learning Adversarial MDPs with Linear Function Approximation. (AISTATS 2022)

**Limitations:**

This paper provides theoretical guarantees for learning linear bandit and linear mixture MDP. There is no negative societal impact.

**Strengths And Weaknesses:**

Strength:

1. All three novel algorithms obtain a $\sqrt{K+D}$ regret guarantee, which improved the previous results and matches the theoretical lower bound on the factors K and D.

2. In the Delayed UOB-REPS Algorithm, the author proposes a novel importance-sampling estimator, which can improve the stability with delay-feed back. This estimator may be of independent interest.

Weakness:

1. This work uses the word "near-optimal regret" or "order-optimal regret (Sections 3-4)" and it may overclaim the contribution. Near-optimal usually means it matches the theoretical lower bound up to logarithmic factors. However, for both three algorithms, gaps between the upper and lower bound on the elements "H, S, A" still exist, which is far from optimal.

2. The Delayed UOB-FTRL and UOB-REPS rely on a minimization over the confidence set with the FTRL algorithm. Therefore, these algorithms require the initial state is fixed ($s_{init}$), which is much more restrictive than previous works.

3. It is still unclear whether the Delayed UOB-FTRL and UOB-REPS are computation-efficient. Both of these algorithms rely on a minimization over the confidence set to compute the occupance measure. Though both the policy space and the possible transition function space $\mathcal{P}$ are continuous and convex, the relation between them and occupance measure $q$ is complicated, and the structure of $q$ may be unregular. Thus, it is still unclear how to perform the minimization over the confidence set efficiently.

---

> ### Author Response · Authors · 2022-08-01
> **Response to reviewer nrUe**
>
> > *"the word "near-optimal regret" overclaim the contribution"*
>
> This is simply a matter of terminology, and we are willing to change the wording if the reviewers think it is necessary. We'll note that there are many papers in the field that use the term "near-optimal" when there are gaps of similar order to ours in $H,S$ and $A$. Take for example the seminal UCRL2 paper [19] (see paper's reference list) titled *"Near-optimal regret bounds for reinforcement learning"*, which has the same $\sqrt{S}$ gap as in our non-delayed term (note that the gaps on the delay term are even smaller).
>
> With that being said, we believe that the important thing is that we are very clear regarding our sub-optimality gaps. In particular, we indicate those in Table 1, at the introduction in line 51, at the proof of Theorem 4.1 (Delayed UOB-FTRL) in lines 208-211, after Theorem 5.1 (Delayed UOB-REPS) in line 277, and as the first open question in the future work (line 319).
>
>
> > *"The algorithms require the initial state is fixed ($s_{init}$), which is much more restrictive than previous works."*
>
> Our analysis easily extends to the case where the initial state is random as well. This assumption is done *without lose of generality* to ease the notation. We would certainly clarify that in our final version. Please also note that this assumption is actually not restrictive compared to a random initial state: You can always increase the horizon by $1$ and think of the fixed initial state as a "dummy state" which transition to the actual (random) initial state at time $2$ (regardless on the chosen action).
>
> Possibly, the reviewer is referring to an *adversarial* initial state. However, most of the existing work on adversarial MDPs actually assume *random* initial state or *fixed* initial state (without lose of generality). Few papers (e.g., [20]) do handle adversarial initial state, but only in stochastic MDPs.
> It is indeed an interesting question weather this is attainable also in adversarial MDPs.
>
>
> > Efficiency of the occupancy measure update step.
>
> The policy update using FTRL or OMD on the (confidence) set of occupancy measures was shown to be computationally efficient in several papers before us (e.g., [49, 34]), and since then this fact was used in much of the adversarial MDPs literature (e.g., [21, 24, 23, 26]), as well as some adversarial MAB papers [50].
>
> The optimization problem that defines the update step is convex with $O(H S^2 A)$ *linear constraints* that define the confidence set of occupancy measures. Thus, the efficient update step can be done using standard tools from classic convex optimization. We thank the reviewer for this comment, and we'll address this point in the final version.
>
> > Distinction between full-information and bandit feedback.
>
> As we mention at the introduction, with full information the agent observes the whole cost function of the relevant episode, where in the more realistic bandit case the agent observes the cost only on the visited state-action pair. We'll clarify this distinction also in the preliminaries as the reviewer suggests.
>
> Note that Lancewicki et al. [2022] also indicate that handling delayed feedback using O-REPS (or FTRL) style method is *"extremely challenging and
> takes involved analysis"* and that *"Extending Delayed O-REPS to bandit feedback remains an important open problem"*. The fact that we consider bandit feedback only enhancesss these challenges, and indeed, Lancewicki et al. [2022] specifically mention the challenge of controlling the variance of the importance-sampling estimator under bandit feedback. Not only that we provide a novel involved analysis for delayed bandit feedback using these methods, we also present a novel algorithmic solution which directly reduces the variance of the estimator.
>
>
> > Related work of He et al. [2022].
>
> Thank you for pointing out this reference of He et al. [2021]. This is indeed an important contribution to the adversarial MDPs literature, as it establishes the optimal regret for adversarial MDPs with linear function approximation and shows that it is achievable with policy optimization methods. We would definitely include this reference in the final version of the paper.

---

> > ### Comment · Reviewer_nrUe · 2022-08-08
> > **Thank you for your answer**
> >
> > Thank you for your answer, I do not have further questions and I have increased my score.

---

### Meta-Review · Area_Chair_E55M · 2022-08-26

**Recommendation:** Accept
**Confidence:** Certain

**Metareview:**

This paper has initially received mixed reviews, but the author response has successfully addressed the concerns of the less enthusiastic reviewers, thus we have eventually reached consensus that the paper is suitable for publication at NeurIPS 2022. That said, I encourage the authors to take all the reviewers' comments into account when preparing the final version, especially when it comes to improving the readability and self-containedness of the proofs in the appendix. As for the usage of the term "near-optimal" in the title, I concur with the reviewers who pointed out that this may not be the perfect choice of words. This is not to say that it is necessary to change the title of the paper, but perhaps a better choice of wording may give a better overall impression to future readers of the paper.

**Award:**

No

---

### Decision · Program_Chairs · 2022-09-14

Accept